# VidEmo: Affective-Tree Reasoning for Emotion-Centric Video Foundation Models

**Zhicheng Zhang**[1*], **Weicheng Wang**[1*], **Yongjie Zhu**[3†]
**Wenyu Qin**[3], **Pengfei Wan**[3], **Di Zhang**[3], **Jufeng Yang**[124‡]
[1] Nankai University [2] Pengcheng Laboratory [3] Kuaishou Technology
[4] Nankai International Advanced Research Institute (SHENZHEN·FUTIAN)
gloryzzc6@sina.com, 2120230639@mail.nankai.edu.cn
{zhuyongjie,qinwenyu,wanpengfei}@kuaishou.com, yangjufeng@nankai.edu.cn
*https://zzcheng.top/VidEmo*

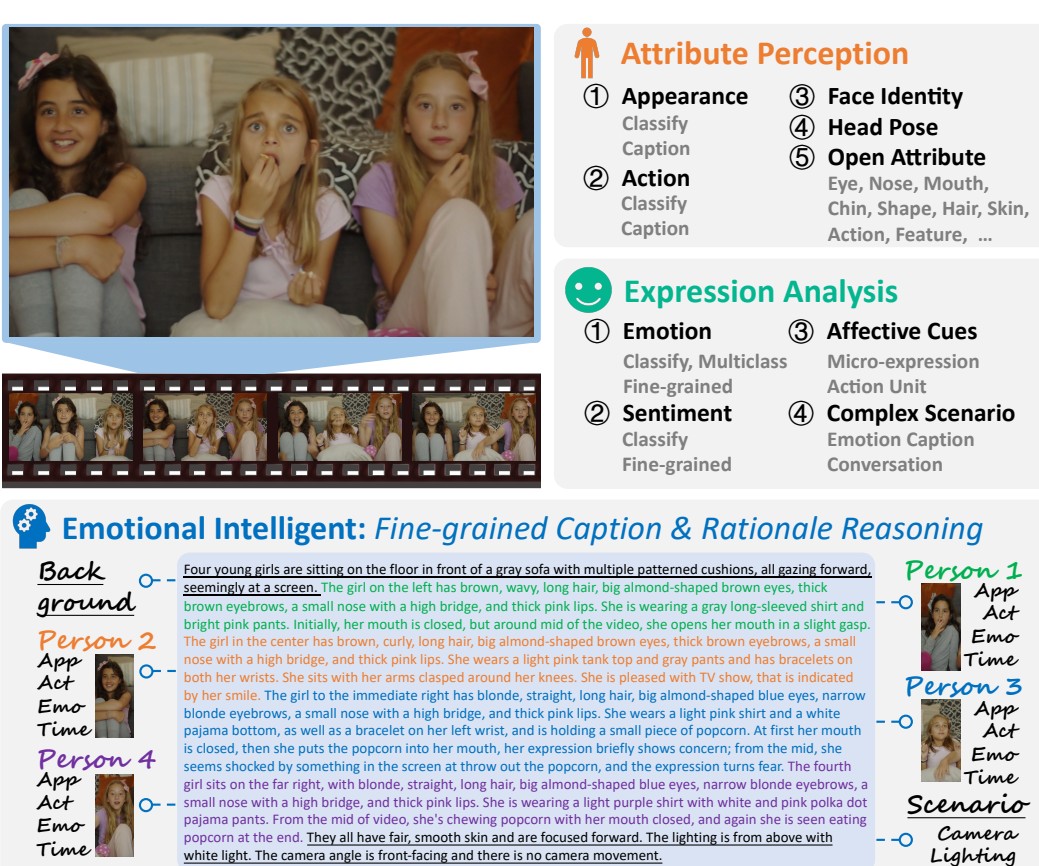

Figure 1: **Selected examples of inputs and outputs obtained from VidEmo.** Apart from providing toolkits for basic attribute perception and expression analysis (top), VidEmo extends the cognitive capacity and is able to generate fine-grained emotional captions with explainable rationale (bottom).

## Abstract

Understanding and predicting emotion from videos has gathered significant attention in recent studies, driven by advancements in video large language models (VideoLLMs). While advanced methods have made progress in video emotion analysis, the intrinsic nature of emotions poses significant challenges. Emotions

---

*: Equal Contribution. †: Project Leader. ‡: Corresponding Author.

are characterized by dynamic and cues-dependent properties, making it difficult to understand complex and evolving emotional states with reasonable rationale. To tackle these challenges, we propose a novel affective cues-guided reasoning framework that unifies fundamental attribute perception, expression analysis, and high-level emotional understanding in a stage-wise manner. At the core of our approach is a family of video emotion foundation models (VidEmo), specifically designed for emotion reasoning and instruction-following. These models undergo a two-stage tuning process: first, curriculum emotion learning for injecting emotion knowledge, followed by affective-tree reinforcement learning for emotion reasoning. Moreover, we establish a foundational data infrastructure and introduce a emotion-centric fine-grained dataset (Emo-CFG) consisting of 2.1M diverse instruction-based samples. Emo-CFG includes explainable emotional question-answering, fine-grained captions, and associated rationales, providing essential resources for advancing emotion understanding tasks. Experimental results demonstrate that our approach achieves competitive performance, setting a new milestone across 15 face perception tasks.

# 1  Introduction

Understanding and predicting human emotions from dynamic videos is an increasingly vital challenge in computer vision, with far-reaching applications in human-computer interaction, surveillance, and healthcare [17, 55, 58]. Despite the success of advanced methods [37, 74, 75], particularly in classifying basic emotional expressions, the ability to predict about complex, evolving emotional states with reasonable rationale remains limited. This is largely due to the dynamic and context-dependent nature of emotions [98, 35], which require models capable of providing both high-level emotional intelligence and rational, explainable outputs [68, 39]. Recently, the emergence of VideoLLMs [8, 65, 81, 86] has provided a promising baseline as a pathway. However, these foundational models often struggle with high-level emotional understanding, as they lack the ability to effectively combine basic facial attributes into representations of complex emotion. Even the cutting-edge milestone, Gemini 2.0 [62], achieves only an accuracy of 26.3% in fine-grained sentiment analysis, highlighting the gap in performance and the need for further innovation in this domain.

To address these challenges, we introduce VidEmo, a novel affective cues-guided reasoning framework based on tree-structure that integrates three core components: fundamental attribute perception, expression analysis, and high-level emotional understanding (see Fig. 1). Across 15 face perception tasks, VidEmo outperforms all existing open-source VideoLLMs, surpassing the previous state-of-the-art benchmark, *i.e.*, Gemini 2.0, as shown in Fig.2. To achieve this, Our VidEmo draws inspiration from recent work on reasoning models (R1), which excel at providing explainable rationales [18, 52, 64, 73]. These models solve complex tasks by incorporating a thinking process alongside the model's operation. Our finding demonstrates that this same reasoning process can be applied to high-level emotion understanding by introducing stage-wise thinking, structured around attribute perception [102, 5], expression analysis [4, 61], and emotion understanding [92, 70]. To be specific, we equip VidEmo with curriculum emotion learning and affective-tree reasoning, which inject emotion reasoning pathways during both the pre-train and post-train stages, respectively. In the pre-train stage, curriculum emotion learning progressively tunes the model from basic facial attributes to more complex emotional states. In the post-train stage, affective-tree reasoning helps the model refine its emotional understanding by using a hierarchical structure, ensuring that emotional responses are both accurate and interpretable. This two-stage process enables VidEmo to effectively analyze and reason about emotions in dynamic video data.

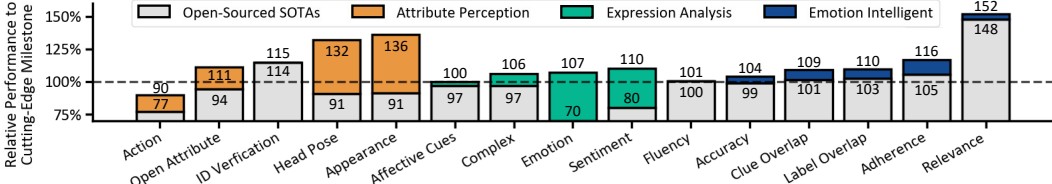

Figure 2: **Results Overview.** Our best model, *VidEmo*-T1, shows superior performance across 15 face perception tasks, surpassing advanced milestone (*Gemini 2.0: 5th Feb, 2025*) on 14 of 15 tasks.

To support our approach, we construct an emotion-centric fine-grained dataset, Emo-CFG, specifically designed to serve as the foundational data infrastructure for emotion understanding. Emo-CFG is a large-scale dataset consisting of 2.1 million diverse, characterized by emotion-centric labels, rigorous data verification, and high diversity, ensuring comprehensive and reliable annotations across a wide range of emotional contexts. By offering rich annotations and a wide variety of emotional contexts, Emo-CFG empowers VidEmo to effectively learn fine-grained emotion understanding from the emotion reasoning pathway.

Our contributions are two-fold: (1) We propose **VidEmo**, a novel affective cues-guided reasoning framework that combines curriculum emotion learning and affective-tree reasoning, enabling fine-grained and interpretable emotion understanding from dynamic video data. Experimental results show that VidEmo achieves over a 16.3% and 14.2% improvement compared to existing open-source VideoLLMs across 15 facial perception tasks on 1-3B and 7-8B scales. (2) We present **Emo-CFG**, a large-scale, emotion-centric dataset comprising 2.1M diverse samples with detailed annotations across attributes, expressions, and emotions, serving as a comprehensive data infrastructure for advancing emotion-centric video analysis.

## 2  Related Work

**Facial Video Analysis.** Face video analysis is a long standing problem towards high-level human understanding which involves various tasks, including attribute perception [102, 5], expression analysis [4, 61], and emotion understanding [92, 70, 26]. Various face perception models leverages strong backbone power for constructing multi-task framework [59]. Going forward to high-level emotion understanding [95, 98, 94], recent methods embrace MLLM [9, 33, 34, 93] for their strong zero-shot perception capacity [59, 88, 14]. EmotionLLaMA [10] introduces an emotion dataset including 28K coarse-grained and 4K fine-grained annotated datasets. OmniEmotion [80] proposes to explicitly integrate facial and audio modeling for emotion recognition. However, existing approaches are often constrained to a limited set of emotion categories or rely on static attribution perception. To advance cognitive human emotion understanding, we propose a fine-grained emotion-centric model empowered by dynamic attribution perception and emotion reasoning.

**Reasoning Model in MLLM.** With the blossom of a series of recent models such as DeepSeek-R1 and OpenAI o-series [52, 18], various works probe into integrating MLLMs with reasoning capacity [2]. Multimodal chain-of-thought (MCoT) prompting [27, 66, 46] offers a step-by-step reasoning trajectory when MLLM faces hard questions including detail grounding [72, 42], agent planing [27], etc. Specifically, MCoT decomposes the question into a group of reasoning steps and builds a chain to guide the model to generate the results of complex problems step-by-step [73, 64, 89]. For instance, LLaVA-CoT [76] prompts MLLMs reasoning steps into the summary, caption, reasoning, and conclusion stages and proposes a stage-level beam search strategy to further enhance reasoning capacity. In this paper, we propose affective cues-based rationale tree as intermediate bridge to meet the gap between abstract emotion and basic attribute.

## 3  VidEmo: Video Emotion Foundation Models

To develop a family of emotion-centric video foundation models, we propose a comprehensive set of toolkits designed for the pre-training, post-training, and reasoning stages, as illustrated in Fig. 3. Through a structured pre-training process, emotion knowledge is injected, followed by post-training to enhance the model's reasoning capabilities. Finally, the reasoning stage allows the model to effectively generate emotional outputs, leveraging learned attributes, expressions, and emotions.

### 3.1  Pre-training: Curriculum Emotion Learning

To inject emotion knowledge into the foundation model, we employ curriculum emotion learning to progressively tuning our base model. The training is structured into three stages: I) Attribute Tuning, II) Expression Tuning, and III) Emotion Tuning. The pre-training focuses on curating data that balances the difficulty of emotion tasks while addressing perplexity. At each stage, we carefully curate the data to ensure that the emotion-related tasks gradually increase in complexity. By starting with simpler attributes and progressively moving towards more complex expressions and emotions, we ensure that the model builds a strong foundational understanding of emotion, which

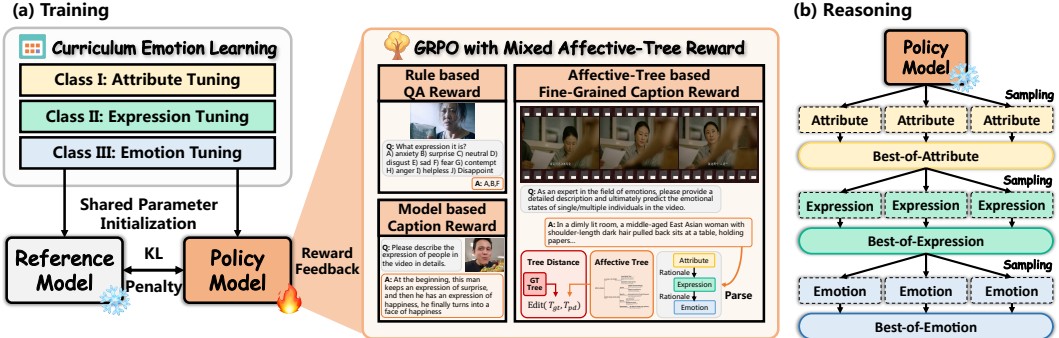

Figure 3: **Pipeline of VidEmo.** (a) Training: The model is trained using curriculum emotion learning, divided into three stages: attribute, expression, and emotion tuning. A reference model provides initial parameters, and a policy model is trained with reward feedback. (b) Reasoning: The policy model performs hierarchical reasoning by sampling from the best attributes, expressions, and emotions to generate the final emotional output.

facilitates smoother emotion knowledge injection throughout the process. Figure 4 presents the visualization results of our model across three key aspects: Attribute Perception, Expression Analysis, and Emotion Understanding.

**Attribute Perception**: The model accurately identifies facial attributes, such as hair color, length, and presence of bangs, with the ground truth comparison clearly shown for validation. For instance, the model correctly identifies a person's hair as blonde and shoulder-length, while also distinguishing the presence or absence of bangs.

**Expression Analysis**: The model analyzes subtle facial expressions, identifying features such as downward-tilted eyes and posture. These features, as seen in the second part of the figure, provide insight into the emotional states of the person, such as sadness or introspection, based on facial and contextual cues, like lighting and body movements.

**Emotion Understanding**: By combining the insights from facial features and contextual cues, the model provides a detailed interpretation of the emotional state. For example, in the final part of the figure, the model identifies a contemplative emotion, indicated by the subject's slightly tilted head, furrowed brows, and subtle eye movements.

## 3.2 Post-training: GRPO via Mixed Affective-Tree Reward

Building on emotion-knowledge-injected base models, we proceed to post-training to explore the emotional reasoning pathway. Recent reinforcement learning (RL) techniques [73] have demonstrated strong capabilities in reasoning, and GRPO [18] has garnered significant attention due to its simplicity and effectiveness. This makes GRPO an ideal starting point for our work.

Formally, let $q$ be a query, GRPO samples a group of outputs $\{o_i\}_{i=1}^{G}$ with the number of $G$ from the old policy model $\pi_{\theta_{old}}$, and train a policy model by maximizing the following objective:

$$\mathcal{J}_{GRPO}(\theta) = \mathbb{E}_{q,\{o_i\}\sim\pi_{\theta_{old}}}\left[\frac{1}{G}\sum_{i=1}^{G}\frac{1}{|o_i|}\sum_{t=1}^{|o_i|}\min\left(r_t(\theta)\hat{A}_{i,t}, \text{clip}(r_t(\theta), 1-\epsilon, 1+\epsilon)\hat{A}_{i,t}\right)\right] - \beta D_{KL}(\pi_\theta \| \pi_{ref}),$$

where $\hat{A}_{i,t}$ is the advantage based on relative rewards in group, $\epsilon$ and $\beta$ are coefficient of KL penalty and clip threshold, and $\pi_\theta, \pi_{\theta_{old}}, \pi_{ref}$ are current, old, and reference policy models, respectively.

**Rule based QA Reward.** The model is evaluated on its ability to respond to emotion-related queries using predefined rules of Acc and F1 score. The evaluation tasks include classification (single-label, multi-label), fine-grained classification, micro-expression detection, and action unit (AU) detection.

**Model based Caption Reward.** For the short caption of action, appearance, and emotion, we use a generative reward model to score the quality of captions generated by the model.

**Affective-Tree based Fine-Grained Caption Reward.** To assess the model's capacity for structured emotional reasoning, we introduce a reward mechanism based on a hierarchical affective tree constructed from fine-grained captions. Given a generated caption $\hat{o}$, we first parse it into a set of aspect–item pairs at three semantic levels: attribute ($\mathcal{A}$), expression ($\mathcal{E}$), and emotion ($\mathcal{M}$). These elements are organized into a three-level affective tree $T_{pred}$, where each node represents an extracted

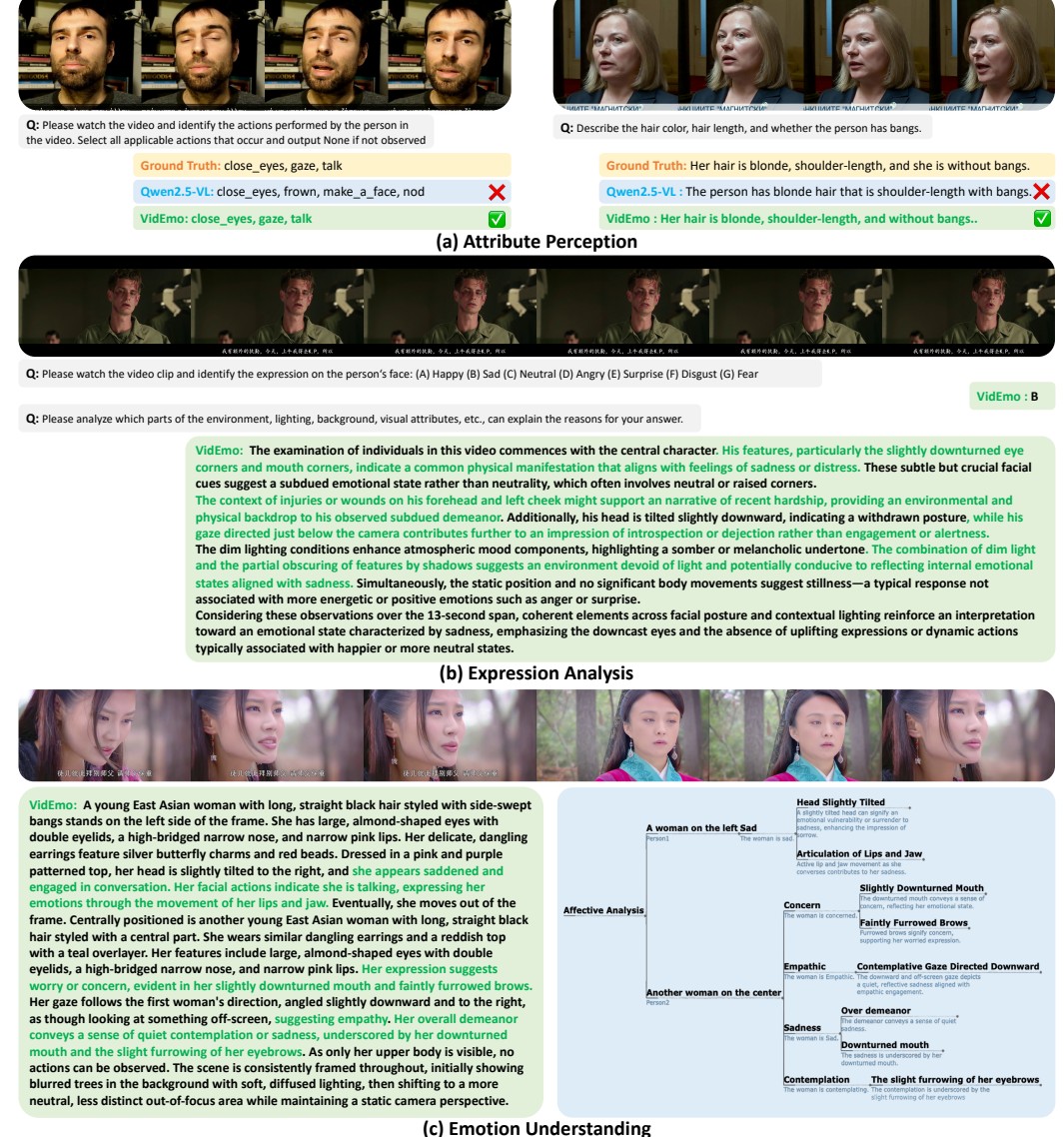

Figure 4: **Visualization** on attribute perception, expression analysis, and emotion understanding.

item and directed edges encode rationale-based dependencies:

$$\mathcal{A} \xrightarrow{\text{rationale}} \mathcal{E} \xrightarrow{\text{rationale}} \mathcal{M}. \tag{1}$$

We compare the predicted tree $T_{\text{pred}}$ with a ground-truth tree $T_{\text{gt}}$, parsed from human-annotated captions, using the tree edit distance [87] $\text{Edit}(T_{\text{gt}}, T_{\text{pred}})$, which quantifies the minimal number of edit operations (insertions, deletions, substitutions) required to transform one tree into the other.

The final reward $R$ is computed using an exponential decay over the tree distance:

$$R = \exp\left(-\lambda \cdot \text{Edit}(T_{\text{gt}}, T_{\text{pred}})\right), \tag{2}$$

where $\lambda > 0$ is a scaling factor controlling the reward sensitivity to tree differences. This formulation encourages the model to generate captions that are not only accurate in content but also structurally explainable, aligning with human reasoning patterns over emotional understanding.

### 3.3 Inference: Reasoning for High-level Emotion Understanding

Our VidEmo facilitates stage-wise training can be smoothly integrated with search-based reasoning strategy. Specifically, we adopt a hierarchical, search-based reasoning approach that decomposes

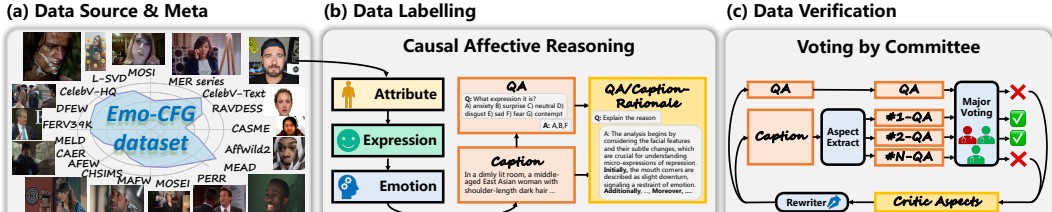

**(a) Data Source & Meta**   **(b) Data Labelling**   **(c) Data Verification**

Figure 5: **Data Curation Pipeline of the Emo-CFG dataset.** (a) The source of data from 17 datasets. (b) The illustration of data labeling steps. (c) The illustration of data verification loop.

emotional understanding into three levels: attribute perception, expression analysis, and emotion inference. At each level, the policy model samples multiple candidate outputs and selects the best one via a reward-guided scoring mechanism, forming a bottom-up reasoning trajectory. It is notice that we disable ER when comparing with other SOTA methods, for a fair comparison setting with only one model response are sampled.

# 4  Emo-CFG: Emotion-Centric Fine-Grained Video Dataset

The Emo-CFG dataset is designed to advance the understanding of emotional dynamics in the video. Motivated by the need for high-quality, emotion-centric data to train emotion reasoning models, Emo-CFG addresses key challenges of diverse emotional contexts, reliable annotations, and rigorous verification. We illustrate the data curation pipeline and statistics of Emo-CFG in Fig. 5 & Fig. 6.

**Data Source & Meta Information.** Our data collecting starts from high-quality video datasets. The data source include 17 datasets from head, avatar, and full-body avatar. By utilizing multiple data types, we ensure a holistic perspective to understanding the nuances of visual and emotional data. Further, we maintain the meta information of each video, including the face bounding box offset, video duration, video resolution, video fps.

$\mathcal{Caption}$ **&** $\mathcal{QA}$ **Instruction Data Labeling.** We utilize two primary data sources for labeling: large-scale, unlabelled datasets for broad coverage, and small-scale, fully labeled datasets for precision. For the labeled datasets, instruction pairs are generated using GPT-4o, which creates multiple templates, including multiple-choice questions, open-ended questions, and short captions. For the unlabelled datasets, we apply a causal affective reasoning strategy to generate labels in a sequential, stage-by-stage manner. Specifically, given a video, we first leverage the state-of-the-art Gemini 2.0 model, prompting it to generate fine-grained $\mathcal{Caption}$ data, focusing on attributes, expressions, and emotions in sequence. Subsequently, $\mathcal{QA}$ pairs are generated using GPT-4o, tailored to different aspects of the video. By combining these attribute and expression labels, the underlying emotion is accurately inferred, enabling a detailed and nuanced understanding of emotional states.

$\mathcal{Caption} - \mathcal{R}$ **&** $\mathcal{QA} - \mathcal{R}$ **Rationale Data Labeling.** Building upon the instruction data, we further explore the relationship between low-level attributes and high-level emotions. We prompt the advanced VideoLLM to conduct self-reflection on the rationale behind the emotional cues $\mathcal{Q} \xrightarrow{\mathcal{R}} \mathcal{A}$ and $\mathcal{Q} \xrightarrow{\mathcal{R}} \mathcal{Caption}$. This step not only enhances the model's interpretability by offering insights behind emotional expressions, but also serves as a crucial stage for enabling reasoning capacity.

$\mathcal{Critic}$ **Data Verification by Committee Voting.** To address the inherent ambiguities in emotional data, which arise from its subjective nature, we implement a committee voting-based verification strategy. We use three heterogeneous VideoLLMs as a committee to verify the correctness of the data and output $\mathcal{Critic}$ items, including incorrect answers and suggested corrections. Verified data is retained, while data that does not pass verification is sent back for rewriting based on the suggested corrections. Additionally, we extract different aspects of the caption data and separate them into multiple QA pairs to ensure alignment with the QA process.

**Data Statistics.** Fig. 6 provides key statistics of the Emo-CFG dataset. In (a), the data taxonomy organizes the dataset into three primary face perception tasks: Emotion Intelligence, Expression Analysis, and Attribution Perception, covering a wide range of facial features and emotional attributes. (b) The data distribution plots show the relative face area and video duration across different datasets, illustrating the diversity and variety of video data present in Emo-CFG. (c) The annotation distribution includes the breakdown of facial views (head, half, full) and video length, accompanied by a word cloud highlighting the most frequently annotated terms, such as "neutral", "face",

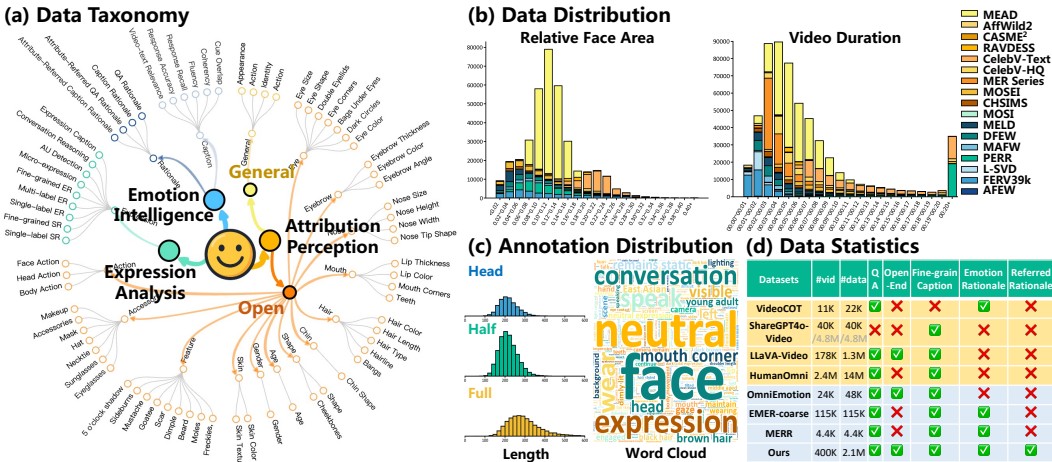

**Figure 6: Data Statistics of our Emo-CFG dataset.** (a) The data taxonomy from three types of face perception tasks. (b) The temporal and spatial distribution of video data. (c) The data label distribution and examples. (d) The comparison with other emotion and video datasets.

Table 1: **Comparison with 18 leading VideoLLMs on 14 face attribute perception tasks of Emo-CFG**, including 6 closed-set attribute perception tasks and 12 open attribute perception tasks. Cls: classification, Cap: caption, ID: identity verification, Pose: head pose estimation, AVG: average.

| Model | Size | Appearance Cls | Appearance Cap | Action Cls | Action Cap | ID Veri | Head Pose | AVG | Eye | Mout. | Nose | Hair | Chin | Shap. | Feat. | Acce. | Age | Gend. | Skin | Act. | AVG |
|---|---|---|---|---|---|---|---|---|---|---|---|---|---|---|---|---|---|---|---|---|---|
| *Closed MLLM API* | | | | | | | | | | | | | | | | | | | | | |
| Gemini 2.0 [62] | - | 42.2 | 56.4 | 41.4 | 62.2 | 86.5 | 25.8 | 52.4 | 72.1 | 72.9 | 87.3 | 87.3 | 64.1 | 61.9 | 72.1 | 80.0 | 78.7 | 94.3 | 81.5 | 61.6 | 76.2 |
| Claude 3.5 Sonnet [3] | - | 39.1 | 58.8 | 35.7 | 61.1 | 63.6 | 22.0 | 46.7 | 70.8 | 67.6 | 54.4 | 77.8 | 68.9 | 56.1 | 54.7 | 77.3 | 79.5 | 93.3 | 48.9 | 61.4 | 67.6 |
| Qwen-VL-MAX [63] | - | 41.1 | 54.3 | 32.5 | 60.0 | 89.7 | 34.5 | 52.0 | 71.9 | 69.6 | 79.4 | 84.4 | 64.0 | 71.5 | 64.4 | 71.7 | 78.9 | 93.3 | 74.0 | 60.9 | 73.7 |
| GPT-4o [51] | - | 29.3 | 51.5 | 05.1 | 40.7 | 79.0 | 27.6 | 38.8 | 51.5 | 45.7 | 62.9 | 77.1 | 48.5 | 46.4 | 54.5 | 78.1 | 68.5 | 86.3 | 75.5 | 52.8 | 62.3 |
| GPT-4o mini [51] | - | 20.5 | 55.8 | 04.1 | 54.0 | 69.3 | 27.8 | 38.6 | 43.2 | 44.5 | 52.0 | 52.4 | 31.6 | 43.5 | 40.1 | 45.0 | 41.0 | 69.6 | 45.9 | 30.7 | 44.9 |
| *Open-sourced 1-3B Video MLLM* | | | | | | | | | | | | | | | | | | | | | |
| LLaVA-OV [25] | 1B | 06.3 | 34.9 | 00.2 | 47.6 | 50.3 | 14.4 | 25.6 | 41.0 | 49.7 | 36.0 | 50.9 | 46.7 | 32.0 | 39.0 | 48.1 | 29.8 | 87.6 | 20.1 | 61.9 | 45.2 |
| InternVL2.5 [8] | 2B | 17.7 | 46.2 | 13.4 | 47.1 | 04.7 | 17.3 | 24.4 | 53.2 | 50.6 | 57.5 | 70.3 | 43.5 | 38.6 | 42.8 | 54.5 | 52.1 | 80.7 | 59.8 | 51.6 | 54.6 |
| VideoLLaMA3 [86] | 2B | 00.3 | 36.8 | 05.2 | 48.5 | 89.2 | 20.6 | 33.4 | 55.9 | 51.5 | 52.0 | 73.7 | 45.2 | 36.7 | 47.1 | 52.7 | 55.0 | 85.4 | 59.3 | 52.1 | 55.5 |
| mPLUG-Owl3 [81] | 2B | 16.0 | 45.4 | 13.8 | **52.3** | 76.4 | 07.7 | 35.3 | 54.4 | 61.4 | 55.4 | 71.4 | 43.7 | 45.4 | 52.3 | 60.8 | 39.8 | 91.6 | 58.7 | 48.5 | 56.9 |
| Qwen2.5-VL [65] | 3B | 43.6 | 41.1 | 30.2 | 49.9 | 95.7 | 15.5 | 46.0 | 64.2 | 54.3 | 51.3 | 72.8 | 29.1 | 40.8 | 52.1 | 58.9 | 70.6 | 93.5 | 76.2 | **62.6** | 60.5 |
| VidEmo-Base | 3B | **57.0** | **67.9** | 37.7 | 47.9 | 100 | 90.7 | 66.9 | 84.9 | 82.7 | 94.0 | 85.2 | 75.9 | 80.8 | 78.0 | 83.4 | 84.0 | 95.0 | 88.8 | 61.1 | 82.8 |
| *Open-sourced 7B+ Video MLLM* | | | | | | | | | | | | | | | | | | | | | |
| ShareGPT4Video [7] | 8B | 10.3 | 38.7 | 13.7 | 51.6 | 03.0 | 17.1 | 22.4 | 53.9 | 54.7 | 37.7 | 74.8 | 13.1 | 28.6 | 46.7 | 45.1 | 45.2 | 51.6 | 57.9 | 53.1 | 46.9 |
| InternVL2.5 [8] | 8B | 36.9 | 38.7 | 17.4 | 49.8 | 61.2 | 15.5 | 36.6 | 56.3 | 59.0 | 55.0 | 72.2 | 52.4 | 36.4 | 52.7 | 61.5 | 60.0 | 76.9 | 57.6 | 59.3 | 58.3 |
| LLaVA-N-Video [41] | 7B | 16.9 | 34.6 | 20.5 | 49.2 | 51.7 | 05.8 | 29.8 | 42.6 | 43.0 | 40.4 | 67.5 | 18.3 | 49.9 | 44.2 | 52.3 | 16.7 | 84.4 | 58.2 | 48.6 | 47.2 |
| LLaVA-OV [25] | 7B | 05.6 | 37.3 | 12.2 | 46.6 | 97.2 | 19.8 | 36.4 | 53.0 | 47.6 | 50.4 | 64.0 | 35.3 | 32.2 | 49.9 | 55.8 | 72.9 | 94.6 | 47.5 | 59.0 | 55.2 |
| VideoLLaMA3 [86] | 7B | 28.3 | 33.5 | 15.8 | 48.7 | 89.2 | 16.4 | 38.6 | 54.4 | 56.7 | 55.5 | 71.9 | 40.5 | 36.6 | 50.6 | 61.3 | 60.4 | 84.7 | 65.6 | 64.8 | 58.6 |
| LLaVA-Video [91] | 7B | 14.5 | 38.2 | 14.1 | 46.0 | 88.7 | 20.3 | 37.0 | 65.0 | 57.0 | 66.2 | 72.6 | 21.3 | 29.4 | 59.3 | 68.3 | 79.4 | 93.0 | 62.5 | 63.8 | 61.5 |
| mPLUG-Owl3 [81] | 7B | 34.3 | 41.6 | 21.3 | 55.1 | 66.4 | 21.1 | 40.0 | 55.0 | 56.2 | 48.0 | 70.5 | 39.9 | 48.1 | 50.5 | 58.8 | 61.8 | 89.7 | 62.0 | 60.8 | 58.4 |
| Qwen2.5-VL [65] | 7B | 44.7 | 45.2 | 21.0 | 52.3 | 99.7 | 22.6 | 47.6 | 68.6 | 70.5 | 83.2 | 74.7 | 66.4 | 55.6 | 60.8 | 73.8 | 77.2 | 94.0 | 76.2 | 64.0 | 72.1 |
| VidEmo-Base | 7B | 60.3 | 72.9 | 38.4 | 55.1 | **99.8** | 93.4 | 69.2 | 86.4 | 85.5 | 95.1 | 85.1 | 77.3 | 81.6 | 78.7 | 85.0 | 85.6 | 95.0 | 89.5 | 71.7 | 84.7 |
| VidEmo-T1 | 7B | **64.8** | **73.1** | **41.4** | 57.4 | 99.7 | **96.7** | 72.1 | 88.2 | 87.8 | 95.6 | 87.8 | 79.2 | 82.0 | 80.8 | 85.7 | 86.9 | 97.0 | 90.4 | 74.3 | 86.3 |

and "expression". (d) Data statistics compares Emo-CFG with other emotion and video datasets, showing that Emo-CFG provides a richer set of annotations and label types, including fine-grained emotion, rationales, and comprehensive video data, making it a unique and valuable resource for emotion-centric research. More details can be refer to Sec. B of appendix.

# 5 Experiment

As shown in Tab. 1 and 2, we conduct experiments to verify the effectiveness of VidEmo on three types of tasks: attribute perception, expression analysis, and emotion understanding. We compare VidEmo with 5 closed MLLM APIs and 13 open-sourced VideoLLMs with scales ranging from 1B to 8B. More training details (Sec. A) and evaluation settings (Sec. D) please refer to appendix.

## 5.1 SOTA Comparison

We benchmark VidEmo on 40 metrics, spanning 6 closed-set attribute tasks, 12 open-set attribute tasks, 9 expression tasks, and 6 high-level emotion understanding tasks.

Table 2: **Comparison with 18 leading VideoLLMs on 11 expression analysis tasks and 6 fine-grained emotion understanding tasks of Emo-CFG**. Sin: single-label classification, Mul: multi-label classification, Fine: fine-grained classification, Mic: micro-expression detection, AU: action unit detection, Cap: caption, Conv: conversation emotion analysis, VTR: video-text relevance, Flu: fluency, RA: response accuracy, IA: instruction adherence, Clu: clue overlap, Lab: label overlap, AVG: average.

| Model | Size | Emotion | | | AVG | Sentiment | | AVG | Cues | | AVG | Complex | | AVG | Emotion Understanding | | | | | | AVG |
|---|---|---|---|---|---|---|---|---|---|---|---|---|---|---|---|---|---|---|---|---|---|
| | | Sin | Mul | Fine | | Sin | Fine | | Mic | AU | | Cap | Conv | | Lab | Clu | IA | RA | VTR | Flu | |
| *Closed MLLM API* | | | | | | | | | | | | | | | | | | | | | |
| Gemini 2.0 [62] | - | 45.3 | 42.2 | 23.6 | 37.0 | 45.0 | 26.0 | 35.5 | 16.2 | 36.1 | 26.1 | 42.0 | 50.6 | 46.3 | 51.0 | 52.6 | 58.2 | 60.2 | 66.8 | 92.1 | 63.5 |
| Claude 3 Sonnet [3] | - | 26.6 | 17.6 | 16.6 | 20.3 | 37.6 | 23.3 | 30.4 | 29.2 | 37.1 | 33.2 | 50.3 | 36.3 | 43.3 | 47.5 | 49.5 | 57.6 | 54.9 | 62.0 | 92.5 | 60.7 |
| Qwen-VL-MAX [63] | - | 29.3 | 25.8 | 24.6 | 26.6 | 32.6 | 21.3 | 27.0 | 11.7 | 24.0 | 17.9 | 46.4 | 49.3 | 47.8 | 47.9 | 50.7 | 57.4 | 56.6 | 63.6 | 92.0 | 61.4 |
| GPT-4o [51] | - | 20.6 | 19.0 | 04.3 | 14.6 | 29.6 | 18.0 | 23.8 | 29.6 | 02.3 | 16.0 | 38.1 | 43.6 | 40.8 | 30.0 | 35.4 | 40.8 | 43.2 | 50.9 | 87.4 | 48.0 |
| GPT-4o mini [51] | - | 19.6 | 20.2 | 07.3 | 15.7 | 29.9 | 16.6 | 23.3 | 21.9 | 08.9 | 15.4 | 40.7 | 47.6 | 44.2 | 35.5 | 39.0 | 46.2 | 46.7 | 54.4 | 91.0 | 52.1 |
| *Open-sourced 1-3B Video MLLM* | | | | | | | | | | | | | | | | | | | | | |
| LLaVA-OV [25] | 1B | 28.3 | 17.5 | 05.3 | 17.0 | 36.3 | 15.0 | 25.6 | 13.4 | 15.8 | 14.6 | 40.2 | 29.6 | 34.9 | 17.2 | 18.2 | 28.6 | 23.5 | 28.8 | 89.0 | 34.2 |
| InternVL2.5 [8] | 2B | 23.0 | 16.1 | 08.3 | 15.8 | 31.6 | 15.6 | 23.6 | 09.7 | 07.4 | 08.5 | 43.6 | 26.0 | 34.8 | 40.1 | 42.9 | 51.6 | 49.6 | 56.4 | 90.8 | 55.2 |
| VideoLLaMA3 [86] | 2B | 29.3 | 18.3 | 11.6 | 19.7 | 34.3 | 16.0 | 25.1 | 06.5 | 06.5 | 06.5 | 20.1 | 31.3 | 25.7 | 24.0 | 22.4 | 29.5 | 33.4 | 40.3 | 86.8 | 39.4 |
| mPLUG-Owl3 [81] | 2B | 32.3 | 18.4 | 05.0 | 18.6 | 35.6 | 07.6 | 21.6 | **28.0** | 22.7 | 25.3 | 45.0 | 34.3 | 39.7 | 27.8 | 27.9 | 35.8 | 34.6 | 41.3 | 89.9 | 42.9 |
| Qwen2.5-VL [65] | 3B | 36.0 | 23.7 | 09.3 | 23.0 | 36.6 | 20.3 | 28.5 | 12.1 | 23.0 | 17.6 | 44.2 | 40.3 | 42.2 | 38.2 | 40.9 | 49.7 | 49.2 | 56.3 | 91.4 | 54.3 |
| VidEmo-Base | 3B | **46.0** | **38.0** | **26.0** | 36.6 | **40.3** | **32.6** | 36.5 | 22.3 | **32.1** | 27.2 | **48.1** | **42.0** | **45.0** | **57.3** | **59.6** | **70.7** | **62.7** | **68.1** | **93.1** | **68.6** |
| *Open-sourced 7B+ Video MLLM* | | | | | | | | | | | | | | | | | | | | | |
| ShareGPT4Video [7] | 8B | 07.6 | 06.0 | 04.6 | 06.1 | 38.0 | 14.3 | 26.1 | 09.7 | 01.4 | 05.6 | 46.2 | 32.3 | 39.2 | 16.1 | 18.8 | 34.4 | 21.8 | 26.0 | 91.1 | 34.7 |
| InternVL2.5 [8] | 8B | 28.0 | 26.2 | 09.0 | 21.0 | 29.3 | 18.3 | 23.8 | 16.2 | 12.8 | 14.5 | 40.8 | 35.0 | 37.9 | 52.3 | 53.4 | 61.5 | 59.8 | 66.1 | 92.4 | 64.2 |
| LLaVA-N-Video [41] | 7B | 24.3 | 23.7 | 10.6 | 19.5 | 39.0 | 14.0 | 26.5 | 10.9 | 13.3 | 12.1 | 44.1 | 39.0 | 41.5 | 33.7 | 33.2 | 43.3 | 38.8 | 45.2 | 90.9 | 47.5 |
| LLaVA-OV [25] | 7B | 31.6 | 22.7 | 10.3 | 21.5 | 36.0 | 20.0 | 28.0 | 11.7 | 15.5 | 13.6 | 42.2 | 43.0 | 42.6 | 36.5 | 39.3 | 49.5 | 46.2 | 53.1 | 91.0 | 52.6 |
| VideoLLaMA3 [86] | 7B | 27.6 | 23.9 | 10.6 | 20.7 | 31.0 | 19.3 | 25.1 | 08.9 | 10.8 | 09.8 | 33.2 | 36.0 | 34.6 | 42.2 | 44.4 | 53.1 | 52.9 | 59.5 | 89.5 | 56.9 |
| LLaVA-Video [91] | 7B | 32.6 | 22.9 | 09.3 | 21.6 | 35.3 | 20.6 | 28.0 | 12.1 | 02.5 | 07.3 | 45.8 | 42.0 | 43.9 | 38.8 | 40.9 | 50.1 | 47.8 | 54.6 | 91.2 | 53.9 |
| mPLUG-Owl3 [81] | 7B | 30.0 | 22.2 | 10.8 | 21.0 | 29.3 | 15.6 | 22.5 | **21.5** | 25.1 | 23.3 | 47.2 | 33.6 | 40.4 | 36.5 | 35.6 | 44.2 | 43.8 | 50.3 | 90.8 | 50.2 |
| Qwen2.5-VL [65] | 7B | 38.6 | 27.0 | 12.3 | 26.0 | 30.0 | 22.3 | 26.1 | 10.5 | 14.4 | 12.5 | 46.2 | 44.3 | 45.2 | 50.7 | 52.1 | 60.0 | 59.7 | 66.3 | 92.7 | 63.6 |
| VidEmo-Base | 7B | 47.3 | 37.6 | 34.6 | 39.8 | **42.3** | 36.0 | 39.1 | 18.2 | **34.2** | 26.2 | 50.0 | 48.6 | 49.3 | 55.9 | 57.4 | 67.9 | 62.6 | 68.3 | **92.8** | 67.5 |
| VidEmo-T1 | 7B | **49.7** | **38.8** | **35.6** | **41.3** | **42.3** | **37.5** | **39.9** | 20.4 | 34.1 | **27.3** | **50.7** | **52.9** | **51.8** | **59.3** | **61.2** | **68.1** | **65.9** | **69.1** | 92.6 | **69.3** |

**Scale**: Our models significantly outperform existing closed and open-source VideoLLMs across all metrics from 1B to 8B scales. At the 1-3B / 7-8B scale, our VidEmo-Base model (3B/7B) achieves an overall average accuracy of 62.4%/64.1%, outperforming the strongest baseline, Qwen2.5-VL at 46.1%/51.7%, by a margin of +16.3%/+12.4%. The consistent improvement across scales demonstrate the effectiveness of our proposed curriculum learning as well as affective-tree reward in scaling up our foundation model.

**Attribute & Expression & Emotion Tasks**: We conduct a comprehensive analysis of VidEmo across three core task categories in the Emo-CFG benchmark: attribute perception, expression analysis, and emotion understanding. In attribute tasks, which include both closed-set (e.g., identity, head pose) and open-set (e.g., hair type, age, skin tone) recognition, VidEmo achieves an average score of 86.3%, surpassing all baselines including Qwen2.5-VL (7B) at 80.6%, yielding a +5.7% improvement. Particularly, our model achieves 99.7% on identity verification, 95.6% on facial shape, and 97.0% on gender prediction, reflecting strong generalization on fine-grained visual perception. In expression analysis, covering single-label, multi-label, and fine-grained classification, as well as micro-expression and AU detection, VidEmo delivers an average of 39.9%, outperforming Qwen2.5-VL (7B) by +6.8%. Notably, VidEmo leads in fine-grained expression classification (35.6% vs. 29.7%) and micro-expression detection (20.4% vs. 13.6%), demonstrating its sensitivity to subtle and transient affective cues. In emotion understanding tasks—spanning instruction-following, fluency, response accuracy, and video-text relevance—VidEmo achieves 69.3% on average, outperforming all prior models, including Gemini 2.0 (63.5%) and Qwen2.5-VL (7B) (63.6%), with improvements of over +5%. It sets new benchmarks on tasks like instruction adherence (68.1%), fluency (69.1%), and video-text relevance (69.3%), showcasing its capacity for coherent, explainable, and semantically grounded emotional inference. We also notice that for the attribute tasks achieves an higher averaged performance. This finding also aligns with the dynamics we observed during training, that the perplexity of model increase with a sequential order of attribute, expression, and emotion.

## 5.2 Discussion

**Open-sourced Models and Closed Models**: We evaluate both open-sourced and closed multi-modal large models (MLLMs) on the Emo-CFG benchmark. Closed models, including Gemini 2.0, Claude 3 Sonnet, GPT-4o, GPT-4o mini, and Qwen-VL-MAX, typically operate as APIs with proprietary architectures and in-distribution training data. In contrast, open-sourced models span both small and large-scale variants (1–7B parameters), including LLaVA-OV, InternVL2.5, Video-oLLaMA3, mPLUG-Owl, Qwen-VL, and our own VidEmo series. Across all three evaluation categories—attribute perception, expression analysis, and emotion understanding—our open-sourced VidEmo-T1 (7B) outperforms all closed-source models. For instance, VidEmo-T1 achieves 86.3% in attribute perception, surpassing Gemini 2.0 by +9.8%, and obtains 39.9% on expression tasks, outperforming Claude 3 by +16.6%. Notably, in high-level emotion understanding, VidEmo-T1 reaches 69.3%, exceeding GPT-4o's 48.0% by a margin of +21.3%.

**Base Model and Reasoning Model** We further compare our base model (VidEmo-Base, 7B) with the reasoning-enhanced model (VidEmo-T1) to assess the effectiveness of affective-tree reasoning. In attribute perception, VidEmo-T1 improves the average performance from 84.7% to 86.3%, with notable gains in key tasks such as head pose estimation (+1.9%, from 93.4 to 96.7), facial feature prediction (+1.3%, from 85.6 to 86.9), and gender recognition (+1.9%, from 89.5 to 90.4). In expression analysis, VidEmo-T1 shows consistent improvements across all sub-tasks. The average increases from 39.1% to 41.3%, with gains in micro-expression detection (+2.2%) and fine-grained expression recognition (+1.0%). Most notably, in emotion understanding, VidEmo-T1 achieves a substantial improvement from 67.5% to 69.3%, with strong gains in fluency (+2.3%, from 67.9 to 70.2), video-text relevance (+2.6%), and instruction adherence (+1.9%).

**Downstream Tasks.** For downstream emotion-related tasks, we evaluate the capacity of VidEmo on facial expression recognition using the DFEW and MAFW datasets. When fine-tuned for specific tasks, VidEmo consistently outperforms both traditional video-based expression recognition methods and zero-shot emotion-oriented CLIP approaches. As shown in Table 3, VidEmo achieves a performance gains compared to state-of-the-art baselines (EMO-LLaMA), with an average improvement of 9.4% over the previous best results across both datasets. Notably, VidEmo achieves relative improvements of 7.8% in UAR and 10.9% in WAR on DFEW, and 5.9% in UAR and 12.8% in WAR on MAFW, demonstrating its superior effectiveness for facial expression recognition tasks.

Table 3: **Downstream tasks on DFEW and MAFW datasets.**

| Model | DFEW | | MAFW | |
|---|---|---|---|---|
| | UAR | WAR | UAR | WAR |
| EmotionCLIP [90] | 13.77 | 19.89 | 9.20 | 11.65 |
| Exp-CLIP [99] | 24.25 | 25.87 | 17.53 | 20.27 |
| EmoCLIP [15] | 36.76 | 46.27 | 25.86 | 33.49 |
| EmoCapCLIP [60] | 42.19 | 43.99 | 30.85 | 34.50 |
| I3D [6] | 46.52 | 58.27 | - | - |
| F-DFER [100] | 53.69 | 65.70 | - | - |
| EST [45] | 53.43 | 65.85 | - | - |
| IAL [28] | 55.71 | 69.24 | - | - |
| CLIPER [29] | 57.56 | 70.84 | - | - |
| DFER-CLIP [101] | 59.61 | 71.25 | 38.89 | 52.55 |
| EMO-LLaMA [75] | 60.23 | 65.89 | 41.57 | 48.63 |
| VidEmo (Ours) | **64.92** | **73.10** | **44.02** | **54.86** |

## 5.3 Ablation Study

To investigate the effectiveness of curriculum emotion learning (CEL), affective-tree reward (ATR), and emotion reasoning (ER), we conduct a component-wise ablation study, as presented in Tab. 4 and more in-depth analysis in Tab. 8, Tab. 9 and Tab. 10 of appendix. In this study, we assess the contribution of each component by analyzing the performance of the model under different configurations, removing one or more of the components. With the ablation studies conducted, we find four interesting observations:

- When none of the components are used, the model achieves an average performance of 51.4. This baseline highlights the importance of incorporating these components into the model for improved performance.

- With the inclusion of CEL alone, the model performance increases to an average of 61.9, demonstrating the positive impact of curriculum emotion learning on the model's ability to handle emotional contexts. Specifically, we observe improvements in the emotion-related metrics, particularly in the expression and emotion attributes.

Table 4: **Ablation study on the proposed components for our VidEmo.** CEL = curriculum emotion learning, ATR = affective-tree reward, ER = emotion reasoning.

| Components | | | Att | Exp | Emo | Avg |
|---|---|---|---|---|---|---|
| CEL | ATR | ER | | | | |
| | | | 63.5 | 27.3 | 63.6 | 51.4 |
| ✓ | | | 79.5 | 38.7 | 67.5 | 61.9 |
| ✓ | ✓ | | 81.3 | 40.1 | 69.3 | 63.6 |
| ✓ | ✓ | ✓ | 84.5 | 43.8 | 72.9 | 67.0 |

Table 5: **User study across three dimensions between Emo-CFG and CelebV-Text.** We evaluate the label quality on precision, rationality, and complementary through pairwise comparison with 50 videos and 25 users. All three dimensions show statistically significant preference for Emo-CFG.

| Dimension | Pairwise | | | #Vid | #Usr | Prefer | p-value |
|---|---|---|---|---|---|---|---|
| | Win | Tie | Loss | | | | |
| Precision | 964 | 204 | 82 | 50 | 25 | 95.5% | 0.00021 |
| Rationality | 1082 | 87 | 81 | 50 | 25 | 92.1% | 0.00015 |
| Complementary | 1172 | 23 | 55 | 50 | 25 | 93.0% | 0.00008 |

- Introducing ATR alongside CEL further enhances the model's performance, with an average score of 63.6. The inclusion of ATR results in more refined emotion handling, as seen in the improvements in the emotion and expression attributes.

- The full model, with CEL, ATR, and ER, achieves the highest performance, with an average score of 67.0. This configuration benefits from the combined effects of all components, especially in emotion reasoning, where the model shows notable improvements across all attributes, particularly in the expression and emotion metrics.

### 5.4 Dataset Verification

As data scale increases beyond 50K samples, maintaining consistent data quality becomes challenging. Our data pipeline offers a systematic solution to this problem. To assess the quality of the generated expressions, we conducted a user study on a manually inspected subset of test samples to verify their alignment with the intended emotional semantics. Specifically, we compared Emo-CFG with CelebV-Text, the largest human-labeled video emotion dataset, across three key dimensions: precision, rationality, and complementarity. Preference rates for Emo-CFG across these dimensions reached 95%, 92%, and 93%, respectively, with statistically significant differences (Wilcoxon signed-rank test, p ¡ 0.01). This result demonstrates that Emo-CFG provides more precise and expressive emotional representations than existing benchmarks.

## 6 Conclusion

In this work, we introduced VidEmo, a family of video-based emotion foundation models designed to unify fine-grained facial attribute perception, expression analysis, and high-level emotion understanding. Our framework integrates curriculum emotion learning with a novel affective-tree reasoning paradigm, enabling interpretable and structured emotion inference. We further curated Emo-CFG, a large-scale, instruction-driven dataset with hierarchical annotations and rationale-grounded data, which serves as a foundamental data infrastructure for training and evaluation. Experimental results on the Emo-CFG benchmark demonstrate that VidEmo consistently outperforms existing open- and closed-source VideoLLMs across 15 tasks, setting up a new milestone in all the attribute perception, expression analysis, and emotion understanding tasks.

**Limitations.** While VidEmo exhibits strong generalization across diverse tasks, several limitations remain. First, like most existing VideoLLMs, VidEmo is susceptible to generating counterfactual content, which can lead to false narratives or emotionally inconsistent descriptions. Second, emotion understanding is inherently multimodal; integrating additional modalities such as audio or contextual cues could significantly enrich affective reasoning and we view VidEmo as a strong foundation for future work in this direction, enabling the exploration of richer, more holistic emotion understanding.

**Acknowledgements.** This work was supported by the National Natural Science Foundation of China (No. 623B2056), the Natural Science Foundation of Tianjin, China (No.24JCZXJC00040), the Fundamental Research Funds for the Central Universities, the Supercomputing Center of Nankai University (NKSC). We sincerely thank the reviewer team (KQSY, 96WN, ntby, and JqQW) for their invaluable feedback to improve our manuscript.

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

# A Implementation and Training Details

Following the approach outlined by Qwen2.5-VL [65], we adopt a Vision Transformer (ViT)-based architecture for the visual encoder and utilize an autoregressive model for the text encoder. For the foundational large language model (LLM), we select models within the 3B to 7B parameter range. VidEmo is pre-trained for 3 epochs with a batch size of 1024 and is subsequently post-trained for 1 epoch with a batch size of 128. We employ the AdamW [48] optimizer with a cosine learning rate schedule. The learning rate is set to 2e-5 for SFT stage and 1e-5 for RL stage, with a warmup rate of 0.03. As shown in Table 6, we follow the existing MLLM training setting and use a two-stage tuning paradigm.

- **Backbone**: We use the same backbone LLM and vision encoder as Qwen2.5-VL.
- **Data**: We train VidEmo by our constructed Emo-CFG and textual knowledge dataset MAGPIE.
- **Hyperparameter**: We follow the common setting and train the model by default learning rate, weight decay, and batch size.

Table 6: **Implementation details and hyperparameters for our VidEmo family.**

| Model | Backbone | | Data | | Hyperparameter | | | |
|-------|----------|--------|------|------|------|------|------|--------|
| | LLM | Vision | SFT | RL | lr | wd | bs | rollout |
| VidEmo-Base-3B | Qwen2.5-3B | ViT | Emo-CFG | - | 2e-5 | 0 | 1024 | - |
| VidEmo-Base-7B | Qwen2.5-7B | ViT | Emo-CFG | - | 2e-5 | 0 | 1024 | - |
| VidEmo-T1-7B | Qwen2.5-7B | ViT | Emo-CFG | Emo-CFG | 1e-5 | 0 | 128 | 8 |

# B Emo-CFG Dataset Details

## B.1 Dataset Construction

In this section, we provide an overview of the training data used in our Emo-CFG dataset, which is sourced from multiple datasets to address various tasks related to emotion and attribute perception. The data is collected to support model training, and the details of each task are summarized in Table 7. We illustrate the scenario, data source, task, data number, and ratio of our training data.

The attribute perception category includes tasks such as appearance recognition, action recognition, and human identity recognition. These tasks are sourced from CelebV-HQ, CelebV-Text, and MEAD, with multi-label question answering (QA) and caption generation tasks. The open attribute perception category involves tasks that focus on the recognition and analysis of open-ended attributes like eye, mouth, nose, shape, gender, and more. Important attributes such as age, gender, and accessories are covered in this category, and each task plays a significant role in identifying open-ended features that contribute to emotional understanding.

In the expression analysis category, we focus on tasks related to sentiment recognition (SR), emotion recognition (ER), affective cues detection, and complex scenario understanding. These tasks are sourced from datasets like MOSEI, CHSIMSv1, and CASME and aim to capture fine-grained emotional expressions and actions. The analysis of micro-expressions and emotion-related cues in this category contributes to the detailed recognition of emotional states.

Lastly, the emotional intelligence category covers important capacities related to video-text relevance, fluency, and emotional reasoning. These capacities focus on understanding the relationship between video and text for emotional intelligence applications. Notably, emotional reasoning tasks make up 60.90% of the dataset in this category, highlighting the importance of reasoning-based tasks in the overall dataset.

## B.2 Dataset Statistics

Our constructed Emo-CFG richs in fine-grained caption for high-level emotion understanding. We illustrate the caption distribution as shown in Figure 7. We displays the distribution of caption lengths across multiple sources, including CelebV-Text, MOSEI, RAVDESS, MELD, CelebV-HQ,

Table 7: The overview of our training data. All the data used for training are sampled from the training or validation split of the source datasets. QA: Question Answering. OPEN: Open-ended Question Answering. SR: Sentiment Recognition. ER: Emotion Recognition.

| Scenario | Data Source | Task | Number | Ratio |
|---|---|---|---|---|
| **Attribute Perception** | | | | |
| Appearance Recognition | CelebV-HQ | Multi-label QA | 32010 | 8.16% |
| Appearance Caption | CelebV-Text | Caption | 59879 | 15.26% |
| Action Recognition | CelebV-HQ | Multi-label QA | 32010 | 8.16% |
| Action Caption | CelebV-Text | Caption | 59797 | 15.24% |
| Human Identity | MEAD | QA | 19998 | 5.10% |
| Head Pose | MEAD | QA | 188590 | 48.10% |
| **Open Attribute Perception** | | | | |
| Eye | | QA & OPEN | 72197 | 5.94% |
| Eyebrow | | QA & OPEN | 71793 | 5.90% |
| Mouth | | QA & OPEN | 94722 | 7.79% |
| Nose | | QA & OPEN | 126248 | 10.38% |
| Hair | | QA & OPEN | 154994 | 12.75% |
| Chin | | QA & OPEN | 20559 | 1.69% |
| Shape | Emo-CFG | QA & OPEN | 78689 | 6.47% |
| Feature | | QA & OPEN | 92830 | 7.63% |
| Accessory | | QA & OPEN | 56835 | 4.67% |
| Age | | QA & OPEN | 145359 | 11.96% |
| Gender | | QA & OPEN | 151901 | 12.49% |
| Skin | | QA & OPEN | 104005 | 8.55% |
| Body Action | | QA & OPEN | 16577 | 1.36% |
| Head Action | | QA & OPEN | 2658 | 0.22% |
| Face Action | | QA & OPEN | 26515 | 2.18% |
| **Expression Analysis** | | | | |
| Single-label SR | MOSEI,MOSI | QA | 19710 | 5.66% |
| Fine-grained SR | CHSIMSv1 | QA | 1824 | 0.52% |
| Single-label ER | MAFW,DFEW,MER2023 | QA | 20060 | 5.76% |
| Multi-label ER | MAFW | Multi-label QA | 7178 | 2.06% |
| Fine-grained ER | MEAD,RAVDESS | QA | 198957 | 57.11% |
| Micro-expression Detection | CASME,CASME$^2$ | QA | 515 | 0.15% |
| Action Unit Detection | AffWild2 | QA | 2180 | 0.63% |
| Conversation Reasoning | PERR,MELD | QA | 38153 | 10.95% |
| Expression Caption | CelebV-Text | Caption | 59797 | 17.16% |
| **Emotional Intelligent** | | | | |
| Video-Text Relevance | | | | |
| Fluency | | | | |
| Coherency | | Fine-grained | | |
| Response Accuracy | Emo-CFG | Caption | 78072 | 39.10% |
| Cue Overlap | | | | |
| Label Overlap | | | | |
| Emotion Reasoning | | Rationale | 121618 | 60.90% |

MOSI, AFEW, DFEW, CHSIMS, MAFW, PERR, and FERV39K. Each histogram represents the frequency of captions of varying lengths, ranging from 0 to 500 words. We can observe that the variability in caption length across these datasets, with some datasets exhibiting a more uniform distribution (e.g., CelebV-Text and MOSI) and others showing skewed distributions (e.g., MAFW and PERR).

We further illustrate the rationale distribution for high-level emotion understanding as shown in Figure 8. The rationale lengths exhibit distinct distributions across the datasets. Some sources, like CelebV-Text and MOSI, show more uniform distributions, while others, such as MAFW and FERV39K, present skewed distributions. These distributions are crucial as they reflect the varying complexity and the level of detail involved in the rationales used for emotion analysis and understanding in multimodal tasks.

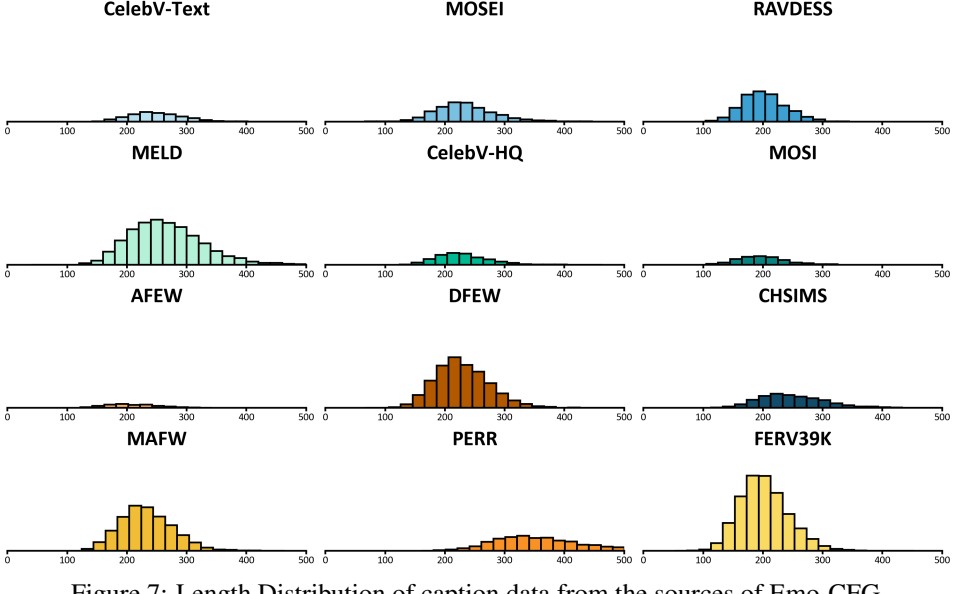

Figure 7: Length Distribution of caption data from the sources of Emo-CFG.

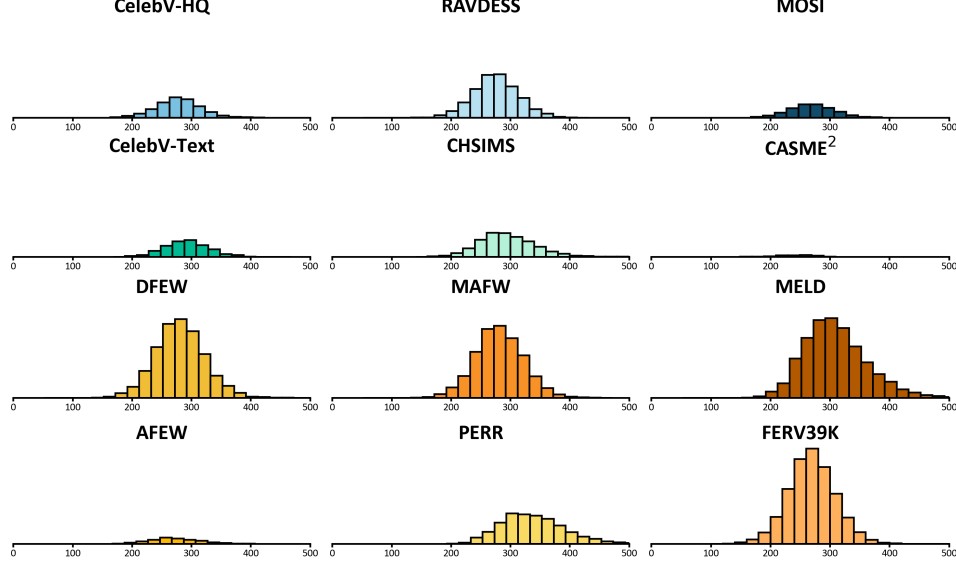

Figure 8: Length Distribution of rationale data from the sources of Emo-CFG.

## C    More Experimental Results

We further explore the design of the proposed components and analyze their effects in detail.

**Supervised Finetuning: Curriculum Emotion Learning.** The effectiveness of our proposed Curriculum Emotion Learning (CEL) approach is demonstrated through an in-depth analysis of the pre-training stages, as outlined in Table 4.

For the single-stage pre-training, where each stage is isolated, we observe varying levels of performance across different emotion-related tasks. Specifically, when the model is trained on attribute data alone, the average performance is lower due to the limited complexity of the tasks. Introducing expression data results in moderate improvements, particularly in expression-related tasks, but the overall performance remains relatively modest (42.2). The most substantial improvement occurs when emotion data is introduced, leading to a noticeable boost in emotion-related tasks (53.9), but still lacking the holistic integration seen in multi-stage training.

For the our proposed multiple-stage pre-training, where the model is exposed to progressively more complex tasks across all three stages, the performance improves significantly. The introduction of both attribute and expression data at earlier stages (52.6 average) enables the model to better integrate and align emotion-related information. With the complete multi-stage pre-training, which includes all three data types, the model achieves a robust performance (61.9 average), indicating that the curriculum learning strategy successfully enhances the model's understanding of emotional complexity. The analysis reveals that progressively increasing task difficulty facilitates the model's ability to learn emotion-based tasks more effectively, aligning with our goal of gradually injecting emotion knowledge into the base model.

Table 8: **In-Depth Analysis on curriculum emotion learning with data curation for our VidEmo.** Attr = Attribution Data, Exp = Expression Data, Emo = Emotion Data.

| CEL | | | Attribute | Expression | Emotion | Average |
|---|---|---|---|---|---|---|
| Attr | Exp | Emo | | | | |
| Single Stage | | | | | | |
| | | | 63.5 | 27.3 | 63.6 | 51.4 |
| ✓ | | | 65.7 | 24.1 | 32.4 | 40.7 |
| | ✓ | | 52.9 | 31.6 | 42.1 | 42.2 |
| | | ✓ | 64.3 | 30.2 | 67.3 | 53.9 |
| Multiple Stage | | | | | | |
| ✓ | ✓ | | 77.2 | 35.1 | 45.5 | 52.6 |
| ✓ | ✓ | ✓ | 79.5 | 38.7 | 67.5 | 61.9 |

**Post-training with Affective-Tree Reward.** The post-training phase with Affective-Tree Reward (ATR) is designed to enhance the model's performance in emotional reasoning tasks. Table 9 presents the results of integrating various components into the post-training process. Initially, when only the group relative policy optimization (GRPO) is applied, the model achieves an average score of 51.2. This score serves as the baseline performance before incorporating the Affective-Tree Reward.

Upon adding the Affective-Tree Reward (ATR), the model's performance increases to an average of 61.4, indicating that the inclusion of this reward mechanism improves the model's ability to generate emotion-related captions. The introduction of the Tree Edit Distance (ATR) further enhances the model's performance, resulting in an average score of 63.6 This improvement is observed across the attribute, expression, and emotion tasks.

These results demonstrate that the addition of the Affective-Tree Reward and Tree Edit Distance enhances the model's performance in emotional reasoning tasks by improving the accuracy of caption generation and ensuring structural alignment with human-annotated captions.

Table 9: **In-Depth Analysis on affective tree reward for our VidEmo.**

| | Attribute | Expression | Emotion | Average |
|---|---|---|---|---|
| GRPO | 75.7 | 33.8 | 44.2 | 51.2 |
| + Tree Reward | 80.9 | 38.1 | 65.3 | 61.4 |
| + Tree Edit Distance (ATR) | 81.3 | 40.1 | 69.3 | 63.6 |

**Emotion Reasoning.** The performance of the model in emotion reasoning tasks is evaluated by varying the number of candidate responses sampled during inference. As shown in Table 10, the baseline model, which uses the Affective-Tree Reward (ATR) with a single output (n=1), achieves an average score of 63.6 across the attribute, expression, and emotion tasks. This baseline represents the basic reasoning model, which is trained with ATR and generates only one response per query.

When the number of candidate outputs is increased, the model's performance improves. For instance, when two candidate responses (n=2) are sampled, the model achieves an average score of 65.1, with notable improvements in expression and emotion tasks. Further increasing the number of candidate outputs to four (n=4) results in a slight performance boost, bringing the average score to

66.2. This trend continues with the sampling of eight candidate responses (n=8), where the model achieves the highest average score of 67.0, along with consistent improvements across all tasks.

This analysis highlights the benefit of the search-based reasoning strategy in enhancing the model's performance. By sampling multiple candidate outputs and selecting the best one based on a reward-guided scoring mechanism, the model is able to refine its emotional reasoning process.

Table 10: **In-Depth Analysis on emotion reasoning for our VidEmo.** Baseline is the basic reasoning model trained with ATR and only outputs one response (n=1).

|  | Attribute | Expression | Emotion | Average |
|---|---|---|---|---|
| Baseline | 81.3 | 40.1 | 69.3 | 63.6 |
| +Emotion Reasoning (n=2) | 82.9 | 42.3 | 70.1 | 65.1 |
| +Emotion Reasoning (n=4) | 84.2 | 43.2 | 71.2 | 66.2 |
| +Emotion Reasoning (n=8) | 84.5 | 43.8 | 72.9 | 67.0 |

Table 11: The overview of our evaluation benchmark. All the data used for evaluation are sampled from the testing split of the source datasets. QA: Question Answering. OPEN: Open-ended Question Answering. SR: Sentiment Recognition. ER: Emotion Recognition.

| Scenario | Data Source | Task | Number | Metric |
|---|---|---|---|---|
| **Attribute Perception** | | | | |
| Appearance Recognition | CelebV-HQ | Multi-label QA | 500 | F1 |
| Appearance Caption | CelebV-Text | OPEN | 500 | GPT score |
| Action Recognition | CelebV-HQ | Multi-label QA | 500 | F1 |
| Action Caption | CelebV-Text | OPEN | 500 | GPT score |
| Human Identity | MEAD | QA | 500 | ACC |
| Head Pose | MEAD | QA | 500 | ACC |
| **Open Attribute Perception** | | | | |
| Eye | | QA & OPEN | 300 | ACC & GPT score |
| Eyebrow | | QA & OPEN | 300 | ACC & GPT score |
| Mouth | | QA & OPEN | 300 | ACC & GPT score |
| Nose | | QA & OPEN | 300 | ACC & GPT score |
| Hair | | QA & OPEN | 300 | ACC & GPT score |
| Chin | | QA & OPEN | 300 | ACC & GPT score |
| Shape | | QA & OPEN | 300 | ACC & GPT score |
| Feature | Emo-CFG | QA & OPEN | 300 | ACC & GPT score |
| Accessory | | QA & OPEN | 300 | ACC & GPT score |
| Age | | QA & OPEN | 300 | ACC & GPT score |
| Gender | | QA & OPEN | 300 | ACC & GPT score |
| Skin | | QA & OPEN | 300 | ACC & GPT score |
| Body Action | | QA & OPEN | 107 | ACC & GPT score |
| Head Action | | QA & OPEN | 4 | ACC & GPT score |
| Face Action | | QA & OPEN | 189 | ACC & GPT score |
| **Expression Analysis** | | | | |
| Single-label SR | MOSEI,MOSI | QA | 300 | ACC |
| Fine-grained SR | CHSIMSv1 | QA | 300 | ACC |
| Single-label ER | MAFW,DFEW,MER2023 | QA | 300 | ACC |
| Multi-label ER | MAFW | Multi-label QA | 300 | F1 |
| Fine-grained ER | MEAD,RAVDESS | QA | 300 | ACC |
| Micro-expression Detection | CASME II | QA | 246 | ACC |
| Action Unit Detection | AffWild2 | Multi-label QA | 87 | F1 |
| Conversation Reasoning | PERR,MELD | QA | 300 | ACC |
| Emotion Caption | CelebV-Text | OPEN | 300 | GPT score |
| **Emotional Intelligent** | | | | |
| Video-Text Relevance | | | | GPT score |
| Fluency | | | | GPT score |
| Coherency | Emo-CFG | Fine-grained | 2600 | GPT score |
| Response Accuracy | | Caption & Rationale | | GPT score |
| Cue Overlap | | | | GPT score |
| Label Overlap | | | | GPT score |

# D   Evaluation Settings

## D.1   Task & Source

We outline the overview of our evaluation benchmark as shown in Table 11, including scenario, data source, task, data number, ratio and evaluation metrics.

**Attribute Perception:**  CelebV-HQ [103] for appearance recognition and action recognition. CelebV-text [82] for appearance caption and action caption. MEAD [69] for head pose estimation and human identity verification.

**Expression Analysis:**  MOSEI [85] and MOSI [84] for single-label sentiment recognition. CH-SIMSv1 [83] for fine-grained sentiment recognition. MAFW [44], DFEW [21] and MER2023 [36] for single-label emotion recognition. MAFW [44] for multi-label emotion recognition. MEAD [69] and REAVDESS [47] for fine-grained emotion recognition. CASME [79], CASME II [78] and

CASME$^2$ [56] for micro-expression detection. Aff-Wild [22] for AU detection. MELD [54] and PERR [16] for Conversation Reasoning. CelebV-text [82] for emotion caption.

**Emotion Understanding:** The annotation for open attribute perception and fine-grained caption in Emo-CFG from 17 source datasets: AFEW [12], CAER [24], CASME$^2$ [56], CelebV-HQ [103], CelebV-text [82], CHSIMSv1 [83], CHSIMSv2 [43], DFEW [21], FERV39K [71], L-SVD [53], MAFW [44], MELD [54], MER2023 [36], MOSEI [85], MOSI [84], PERR [16], REAVDESS [47], Aff-Wild [22], MEAD [69].

## D.2    Competitive Alternatives

- **Gemini 2.0** Gemini 2.0 is a cutting-edge closed-source multimodal large language model developed by Google DeepMind. It is designed to handle both textual and visual inputs, excelling in tasks such as video understanding, summarization, and generation.

- **Claude-3.5-Sonnet** Anthropic's Claude-3.5-Sonnet is a closed-source VideoLLM that builds on the Claude series with enhanced capabilities in video comprehension and interaction.

- **GPT-4o / 4o mini** GPT-4o and its lightweight variant, GPT-4o mini, are closed-source VideoLLMs developed by OpenAI. These models are optimized for visual understanding tasks, offering a balance between computational efficiency and performance.

- **Qwen-VL-Max** Qwen-VL-Max is designed to process complex video content in conjunction with textual inputs, making it a versatile tool for video summarization, captioning, and question-answering tasks.

- **Qwen2.5-VL** Qwen2.5-VL is an advanced open-source vision-language model that excels in multimodal tasks such as object localization, and long-video comprehension. Its innovative architecture enables efficient visual recognition and interaction with extended temporal video data.

- **InternVL2.5** InternVL2.5 pushes the boundaries of open-source multimodal models by introducing advanced scaling strategies across model architecture, diverse video-text datasets, and test-time optimization.

- **mPLUG-Owl3** mPLUG-Owl3 is a cutting-edge multi-modal large language model designed to excel in long image-sequence understanding, including tasks involving lengthy videos and interleaved image-text scenarios.

- **VideoLLaMA3** VideoLLaMA3 adopts a vision-centric training paradigm, emphasizing the use of high-quality image-text data to improve multimodal capabilities.

- **LLaVA-OV** By leveraging insights from data, models, and visual representations, LLaVA-OV achieves significant performance improvements in three major computer vision tasks while enabling strong transfer learning across modalities.

- **ShareGPT4Video** ShareGPT4Video introduces a framework for video understanding and generation by leveraging high-quality captions generated through a specific summary prompt.

- **LLaVA-NeXT-Video** LLaVA-NeXT-Video is an advanced open-sourced large multimodal model designed for comprehensive video understanding, leveraging interleaved data formats to enhance performance across multi-image, video, and 3D tasks.

- **LLaVA-Video** LLaVA-Video is a video understanding model that processes video sequences using a straightforward approach, supporting both fps and uniform frame sampling. It is modular and scalable, allowing for efficient training and inference with limited resources, and achieves performance comparable to some 7B models on multiple benchmarks.

Table 12 provides model cards for different MLLMs, including reference papers, parameter scale, and links to pre-trained weights.

## D.3    Evaluation Metrics

We evaluate the performance of the model using various metrics (GPT score, Accurary, and F1 score), depending on the task. For text-oriented tasks such as Appearance Caption, Action Caption, Emotion Caption, Fine-grained Caption, and Open-ended QA, we employ GPT scores, using GPT-4o to score the predictions based on labels and model responses. For recognition tasks that

Table 12: Model cards for Open-sourced MLLMs.

| Model | Scale | Link |
|---|---|---|
| LLaVA-OV [25] | 1B | https://huggingface.co/llava-hf/llava-onevision-qwen2-0.5b-ov-hf |
| InternVL2.5 [8] | 2B | https://huggingface.co/OpenGVLab/InternVL2_5-2B |
| VideoLLaMA3 [86] | 2B | https://huggingface.co/DAMO-NLP-SG/VideoLLaMA3-2B |
| mPLUG-Owl3 [81] | 2B | https://huggingface.co/mPLUG/mPLUG-Owl3-2B-241014 |
| Qwen2.5-VL [65] | 3B | https://huggingface.co/Qwen/Qwen2.5-VL-3B-Instruct |
| ShareGPT4Video [7] | 8B | https://huggingface.co/Lin-Chen/sharegpt4video-8b |
| InternVL2.5 [8] | 8B | https://huggingface.co/OpenGVLab/InternVL2_5-8B |
| LLaVA-NeXT-Video [41] | 7B | https://huggingface.co/llava-hf/LLaVA-NeXT-Video-7B-hf |
| LLaVA-OV [25] | 7B | https://huggingface.co/llava-hf/llava-onevision-qwen2-7b-ov-hf |
| VideoLLaMA3 [86] | 7B | https://huggingface.co/DAMO-NLP-SG/VideoLLaMA3-7B |
| LLaVA-Video [91] | 7B | https://huggingface.co/lmms-lab/LLaVA-NeXT-Video-7B |
| mPLUG-Owl3 [81] | 7B | https://huggingface.co/mPLUG/mPLUG-Owl3-7B-240728 |
| Qwen2.5-VL [65] | 7B | https://huggingface.co/Qwen/Qwen2.5-VL-7B-Instruct |

involve classification, such as Human Identity, Head Pose, choice-based QA, Single-label Sentiment Recognition (SR), Fine-grained SR, Single-label Emotion Recognition (ER), Fine-grained ER, Micro-expression Detection, and Conversation Reasoning, we use accuracy as the primary evaluation metric. Prior to calculating accuracy, GPT-4o is used to convert the model's responses into standardized labels to ensure consistency in the evaluation. For multi-label tasks, including Appearance Recognition, Action Recognition, Multi-label ER, and Action Unit Detection, we utilize the F1-score to evaluate the model's performance, capturing both precision and recall in these multi-label settings.

### D.4 Prompts used for Evaluation

We utilize three distinct prompts during the evaluation phase, each designed for specific aspects of model performance: F1 score, accuracy evaluation, and GPT score computation.

The f1 score prompt, shown in Figure 9, is used to convert the model's response into multiple choice options. The prompt requires the model to extract relevant labels directly from the provided answer, ensuring that only valid and directly relevant labels are included. If the answer is invalid or none, the prompt instructs the model to output "none." The answer options are then concatenated together, and the model is instructed not to include any additional phrases such as "output:" or "Here is the output."

The accuracy evaluation prompt, depicted in Figure 10, is designed to assess the correctness of the model's response. It presents the model with a question, the ground truth answer, and the model's own response. The model is tasked with judging whether the response matches the ground truth and is required to output a simple "yes" or "no" depending on whether the model's response is correct.

The GPT score evaluation prompt, illustrated in Figure 11, is employed to compute a numerical score reflecting the accuracy and the degree of match between the model's response and the ground truth. The score is provided as a number between 0 and 100, with no additional commentary or text. This prompt allows for a more granular evaluation of the model's performance, especially in terms of accuracy and relevance.

These prompts are critical for evaluating different facets of model performance, ensuring that the model's responses are both accurate and relevant.

**Prompt for Label Extraction**

Based on the question, please convert the provided answer into multiple choices connected by commas. output none if the answer is none or invalid. Choice candidates from: {Choices from Question}.

Question: {Question}
Model's Response: {Model's Response}

[Requirement]
1. Directly output the converted phrases from the answer. Only output the phrases.
2. Don't say like "Here is the output" or 'output:'.
3. Each option is directly concatenated together in the output.

[Output Example]
mustache,receding_hairline

Figure 9: Prompt for extracting multiple phases and compute f1 score from the response of models.

**Prompt for Acc Evaluation**

You are an expert in evaluating video facial expression analysis.

Question: {Question}
Ground Truth:{The ground true answer}
Model's Response: {Model's Response}

Please judge the correctness of model's response according to the given answer and question.

[Requirement]
1. Directly output the judgement: yes or no

[Example]
Yes

Figure 10: Prompt for computing accuracy score.

**Prompt for GPT Score Evaluation**

You are an expert in evaluating the accuracy of video facial actions.

Question: {Question}
Ground Truth:{The ground true answer}
Model's Response: {Model's Response}

Please score the model's prediction according to the correctness and matching degree.

[Requirement]
1. Directly output the score number from 0 to 100.
2. No sentence or word.
[Example]
80

Figure 11: Prompt for computing GPT score.

# E  Full Related Work

## E.1  Facial Video Analysis

Face video analysis is a long standing problem towards high-level human understanding which involves various tasks, including attribute perception [102, 5], expression analysis [4, 61], and emotion understanding [92, 70]. Advanced models can be divided into those for attribution perception tasks [97, 30] and for high-level emotion understanding tasks [92, 70, 4, 36]. Various face perception models leverages strong backbone power for constructing multi-task framework [59]. Pioneering methods are developed to solve pre-defined tasks, while MLLM-based method are proposed to enhance their zero-shot capacity. Going forward to high-level emotion understanding [95, 98, 38], recent methods embrace MLLM for their strong zero-shot perception capacity [59, 88, 33, 14]. EMO [75] firstly incorporates facial priors, including facial embeddings, landmarks, and age-gender-race attributes, through a mlp-based connnector to improve emotion understanding. EmotionLLaMA [10] introduces an emotion dataset including 28K coarse-grained and 4K fine-grained annotated datasets. OmniEmotion [80] proposes to explicitly integrate facial and audio modeling for emotion recognition. FacialDynamic [96] construct a existing largest human-labelled dataset with 5K samples. ExpLLM [23] recently explore to leverage chain-of-thought strategy to empower LLM with the reasoning capability. However, existing approaches are often constrained to a limited set of emotion categories or rely on static attribution perception. To advance cognitive human emotion understanding, we propose a fine-grained emotion-centric model empowered by dynamic attribution perception and emotion reasoning.

## E.2  Video Extension in MLLM

VideoLLMs [40, 50, 91, 32, 49] have gained significant attention in recent years by leveraging existing pre-trained foundational models, particularly powerful Large Language Models (LLMs), to enhance support for video inputs and outputs [11, 19, 104, 77]. The key components of VideoLLMs include: 1) a video encoder responsible for encoding inputs from different modalities into feature representations that the model can understand, *e.g.*, ViT [13], CLIP [57]; 2) an input projector to align encoded spatiotemporal features from video with textual feature space in LLM. Input projectors can be implemented by linear projection or compressed-based projection such as Q-Former [31] or P-Former [20]; and 3) LLM Backbone based on pre-trained models like GPT [1] or LLaMA [67], which processes representations from different modalities and performs semantic understanding.

## E.3  Reasoning Model in MLLM

With the blossom of a series of recent models such as DeepSeek-R1 and OpenAI o-series [52, 18], various works probe into integrating MLLMs with reasoning capacity [2]. Multimodal chain-of-thought (MCoT) prompting [27, 66] offers a step-by-step reasoning trajectory when MLLM faces hard questions including detail grounding [72], agent planing [27], etc. Specifically, MCoT decomposes the question into a group of reasoning steps and builds a chain to guide the model to generate the results of complex problems step-by-step [73, 64, 89]. Recent works have demonstrated that CoT prompting substantially improves the MLLM's capability on reasoning tasks. For instance, LLaVA-CoT [76] prompts MLLMs reasoning steps into the summary, caption, reasoning, and conclusion stages and proposes a stage-level beam search strategy to further enhance reasoning capacity. LLaVA-Reasoner [89] pioneers the use of forced Chain-of-Thoughts, establishing a new direction for structured prompting techniques. In this paper, we propose affective cues-based rationale tree as intermediate bridge to meet the gap between abstract emotion and basic attribute.

# F   Visualization

In this section, we present visualization samples generated by our VidEmo model, as well as those from our constructed Emo-CFG dataset. For more comprehensive visualization results, please refer to the video demos provided in the attached zip file.

## F.1   Results from our VidEmo model

**Attribute Perception.** We show the visualization comparison between our VidEmoand cutting-edge milestone Gemini 2.0 for appearance recognition and appearance caption in Figure 12, action recognition and action caption in Figure 13, head pose recognition and human identity recognition in Figure 14, open attribute perception in Figure 15.

**Expression Analysis.** We show the visualization comparison between our VidEmoand cutting-edge milestone Gemini 2.0 for single-label emotion recognition, fine-grained emotion recognition and multi-label emotion recognition in Figure 16, single-label sentiment recognition and fine-grained sentiment recognition in Figure 17, micro-expression detection and action unit detection in Figure 18, conversation reasoning and emotion caption in Figure 19.

**Emotion Understanding.** We show the visualization comparison between our VidEmoand cutting-edge milestone Gemini 2.0 for fine-grained video caption in Figure 20

## F.2   Samples from our Emo-CFG Dataset

**Attribute Perception.** As shown in Figure 21, we provide samples from public face attribute datasets, and we convert their annotations into question-answer pairs. In addition, we visualized the labeled face attribute samples in Figure 22.

**Expression Analysis.** As shown in Fig 23 and Fig 24, we provide samples from public emotion recognition datasets and our fine-grained caption dataset.

**Emotion Understanding.** As shown in Fig 25, we provide samples for emotion reasoning from the provided dataset.

**Meta Labels of Face Landmarks and Parsing Masks.** As shown in Fig 26, we provide meta labels of face boxes, face landmarks and parsing masks.

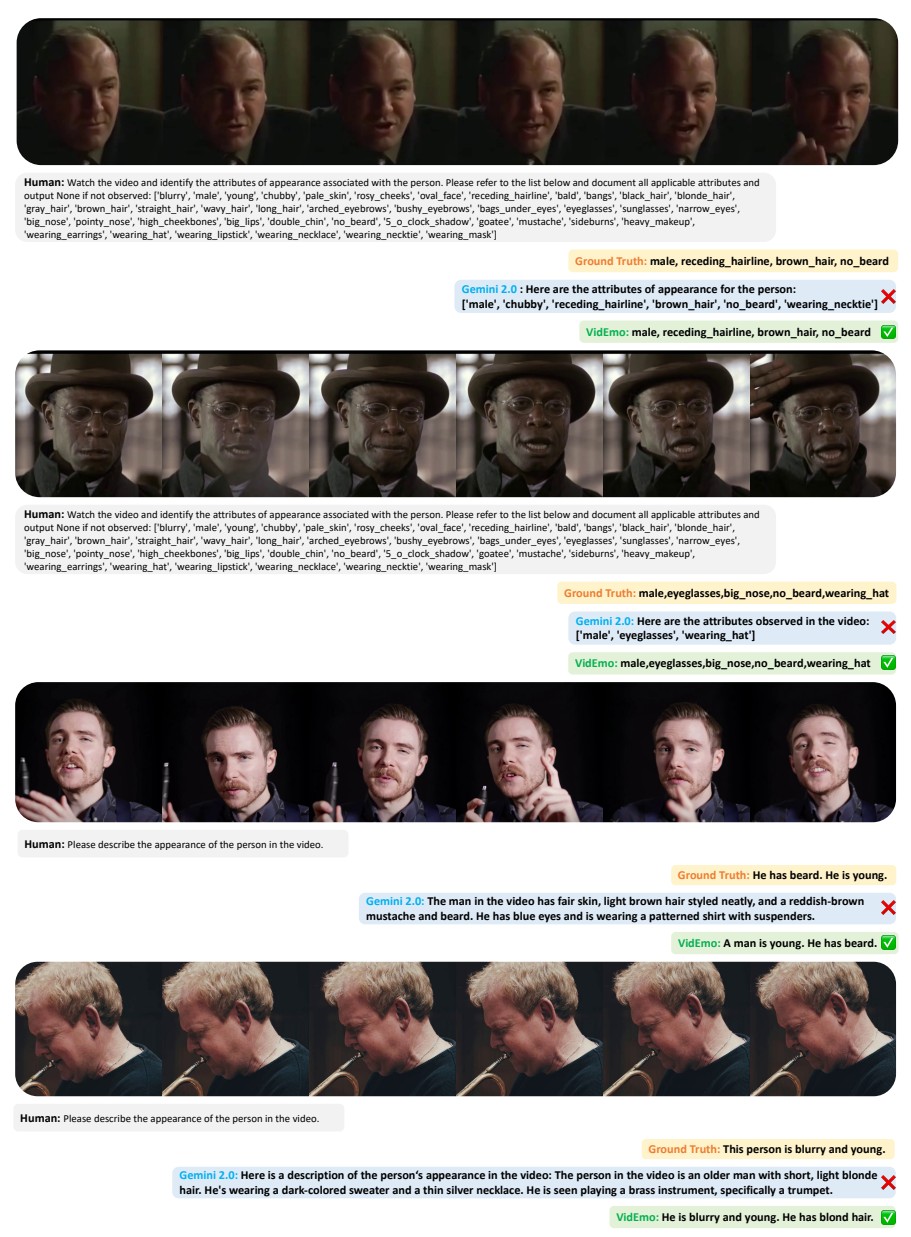

Figure 12: Visualization comparison results for appearance recognition and appearance caption.

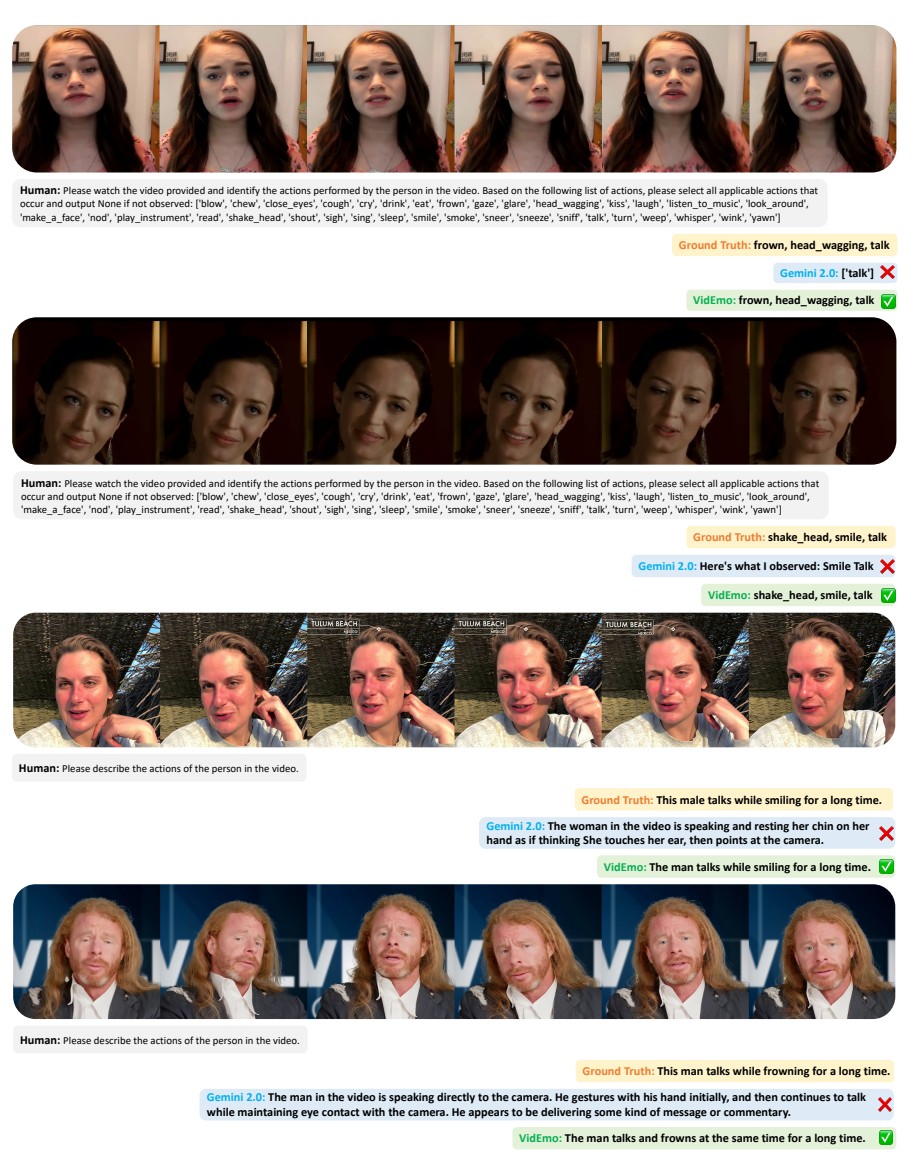

Figure 13: Visualization comparison results for action recognition and action caption.

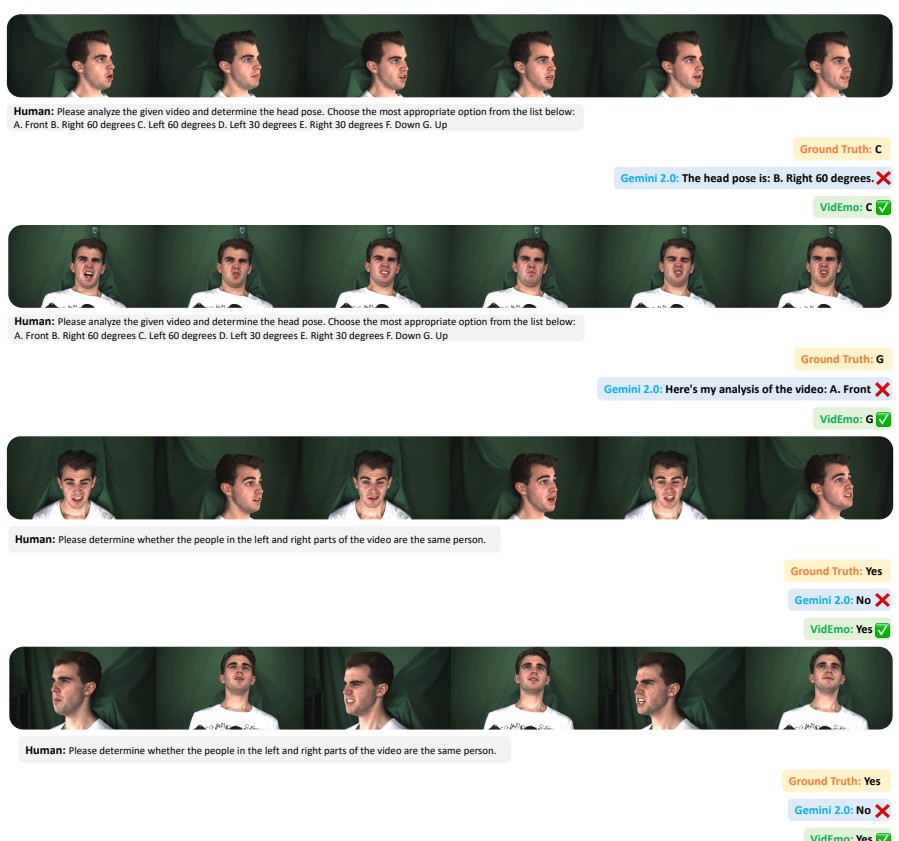

Figure 14: Visualization comparison results for head pose estimation and identity verification.

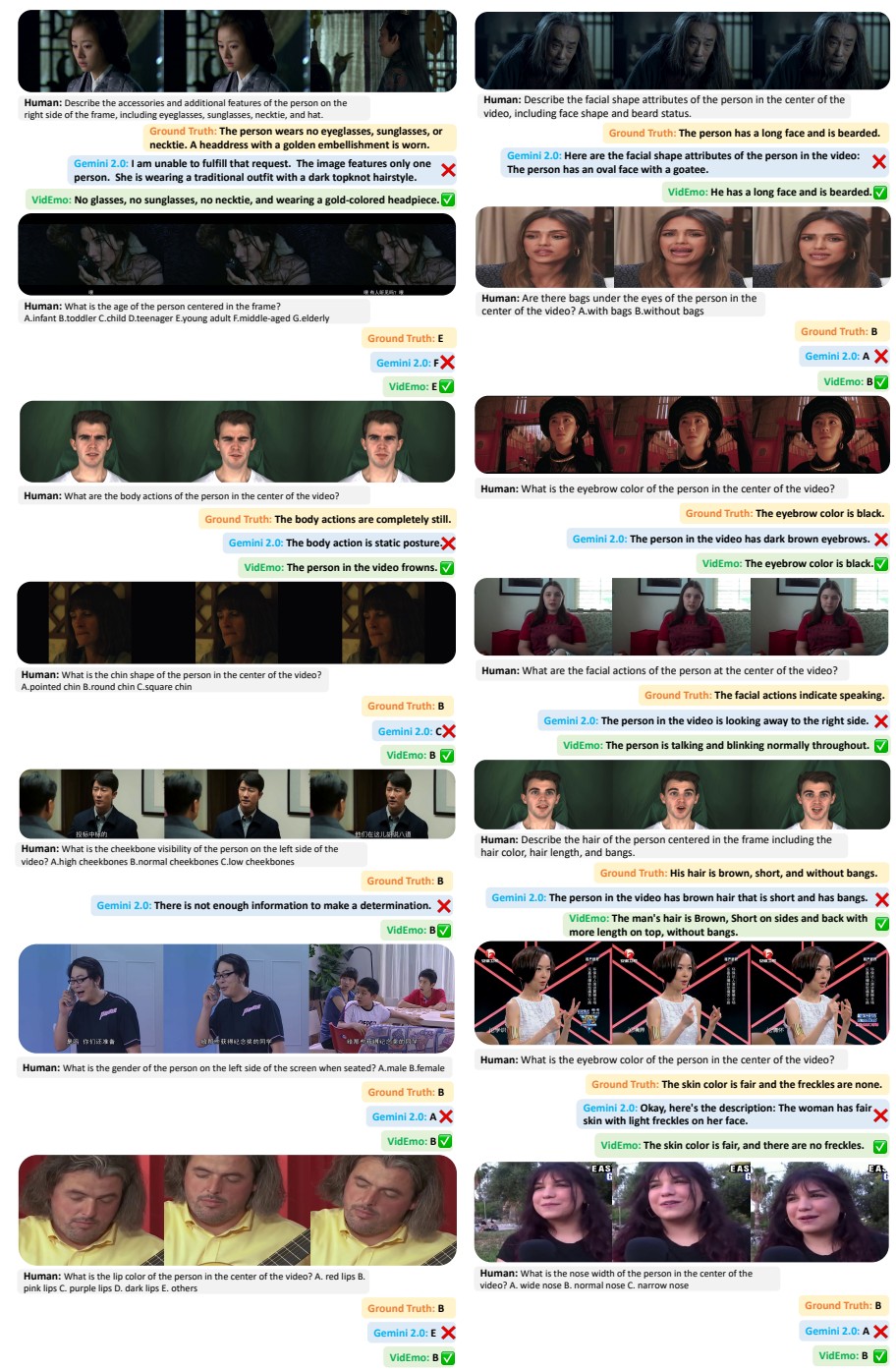

Figure 15: Visualization comparison results for open attribute perception.

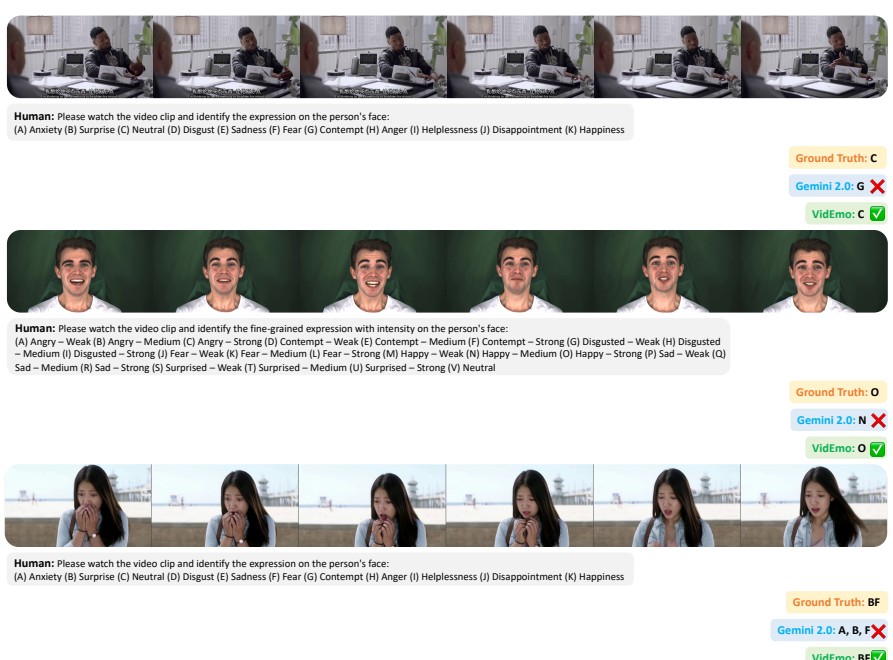

Figure 16: Visualization comparison results for single-label emotion recognition, multi-label emotion recognition and fine-grained emotion recognition.

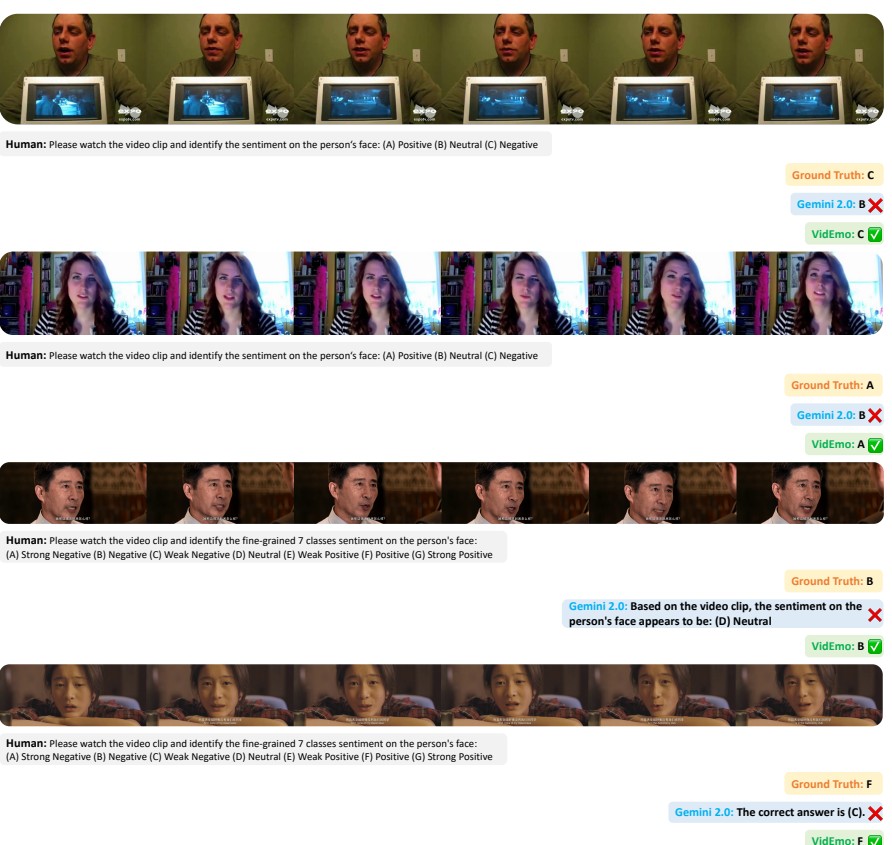

Figure 17: Visualization comparison results for micro-expression detection and action unit detection.

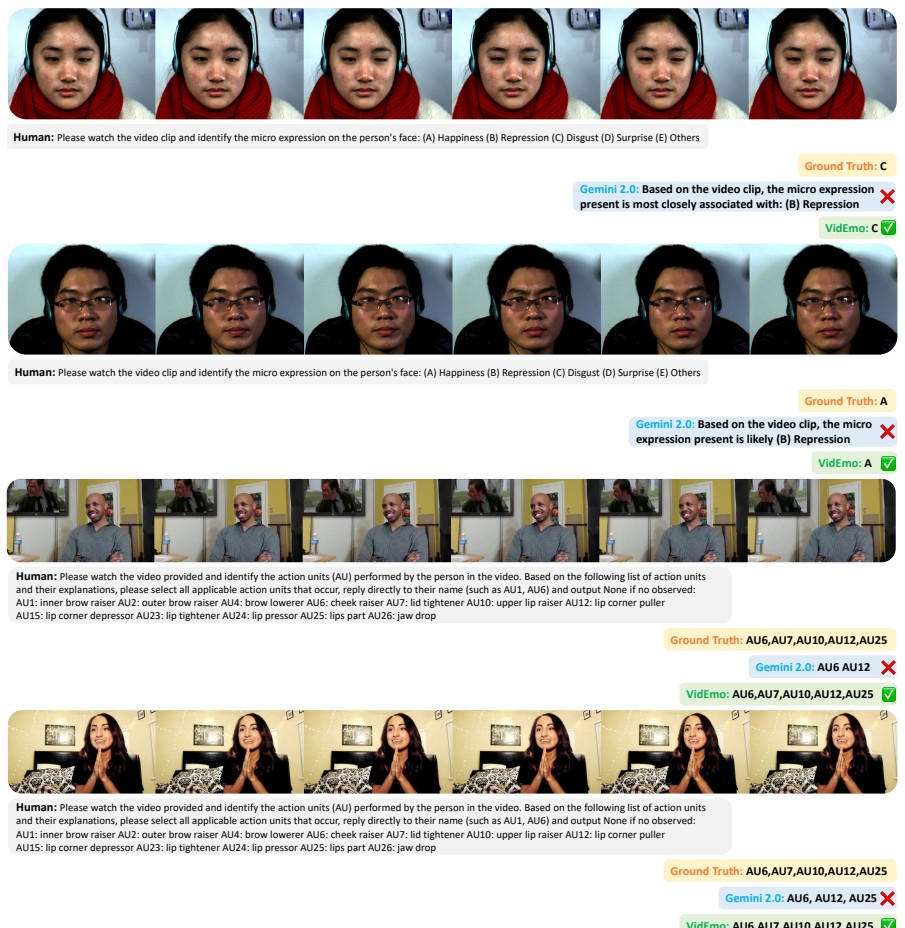

Figure 18: Visualization comparison results for single-label sentiment recognition and fine-grained sentiment recognition.

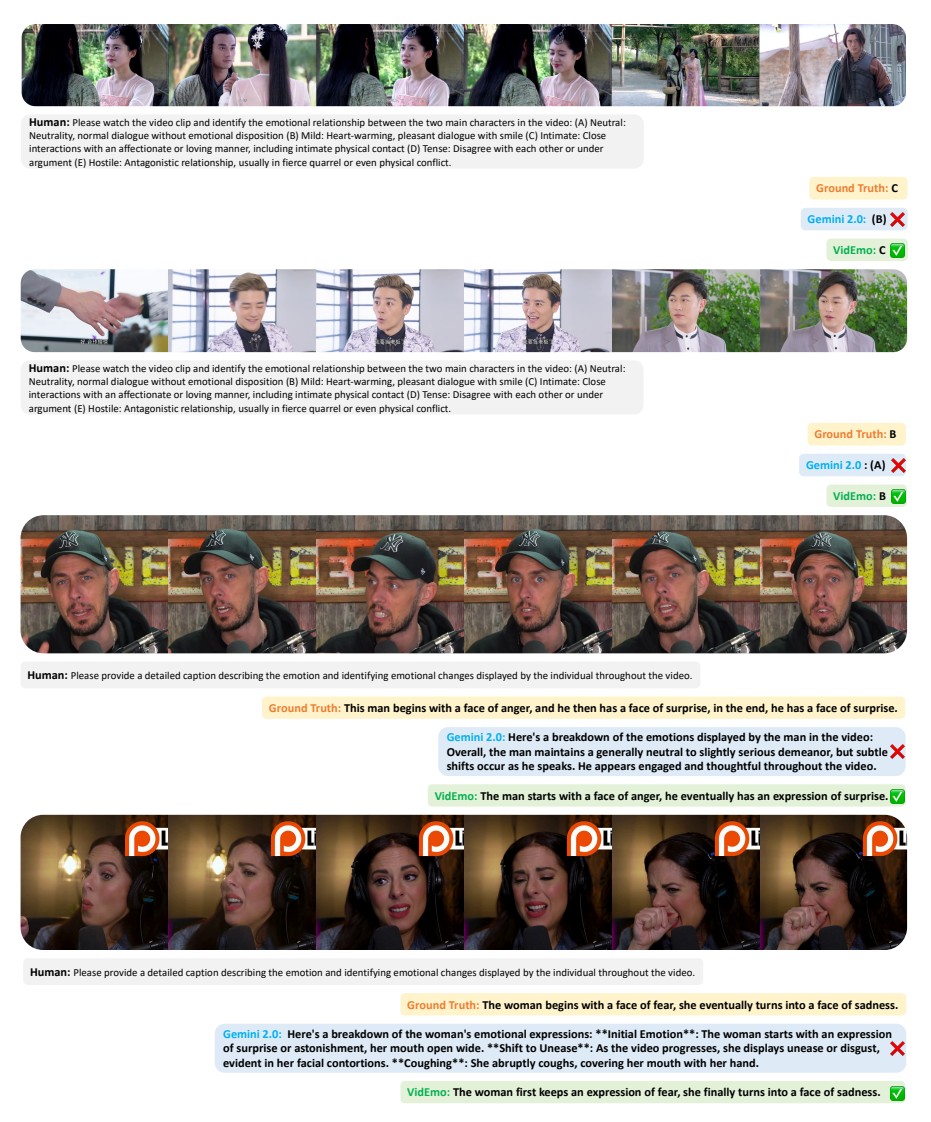

Figure 19: Visualization comparison results for conversation reasoning and emotion caption.

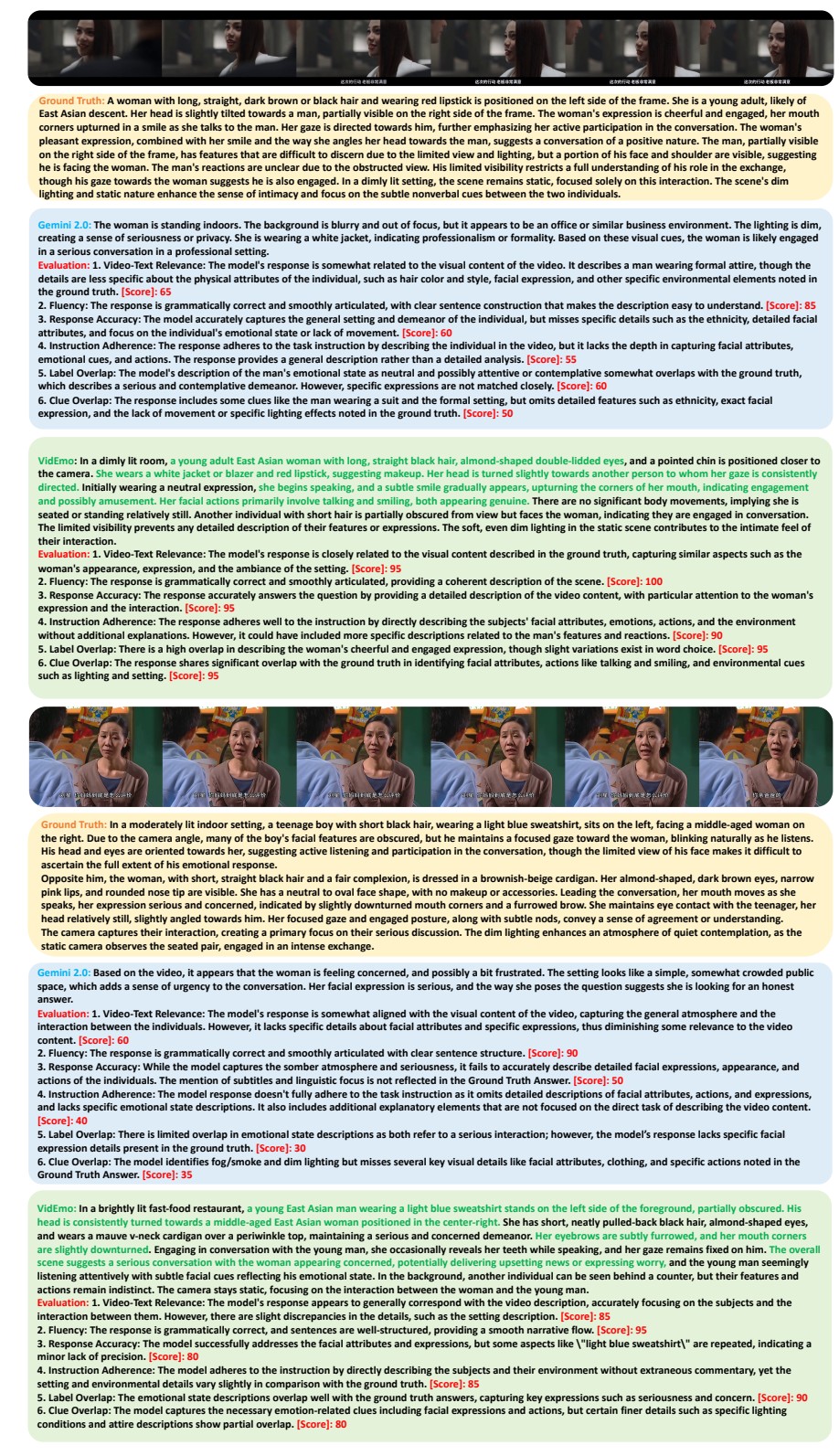

**Ground Truth:** A woman with long, straight, dark brown or black hair and wearing red lipstick is positioned on the left side of the frame. She is a young adult, likely of East Asian descent. Her head is slightly tilted towards a man, partially visible on the right side of the frame. The woman's expression is cheerful and engaged, her mouth corners upturned in a smile as she talks to the man. Her gaze is directed towards him, further emphasizing her active participation in the conversation. The woman's pleasant expression, combined with her smile and the way she angles her head towards the man, suggests a conversation of a positive nature. The man, partially visible on the right side of the frame, has features that are difficult to discern due to the limited view and lighting, but a portion of his face and shoulder are visible, suggesting he is facing the woman. The man's reactions are unclear due to the obstructed view. His limited visibility restricts a full understanding of his role in the exchange, though his gaze towards the woman suggests he is also engaged. In a dimly lit setting, the scene remains static, focused solely on this interaction. The scene's dim lighting and static nature enhance the sense of intimacy and focus on the subtle nonverbal cues between the two individuals.

**Gemini 2.0:** The woman is standing indoors. The background is blurry and out of focus, but it appears to be an office or similar business environment. The lighting is dim, creating a sense of seriousness or privacy. She is wearing a white jacket, indicating professionalism or formality. Based on these visual cues, the woman is likely engaged in a serious conversation in a professional setting.
**Evaluation: 1. Video-Text Relevance:** The model's response is somewhat related to the visual content of the video. It describes a man wearing formal attire, though the details are less specific about the physical attributes of the individual, such as hair color and style, facial expression, and other specific environmental elements noted in the ground truth. **[Score]: 65**
**2. Fluency:** The response is grammatically correct and smoothly articulated, with clear sentence construction that makes the description easy to understand. **[Score]: 85**
**3. Response Accuracy:** The model accurately captures the general setting and demeanor of the individual, but misses specific details such as the ethnicity, detailed facial attributes, and focus on the individual's emotional state or lack of movement. **[Score]: 60**
**4. Instruction Adherence:** The response adheres to the task instruction by describing the individual in the video, but it lacks the depth in capturing facial attributes, emotional cues, and actions. The response provides a general description rather than a detailed analysis. **[Score]: 55**
**5. Label Overlap:** The model's description of the man's emotional state as neutral and possibly attentive or contemplative somewhat overlaps with the ground truth, which describes a serious and contemplative demeanor. However, specific expressions are not matched closely. **[Score]: 60**
**6. Clue Overlap:** The response includes some clues like the man wearing a suit and the formal setting, but omits detailed features such as ethnicity, exact facial expression, and the lack of movement or specific lighting effects noted in the ground truth. **[Score]: 50**

**VidEmo:** In a dimly lit room, a young adult East Asian woman with long, straight black hair, almond-shaped double-lidded eyes, and a pointed chin is positioned closer to the camera. She wears a white jacket or blazer and red lipstick, suggesting makeup. Her head is turned slightly towards another person to whom her gaze is consistently directed. Initially wearing a neutral expression, she begins speaking, and a subtle smile gradually appears, upturning the corners of her mouth, indicating engagement and possibly amusement. Her facial actions primarily involve talking and smiling, both appearing genuine. There are no significant body movements, implying she is seated or standing relatively still. Another individual with short hair is partially obscured from view but faces the woman, indicating they are engaged in conversation. The limited visibility prevents any detailed description of their features or expressions. The soft, even dim lighting in the static scene contributes to the intimate feel of their interaction.
**Evaluation: 1. Video-Text Relevance:** The model's response is closely related to the visual content described in the ground truth, capturing similar aspects such as the woman's appearance, expression, and the ambiance of the setting. **[Score]: 95**
**2. Fluency:** The response is grammatically correct and smoothly articulated, providing a coherent description of the scene. **[Score]: 100**
**3. Response Accuracy:** The response accurately answers the question by providing a detailed description of the video content, with particular attention to the woman's expression and the interaction. **[Score]: 95**
**4. Instruction Adherence:** The response adheres well to the instruction by directly describing the subjects' facial attributes, emotions, actions, and the environment without additional explanations. However, it could have included more specific descriptions related to the man's features and reactions. **[Score]: 90**
**5. Label Overlap:** There is a high overlap in describing the woman's cheerful and engaged expression, though slight variations exist in word choice. **[Score]: 95**
**6. Clue Overlap:** The response shares significant overlap with the ground truth in identifying facial attributes, actions like talking and smiling, and environmental cues such as lighting and setting. **[Score]: 95**

**Ground Truth:** In a moderately lit indoor setting, a teenage boy with short black hair, wearing a light blue sweatshirt, sits on the left, facing a middle-aged woman on the right. Due to the camera angle, many of the boy's facial features are obscured, but he maintains a focused gaze toward the woman, blinking naturally as he listens. His head and eyes are oriented towards her, suggesting active listening and participation in the conversation, though the limited view of his face makes it difficult to ascertain the full extent of his emotional response.
Opposite him, the woman, with short, straight black hair and a fair complexion, is dressed in a brownish-beige cardigan. Her almond-shaped, dark brown eyes, narrow pink lips, and rounded nose tip are visible. She has a neutral to oval face shape, with no makeup or accessories. Leading the conversation, her mouth moves as she speaks, her expression serious and concerned, indicated by slightly downturned mouth corners and a furrowed brow. She maintains eye contact with the teenager, her head relatively still, slightly angled towards him. Her focused gaze and engaged posture, along with subtle nods, convey a sense of agreement or understanding. The camera captures their interaction, creating a primary focus on their serious discussion. The dim lighting enhances an atmosphere of quiet contemplation, as the static camera observes the seated pair, engaged in an intense exchange.

**Gemini 2.0:** Based on the video, it appears that the woman is feeling concerned, and possibly a bit frustrated. The setting looks like a simple, somewhat crowded public space, which adds a sense of urgency to the conversation. Her facial expression is serious, and the way she poses the question suggests she is looking for an honest answer.
**Evaluation: 1. Video-Text Relevance:** The model's response is somewhat aligned with the visual content of the video, capturing the general atmosphere and the interaction between the individuals. However, it lacks specific details about facial attributes and specific expressions, thus diminishing some relevance to the video content. **[Score]: 60**
**2. Fluency:** The response is grammatically correct and smoothly articulated with clear sentence structure. **[Score]: 90**
**3. Response Accuracy:** While the model captures the somber atmosphere and seriousness, it fails to accurately describe detailed facial expressions, appearance, and actions of the individuals. The mention of subtitles and linguistic focus is not reflected in the Ground Truth Answer. **[Score]: 50**
**4. Instruction Adherence:** The model response doesn't fully adhere to the task instruction as it omits detailed descriptions of facial attributes, actions, and expressions, and lacks specific emotional state descriptions. It also includes additional explanatory elements that are not focused on the direct task of describing the video content. **[Score]: 40**
**5. Label Overlap:** There is limited overlap in emotional state descriptions as both refer to a serious interaction; however, the model's response lacks specific facial expression details present in the ground truth. **[Score]: 30**
**6. Clue Overlap:** The model identifies fog/smoke and dim lighting but misses several key visual details like facial attributes, clothing, and specific actions noted in the Ground Truth Answer. **[Score]: 35**

**VidEmo:** In a brightly lit fast-food restaurant, a young East Asian man wearing a light blue sweatshirt stands on the left side of the foreground, partially obscured. His head is consistently turned towards a middle-aged East Asian woman positioned in the center-right. She has short, neatly pulled-back black hair, almond-shaped eyes, and wears a mauve v-neck cardigan over a periwinkle top, maintaining a serious and concerned demeanor. Her eyebrows are subtly furrowed, and her mouth corners are slightly downturned. Engaging in conversation with the young man, she occasionally reveals her teeth while speaking, and her gaze remains fixed on him. The overall scene suggests a serious conversation with the woman appearing concerned, potentially delivering upsetting news or expressing worry, and the young man seemingly listening attentively with subtle facial cues reflecting his emotional state. In the background, another individual can be seen behind a counter, but their features and actions remain indistinct. The camera stays static, focusing on the interaction between the woman and the young man.
**Evaluation: 1. Video-Text Relevance:** The model's response appears to generally correspond with the video description, accurately focusing on the subjects and the interaction between them. However, there are slight discrepancies in the details, such as the setting description. **[Score]: 85**
**2. Fluency:** The response is grammatically correct, and sentences are well-structured, providing a smooth narrative flow. **[Score]: 95**
**3. Response Accuracy:** The model successfully addresses the facial attributes and expressions, but some aspects like \"light blue sweatshirt\" are repeated, indicating a minor lack of precision. **[Score]: 80**
**4. Instruction Adherence:** The model adheres to the instruction by directly describing the subjects and their environment without extraneous commentary, yet setting and environmental details vary slightly in comparison with the ground truth. **[Score]: 85**
**5. Label Overlap:** The emotional state descriptions overlap well with the ground truth answers, capturing key expressions such as seriousness and concern. **[Score]: 90**
**6. Clue Overlap:** The model captures the necessary emotion-related clues including facial expressions and actions, but certain finer details such as specific lighting conditions and attire descriptions show partial overlap. **[Score]: 80**

Figure 20: Visualization comparison results for fine-grained emotion caption. We achieve competitive performance with Gemini 2.0 on six different metrics.

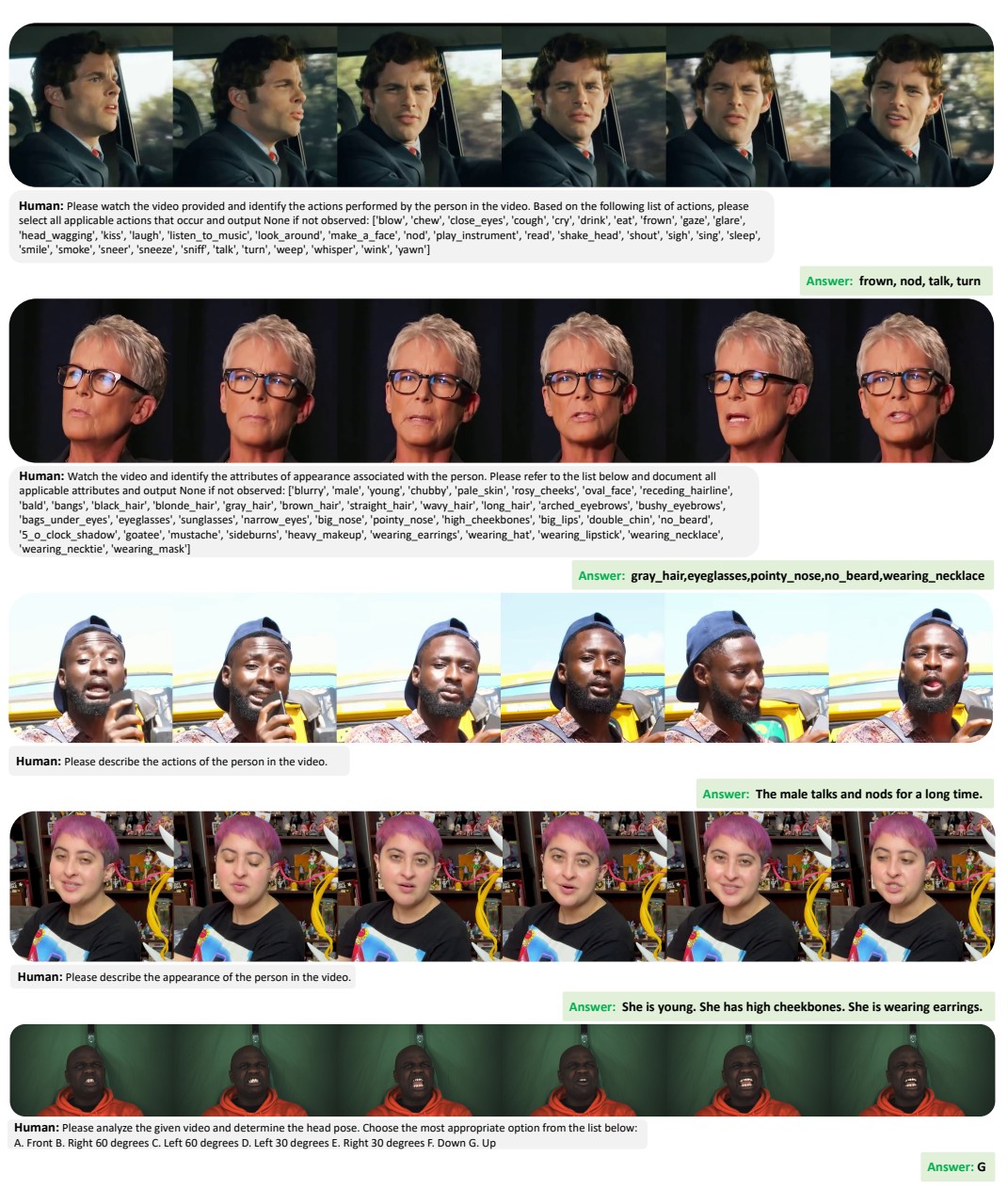

Figure 21: Visualization samples for attribute perception in classification-type tasks.

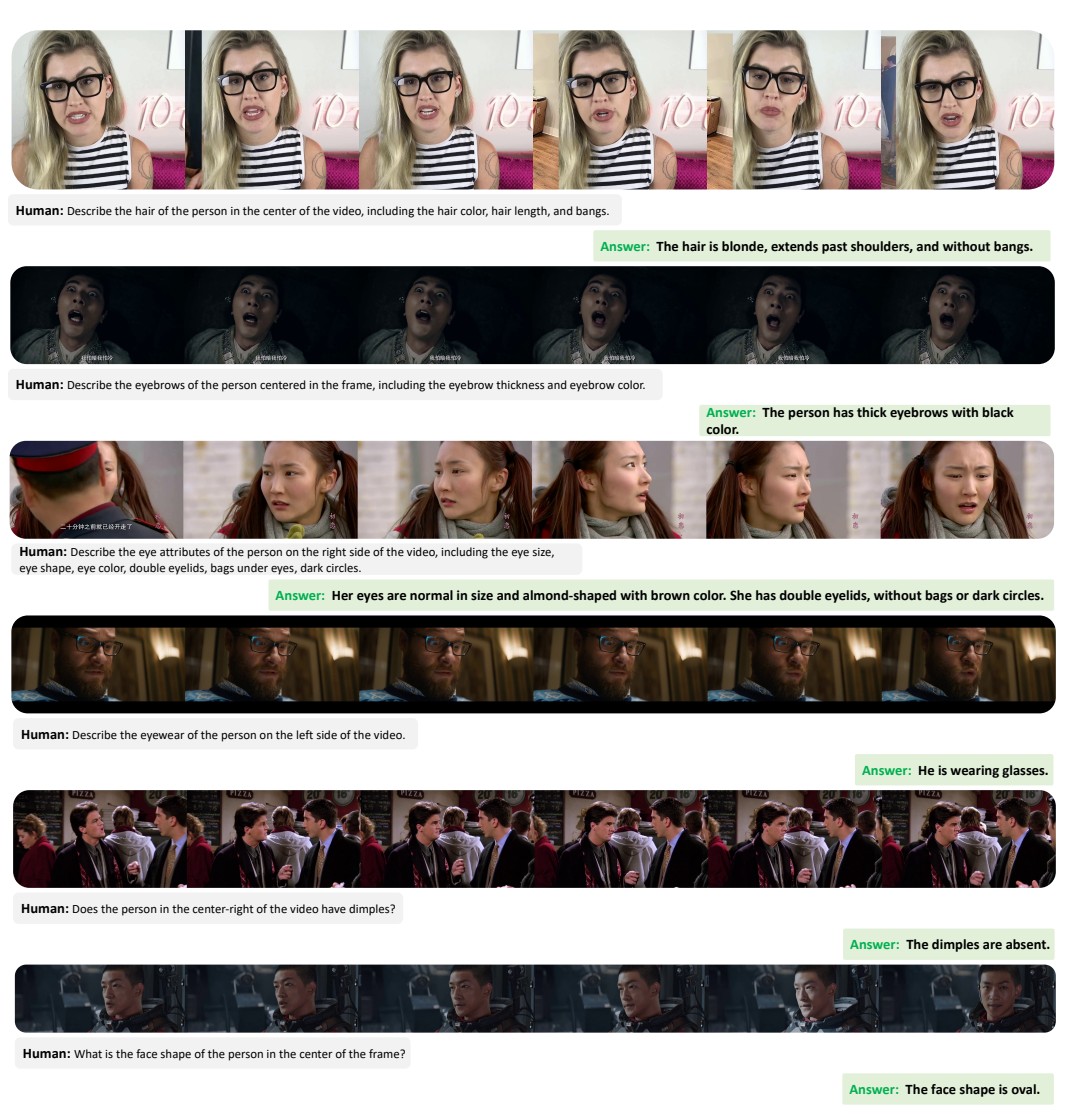

Figure 22: Visualization samples for attribute caption in caption-type tasks.

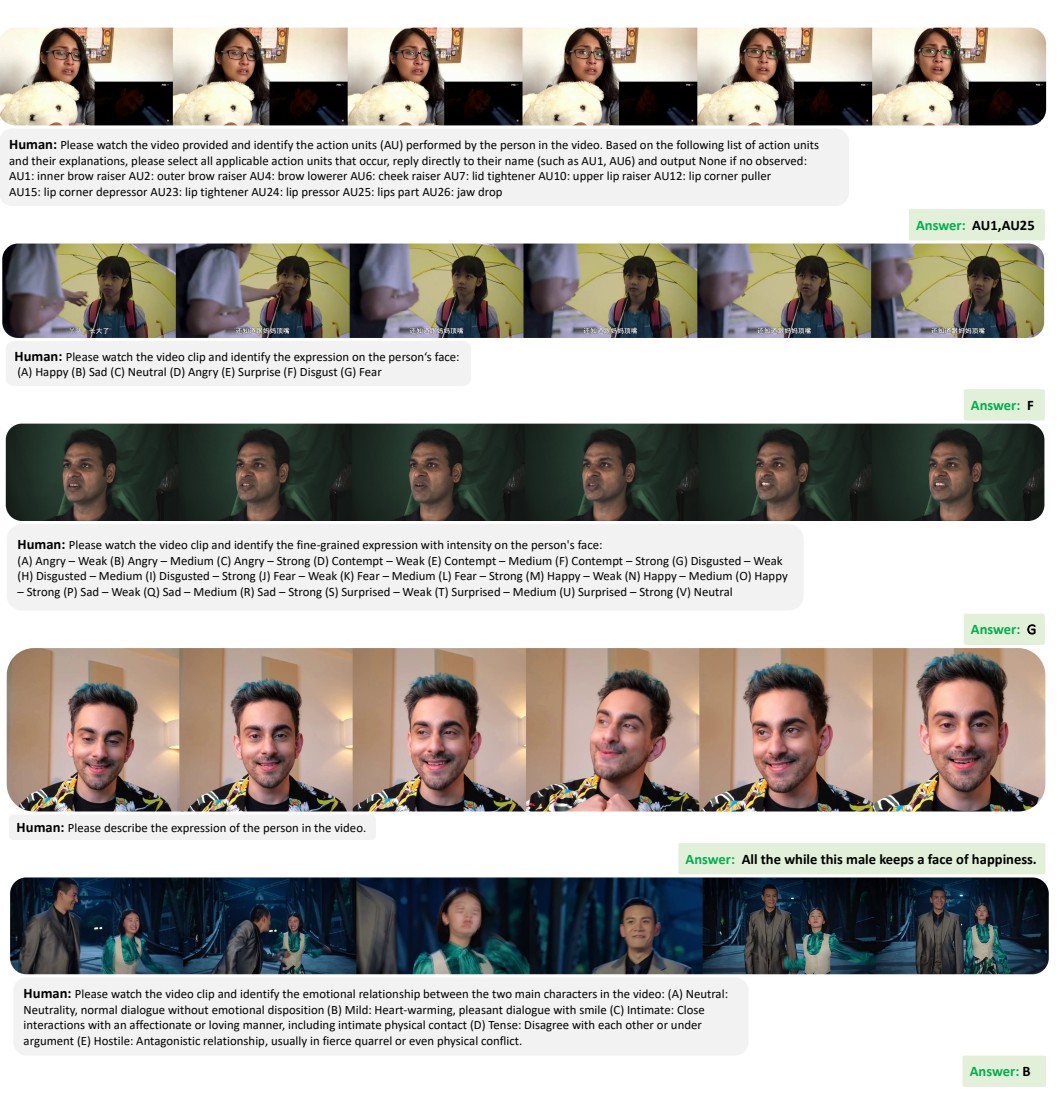

**Human:** Please watch the video provided and identify the action units (AU) performed by the person in the video. Based on the following list of action units and their explanations, please select all applicable action units that occur, reply directly to their name (such as AU1, AU6) and output None if no observed:
AU1: inner brow raiser AU2: outer brow raiser AU4: brow lowerer AU6: cheek raiser AU7: lid tightener AU10: upper lip raiser AU12: lip corner puller
AU15: lip corner depressor AU23: lip tightener AU24: lip pressor AU25: lips part AU26: jaw drop

**Answer: AU1,AU25**

**Human:** Please watch the video clip and identify the expression on the person's face:
(A) Happy (B) Sad (C) Neutral (D) Angry (E) Surprise (F) Disgust (G) Fear

**Answer: F**

**Human:** Please watch the video clip and identify the fine-grained expression with intensity on the person's face:
(A) Angry – Weak (B) Angry – Medium (C) Angry – Strong (D) Contempt – Weak (E) Contempt – Medium (F) Contempt – Strong (G) Disgusted – Weak
(H) Disgusted – Medium (I) Disgusted – Strong (J) Fear – Weak (K) Fear – Medium (L) Fear – Strong (M) Happy – Weak (N) Happy – Medium (O) Happy
– Strong (P) Sad – Weak (Q) Sad – Medium (R) Sad – Strong (S) Surprised – Weak (T) Surprised – Medium (U) Surprised – Strong (V) Neutral

**Answer: G**

**Human:** Please describe the expression of the person in the video.

**Answer: All the while this male keeps a face of happiness.**

**Human:** Please watch the video clip and identify the emotional relationship between the two main characters in the video: (A) Neutral:
Neutrality, normal dialogue without emotional disposition (B) Mild: Heart-warming, pleasant dialogue with smile (C) Intimate: Close
interactions with an affectionate or loving manner, including intimate physical contact (D) Tense: Disagree with each other or under
argument (E) Hostile: Antagonistic relationship, usually in fierce quarrel or even physical conflict.

**Answer: B**

Figure 23: Visualization samples for expression analysis.

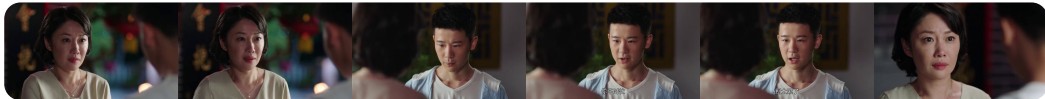

**Caption:** In a dimly lit indoor setting, a middle-aged East Asian woman with short, straight black hair and a fair complexion stands on the left side of the frame. Her head is slightly tilted downwards, and her gaze is directed off-screen to the right. Her downturned mouth and eyes, along with slightly furrowed eyebrows, give her a pensive and somewhat sad appearance. Tears are visible on her face as she engages in a serious conversation, her mouth moving as she speaks to a young adult East Asian man who enters the frame on the right side. The young man, with short cropped black hair, is wearing a light blue over-shirt and a simple white t-shirt. He listens attentively, his gaze fixed on the woman, and his expression ranges from neutral to slightly concerned, with furrowed eyebrows at times. He responds to the woman, speaking in return. The scene is serious, centered on the interaction between the woman and the man, with soft, diffused lighting casting subtle shadows on their faces. The blurred background hints at traditional East Asian decor. The camera focuses on their faces and upper bodies, emphasizing their expressions and exchange. The woman's sadness is the primary focus, while the man's attentiveness and slightly concerned expression suggest he is providing support or comfort.

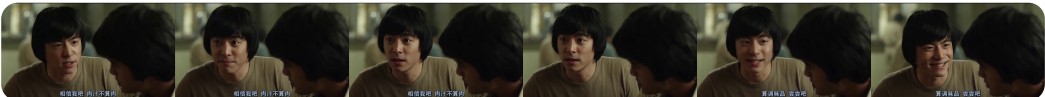

**Caption:** In a static close-up shot with soft, even lighting, two young East Asian men, likely in their young adulthood, are engaged in a conversation. On the left side of the frame, a man with short black hair styled in a bowl cut, a rounded nose tip, and wearing a light brown t-shirt appears clean-shaven. As he begins speaking, his head lifts slightly and turns towards the man on the right. His initially downturned mouth corners turn upwards into a smile, revealing his teeth as his smile widens. He seems happy or amused while talking, and this change in expression is directly related to his talking and smiling. Positioned on the right side of the frame, the other man also has short black hair in a bowl cut and wears a similarly colored shirt. He maintains a relatively still posture with his head turned towards the man speaking on the left, seemingly attentive to the ongoing conversation. Initially appearing neutral or perhaps pensive, his demeanor complements the static nature of the scene. Due to the close framing of the shot, no body actions are visible apart from the subtle head movements, and the scene remains static throughout with no apparent camera movement, giving the overall impression of a casual, friendly exchange between the two men.

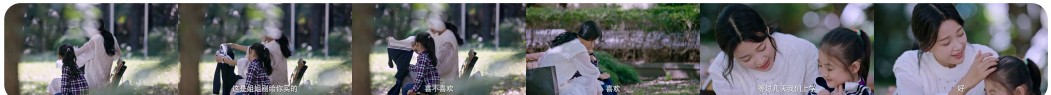

**Caption:** A young woman with fair skin, dark brown eyes, and shoulder-length black hair tied in a low ponytail sits on a bench in a sunlit park, wearing a light-colored cable-knit sweater vest, small pearl earrings, and a watch. She holds a dark blue jacket with white stripes, seemingly a school uniform, and presents it to a child sitting slightly lower on the bench. The child has fair skin and shoulder-length black hair styled in a low ponytail, adorned with colorful hair ties, and wears a plaid shirt and jeans. Initially, both the woman and the child gaze downwards at the jacket with neutral and focused expressions. The woman smiles affectionately at the child, her mouth corners upturning and eyes crinkling slightly, as she lifts her head slightly to the right to engage with the child. Her gentle touch as she adjusts the child's clothing reflects her caring demeanor. The child's expression softens in response, returning the woman's smile with a happy expression, their head lifting to meet her gaze. The child maintains eye contact, their smile widening to reveal their teeth in a relaxed and happy expression. The scene, bathed in soft, natural light that enhances the warmth and connection between them, is captured by a static camera focusing solely on their interaction.

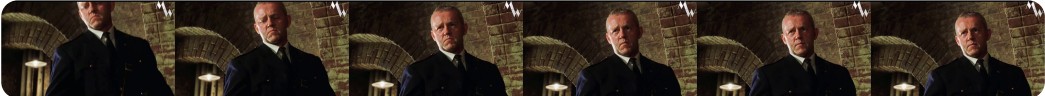

**Caption:** In a dimly lit setting featuring a brick archway and a hanging light, a middle-aged man, positioned centrally in the frame, is the focal point. He wears a dark suit and a black necktie, presenting a formal appearance. His short hair is light brown, and he has some stubble on his face. His sideburns are visible, and he is otherwise clean-shaven. He is not wearing glasses or any visible accessories other than his tie. He maintains a serious expression throughout, characterized by a slight furrow in his brows and downturned mouth corners. His head is slightly tilted downwards, and his gaze remains directed off-screen, focused on something below his line of sight. He remains completely still, exhibiting no body movement. His facial expression, along with his fixed gaze and still posture, conveys an air of seriousness and focused attention. The man's lack of movement and unchanging expression emphasizes his absorption in whatever lies beyond the frame. The background remains constant, with no change in camera angle or movement. The dim lighting and the arched brick passageway contribute to the scene's overall somber and serious tone

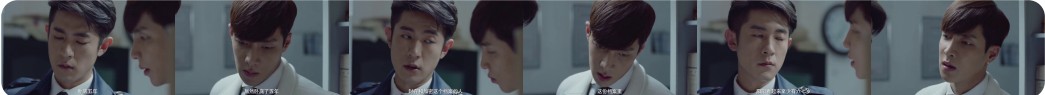

**Caption:** The video features two men, both seemingly East Asian, engaged in a serious discussion against a plain indoor backdrop. One young adult man, positioned on the left side of the frame, wears a police uniform with a dark blue tie featuring lighter blue stripes. His short, dark hair is styled upwards away from his forehead, revealing double eyelids, almond-shaped eyes, and a high nose bridge with a rounded tip. His lips have a natural pinkish hue. He maintains a focused and attentive gaze towards the man on the right, his head slightly tilted and turned. Subtle movements of his lips and jaw indicate he's speaking, while his serious, neutral expression and largely unchanged posture suggest a focused conversation. The second young adult man appears more casually dressed in a dark jacket over a white collared shirt and a dark necktie. His short, dark hair is styled with bangs that fall just above his eyebrows. His double eyelids are less prominent, and his nose features a high bridge and a pointed tip. His lips, like the first man's, have a natural pinkish hue. He mainly listens attentively, his head slightly turned towards the man in the police uniform. His downward gaze and serious expression indicate contemplation or reserved reaction. Subtle lip movements suggest brief verbal responses as he primarily engages through active listening. The static nature of their interaction is emphasized by neither man making significant body movements; their upper bodies remain mostly still, underlining the focus on their verbal exchange. Both men's faces are evenly illuminated, highlighting their features and expressions as they converse.

Figure 24: Visualization samples for emotion understanding in fine-grained caption task.

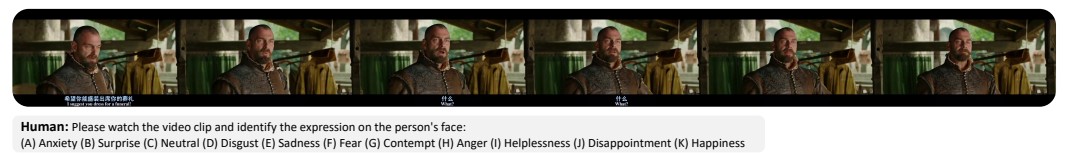

**Human:** Please watch the video clip and identify the expression on the person's face:
(A) Anxiety (B) Surprise (C) Neutral (D) Disgust (E) Sadness (F) Fear (G) Contempt (H) Anger (I) Helplessness (J) Disappointment (K) Happiness

**Answer: B**

**Human:** Please analyze which parts of the environment, lighting, background, visual attributes, etc., can explain the reasons for your answer.

**Reasoning:** The facial recognition analysis begins by focusing on the attributes relevant to the expression of surprise, such as eye movements and changes in the mouth's position, indicators commonly associated with this emotion. In situations involving surprise, the eyes typically widen to increase the field of vision, which corresponds with attentive readiness to unexpected stimuli. However, the current information on eye features in this analysis is limited—the eye corners are noted as neutral, and specifics about eye widening are not provided, potentially due to lighting and image quality issues.
Nonetheless, the mouth's activity offers notable insight; it is initially slightly open, aligning well with a surprised expression where the mouth often naturally opens as part of a reflex to unexpected events. The subsequent closing of the mouth might suggest the surprise is resolved swiftly, an observation supported by the brief time frame of seven seconds.
The man's head remains straight and directed forward, further lending to the context of engagement with a surprising stimulus, with the gaze directed just above the camera. While the gaze itself isn't explicitly widened, its direction suggests an encounter with something unexpected above the usual line of sight, characteristic of a surprise response.
Despite the neutral positioning of the eyebrows, which may also raise slightly in surprise, they do not contradict the expression given their stability within the limited provided data.
Finally, the lighting conditions (diffused and natural) imply that visibility is adequate for genuine micro-expressions to be captured, though some shadows exist. The presence of natural light without obscuring accessories indicates authenticity in the individual's reaction.
In summary, a composite view of the attributes—the initial mouth opening, the direct yet slightly elevated gaze, and the contextual clues around these attributes— suggests an interpretation aligned with the physiological markers of surprise. Despite limited detail on eye widening and eyebrow activity, the coordination of mouth action, gaze direction, and overall face positioning corroborates a fleeting, possibly mild, surprise reaction.

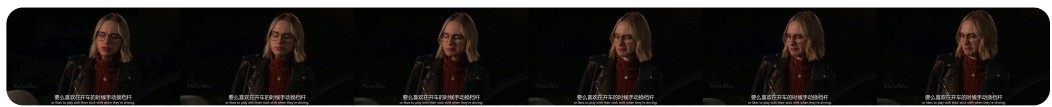

**Human:** Please watch the video clip and identify the expression on the person's face:
(A) Anxiety (B) Surprise (C) Neutral (D) Disgust (E) Sadness (F) Fear (G) Contempt (H) Anger (I) Helplessness (J) Disappointment (K) Happiness

**Answer: D**

**Human:** Please analyze which parts of the environment, lighting, background, visual attributes, etc., can explain the reasons for your answer.

**Reasoning:** The analysis of the video's emotional indicators begins with facial expressions that convey clear emotional messages. The positioning of the eyes provides a vital signal; they are slightly downturned at the corners, which is often associated with negative emotions such as disapproval or unhappiness. Complementarily, the downturned corners of her mouth further suggest a negative emotional tone, potentially reflecting disapproval or distaste.
The eyebrows, however, offer one of the most significant clues—being slightly furrowed. This action is a classic sign of disgust or disapproval, as it suggests a focused, negative evaluation of something unpleasant or disagreeable. The head posture, with a slight downward tilt, adds another layer by potentially indicating a reflective or introspective stance, which often accompanies emotional states involving judgment or disapproval.
The gaze direction—looking downward and slightly to the left—can also imply an avoidance of direct eye contact, possibly supporting the notion of feeling negatively about an unseen stimulus or interlocutor. Eyeglasses and low lighting might obscure finer details, but the analysis of the above features provides a cohesive understanding of the emotional state.
Taken together, these features—the downward-tilted head, the slightly furrowed brows, downturned mouth corners, and downward gaze—form a synergistic pattern commonly associated with the expression of disgust. This combination suggests an emotionally charged evaluation process characteristic of perceived distaste or aversion.

Figure 25: Visualization samples for emotion understanding in rationale analysis task.

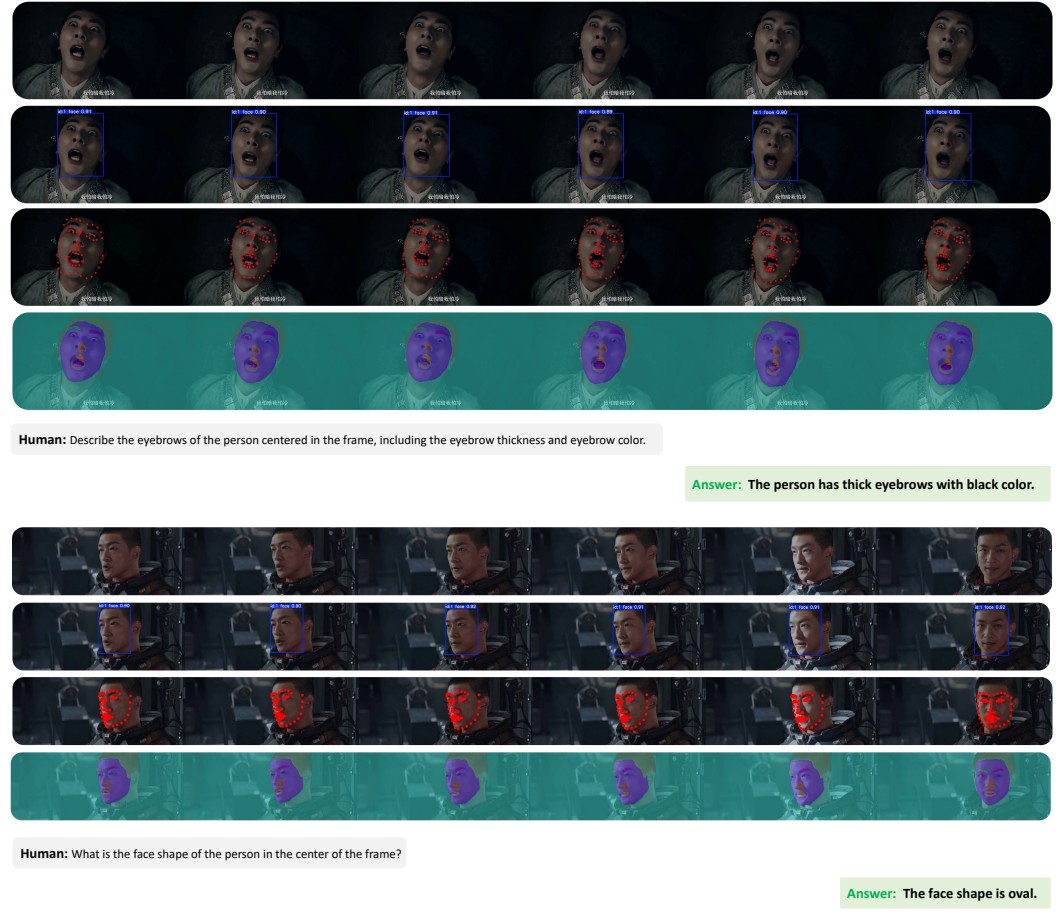

**Human:** Describe the eyebrows of the person centered in the frame, including the eyebrow thickness and eyebrow color.

**Answer:** The person has thick eyebrows with black color.

**Human:** What is the face shape of the person in the center of the frame?

**Answer:** The face shape is oval.

Figure 26: Visualization samples of meta labels of face boxes, face landmarks and parsing masks.

