# OpenReview forum: "VidEmo: Affective-Tree Reasoning for Emotion-Centric Video Foundation Models"
_NeurIPS.cc/2025/Conference — NeurIPS 2025 poster_

### Official Review · Reviewer_JqQW · 2025-06-26

**Clarity:** 3
**Significance:** 3
**Originality:** 3
**Rating:** 5
**Confidence:** 4

**Summary:**

This paper introduces VidEmo, a family of emotion-centric video foundation models trained with a novel two-stage training pipeline: Curriculum Emotion Learning and Affective-Tree Reinforcement Learning. It also presents a new large-scale benchmark dataset, Emo-CFG, containing 2.1M annotated samples with fine-grained emotional reasoning, captions, and rationales. The approach outperforms state-of-the-art models on 15 facial perception and emotion tasks, including closed and open-source baselines.

**Questions:**

Section 4.2 is not very clear about how the affective tree is parsed from the generated captions. Could you elaborate more on this aspect?

**Ethical Concerns:**

["NO or VERY MINOR ethics concerns only"]

**Final Justification:**

The Authors addressed all my questions adequately. I particularly appreciated that they have decided to include the suggested ablation study, which confirms the effectiveness of the proposed curriculum learning.

I have also read the remaining reviews and the Authors' responses to them. I believe they addressed the vast majority of the concerns of Reviewer ntby, who recommended rejection, with valid arguments.

Thus, I decided to keep my score (5), recommending the acceptance of this manuscript.

**Limitations:**

Limitations are appropriately acknowledged. I suggested one more is added in the "Weaknesses" section of my review.

**Paper Formatting Concerns:**

Nothing

**Quality:**

4

**Strengths And Weaknesses:**

Strengths:
- The affective-tree structure elegantly decomposes emotional understanding into interpretable stages.
- The post-hoc reinforcement learning (via GRPO) is a very sensible option that proves to bring additional gains.
- Emo-CFG is a strong contribution: well-annotated and validated using committee-based LLM feedback.
- The system is designed to provide high interpretability, increasing transparency in a typically opaque task.
- The experiments show superior performance over 18 strong baselines.

Weaknesses:
- As acknowledged by the authors, their model uses only on visual data. The integration of audio or textual context (e.g., speech, subtitles), which is left as future work, would certainly bring better and more robust predictions, as emotional expression often depends on these.
- An ablation study proving the improvements of the curriculum learning (vs. regular random training) and only post-training without curriculum learning is missing. It would be great if the authors could add this study to the appendices.
- Although the committee-based verification helps, reliance on LLM-generated labels and validation could encode biases from those models. The authors should add this limitation/risk to the appropriate section.

---

> ### Author Rebuttal · Authors · 2025-07-31
>
> # Response to Reviewer JqQW
>
> Thank you for your insightful comments and questions. For your reference, we summarized the main results in our response to Reviewer KQSY.
>
> **A1: Multimodal Support** We agree that multimodal support is essential for advancing emotion understanding and plays a key role in future research. Our work lays a strong foundation for future multimodal integration from two key perspectives. (1) **Pretraining Data**: VidEmo scales the pretraining dataset from 60K (e.g., VCE) to 2.1M multimodal emotion-centric samples, many of which include paired text and are suitable for audio-text expansion. (2) **Model Transferability**: VidEmo’s structured affective reasoning and distilled emotion logits can be readily used to supervise downstream multimodal models, thereby enhancing their emotional grounding and reasoning capabilities. To support follow-up research, we will release the model, dataset, and code upon acceptance.
>
>
>
> **A2: Ablation Studies** As suggested, we conducted an ablation study to verify the effectiveness of the proposed curriculum learning strategy.
>
> (1) Curriculum learning vs. Regular Training during the SFT stage: We compare our curriculum learning strategy with conventional random training during the supervised fine-tuning stage. As shown below, curriculum learning yields substantial gains across all tasks, improving the overall average by +10.5.
>
> | Method              | Attribute | Expression | Emotion  | Average  |
> | ------------------- | --------- | ---------- | -------- | -------- |
> | Regular Training    | 63.5      | 27.3       | 63.6     | 51.4     |
> | Curriculum learning | **79.5**  | **38.7**   | **67.5** | **61.9** |
>
> (2) Post-training with vs. without Curriculum Learning in the RL Stage: To further evaluate its impact, we compare post-training (via GRPO) initialized with and without curriculum learning. The results show that curriculum-informed initialization significantly boosts emotion understanding performance, leading to a +12.4 gain in overall average.
>
> | Method                                 | Attribute | Expression | Emotion  | Average  |
> | -------------------------------------- | --------- | ---------- | -------- | -------- |
> | Post-training w.o. Curriculum learning | 75.7      | 33.8       | 44.2     | 51.2     |
> | Post-training w. Curriculum learning   | **81.3**  | **40.1**   | **69.3** | **63.6** |
>
> For more detailed results, please refer to Tables 3–6 in our supplementary materials.
>
>
>
> **A3: Limitation induced by LLM bias** As suggested, we investigated potential biases and discussed the findings. This discussion will be included in Section 3 as part of our limitations analysis.
>
> **(1) Human Label Verification**: As suggested, we further validated label fidelity by extracting fine-grained captions from the generated videos and comparing them with the original human-assigned labels of CelebV-Text. The consistency rate reached 90.3%.
>
> **(2) User Study Verification**: To assess the quality of the generated expressions, we conducted a user study on a manually inspected subset of test samples to verify their alignment with the intended emotional semantics. Specifically, we compared Emo-CFG with CelebV-Text, the largest human-labeled video emotion dataset, across three key dimensions: precision, rationality, and complementarity. Preference rates for Emo-CFG across these dimensions reached 95%, 92%, and 93%, respectively, with statistically significant differences (Wilcoxon signed-rank test, *p* < 0.01). This result demonstrates that Emo-CFG provides more precise and expressive emotional representations than existing benchmarks.
>
> | Dimension       | Win  | Tie  | Loss | # Video | # User | Prefer |
> | --------------- | ---- | ---- | ---- | ------- | ------ | ------ |
> | Precision       | 964  | 204  | 82   | 50      | 25     | 95.52% |
> | Rationality     | 1082 | 87   | 81   | 50      | 25     | 92.16% |
> | Complementarity | 1172 | 23   | 55   | 50      | 25     | 93.04% |
>
>
>
> **A4: Affective Tree Parsing** We appreciate the reminder. The affective tree parsing described in Section 4.2 is implemented via a hybrid pipeline combining rule-based extraction and LLM-based structure inference. First, we use a customized NER-style extractor built on the SpaCy framework to identify aspect–item pairs from the generated captions, covering three semantic levels: low-level attributes (e.g., “closed eyes”, “shoulder-length hair”), mid-level expressions (e.g., “frowning”, “gasping”), and high-level emotions (e.g., “fear”, “sadness”). Next, we use a few-shot prompted LLM (InternLM-XComposer 2.5) to construct the hierarchical reasoning structure from the generated caption. The model outputs a structured JSON-like affective tree that connects attribute → expression → emotion through rationale-informed links. This structured output forms the predicted tree $T{\text{pred}}$, which is compared against the ground truth $T{\text{gt}}$ for reward computation. An example of the parsed affective tree can be found in Fig. 6c of the manuscript.

---

> ### Author Response · Authors · 2025-08-05
> **Looking Forward to Your Valuable Feedback**
>
> Dear reviewer,
>
> Thank you once again for taking the time to review our paper and for providing such insightful and constructive feedback. Your comments have been invaluable in helping us further polish our manuscript.
>
> We have carefully considered each of your comments and have provided detailed responses. We sincerely hope that our efforts adequately address your concerns and contribute positively to your evaluation.
>
> As the author-reviewer discussion period has been extended until Aug 8, we are fortunate to have sufficient time to collaboratively discuss and refine the manuscript. We would greatly appreciate any additional feedback you may have. If you have any additional questions or require any clarifications, please do not hesitate to reach out to us.

---

> > ### Comment · Reviewer_JqQW · 2025-08-05
> > **Response to Authors' rebuttal**
> >
> > Dear Authors,
> >
> > Thank you for addressing all my questions adequately. I particularly appreciated that you have decided to include the suggested ablation study, which confirms the effectiveness of the proposed curriculum learning.
> >
> > I have also read the remaining reviews and your responses to them. In particular, I think you addressed the vast majority of the concerns of Reviewer ntby, who recommended rejection, with valid arguments.
> >
> > Thus, I decided to keep my score (5), recommending the acceptance of your manuscript.

---

> > > ### Author Response · Authors · 2025-08-05
> > >
> > > Dear reviewer JqQW,
> > >
> > > Thank you for kindly recognizing our contributions to Emotional VideoLLM and providing invaluable comments on this work. We authors greatly appreciate the efforts you have made to improve our manuscript. If accepted, we will include `JqQW` in our acknowledgments.
> > >
> > > Best Regards,
> > > Authors of paper 15702

---

### Official Review · Reviewer_ntby · 2025-06-30

**Clarity:** 3
**Significance:** 3
**Originality:** 2
**Rating:** 3
**Confidence:** 4

**Summary:**

This paper proposes a novel video emotion base model called VidEmo, which enables fine-grained and interpretable understanding of complex emotions in dynamic videos by combining curriculum emotion learning and affective-tree reasoning frameworks. The paper also constructs an emotion-centric fine-grained dataset, Emo-CFG, containing 2.1 million diverse instruction-based samples, which provides an important resource for emotion understanding tasks. Experimental results show that VidEmo outperforms existing open- and closed-source models in 15 face perception tasks, with advantages in emotion understanding and reasoning capabilities in particular. The authors also point out the limitations of the model in terms of possible generation of false content and multimodal integration, which provide directions for future research.

**Questions:**

See Weaknesses 1~5

**Ethical Concerns:**

["NO or VERY MINOR ethics concerns only"]

**Final Justification:**

The authors gave detailed responses in their rebuttal, which have addressed most of my concerns. However, there is a certain gap between the author's rebuttal and the original manuscript. Much of the important information, as well as the related analysis and discussion, is disclosed in the rebuttal but is completely missing from the original manuscript. The rebuttal appears to be an extension of the original manuscript rather than an explanation of its unclear points. Frankly speaking, the prerequisite for this paper to be accepted for publication is that the manuscript undergoes thorough revision and improvement. Furthermore, in my opinion, when stripped of the trappings of large language models, the innovation and contribution of this paper are actually limited. There is still a certain gap compared to the high standards of NeurIPS. Therefore, given the author's careful and thoughtful response, I am willing to raise my score, but I still do not recommend acceptance.

**Limitations:**

The proposed models of this paper may be used for emotion monitoring or privacy invasion (e.g., inferring mental states through micro-expression analysis), but the paper only briefly mentions "limitations" without discussing specific safeguards (e.g., data desensitization or usage restrictions).

**Quality:**

3

**Strengths And Weaknesses:**

Strengths:

1. A new video emotion base model (VidEmo) is proposed, combining curriculum emotion learning (phased injection of emotion knowledge) and affective-tree reinforcement learning (hierarchical reasoning) to enable fine-grained analysis and interpretable reasoning about dynamic and complex emotions in videos. The models derived from the base model achieve better performance on 15 facial perception tasks (e.g., attribute recognition, expression analysis, emotion understanding) compared to existing VideoLLMs.
2. A new emotion-centric fine-grained dataset (Emo-CFG) is constructed. It covers detailed annotations across attributes, expressions and emotions, and supports full chain learning from low-level attributes to high-level emotions. The data quality is ensured through a committee voting-based verification strategy, which addresses the inherent ambiguities in emotional data due to its subjective nature.

Weaknesses:

1. The source data of the proposed Emo-CFG dataset is from 17 existing datasets. The paper does not give an in-depth analysis and discussion on the question of whether the possible bias in sample labeling and distribution between different source datasets affects the quality of Emo-CFG.
2. The generation of some of the annotations in the constructed dataset relies on GPT-4o and Gemini 2.0, which may introduce biases or errors inherent to large models, and the paper does not analyze in detail the impact of differences between synthetic data and manual annotations.
3. Affective-tree rewards may have limitations. The construction of ground-truth tree may be subjectively influenced by the annotator. The consistency of reasoning paths with human intuition was not quantitatively analyzed in the paper.
4. The authors acknowledge that the model may generate "false narratives" or "emotionally inconsistent descriptions" but do not propose specific mitigating measures (e.g., post-processing filtering or uncertainty calibration) that may affect the reliability of actual deployments of the proposed models.
5. The evaluation is insufficient. The proposed model is not compared with the latest dedicated expression/emotion analysis models (e.g., Emo-LLaMA, Exp-CLIP, ExpLLM) on the same dataset, and open-source model comparisons are focused on generic VideoLLMs, which makes it difficult to prove that it outperforms the optimal models in the field of expression/emotion analysis.

---

> ### Author Rebuttal · Authors · 2025-07-31
>
> # Response to Reviewer ntby
>
> Thank you for your insightful comments and questions. For your reference, we summarized the main results in our response to Reviewer KQSY.
>
> **A1: Dataset Source**
>
> (1) In fact, we conducted a comprehensive literature review and initially collected 108 video datasets following the emotion data taxonomy (Fig. 4a). To mitigate source bias, we applied a series of filtering rules to select 17 high-quality datasets that offer broad coverage across video length (Fig. 4b), visual clarity (Fig. 4b), and caption-related affective cues (Fig. 4c). Datasets exhibiting strong domain skew or poor annotation quality are deliberately excluded to ensure the reliability and representativeness of the final dataset composition.
>
> (2) To directly address concerns regarding source-induced bias, we have conducted an ablation study where samples from different data sources and emotional stages (attribute → expression → emotion) are progressively integrated (Tab. 7 of supplementary). The results demonstrate that multi-stage curriculum learning, which incorporates all data sources in a structured manner, significantly outperforms single-stage variants across all tasks.
>
> | Data Source      | Attribute | Expression | Emotion  | Average  |
> | ---------------- | --------- | ---------- | -------- | -------- |
> | Attr             | 65.7      | 24.1       | 32.4     | 40.7     |
> | Exp              | 52.9      | 31.6       | 42.1     | 42.2     |
> | Emo              | 64.3      | 30.2       | 67.3     | 53.9     |
> | Attr+Exp         | 77.2      | 35.1       | 45.5     | 52.6     |
> | **Attr+Exp+Emo** | **79.5**  | **38.7**   | **67.5** | **61.9** |
>
>
>
> **A2: Dataset Annotation**
>
> (1) **Cognition-inspired Human-led, Model-assisted Data Construction**: Indeed, Emo-CFG is not solely labeled by VideoLLMs, but  also constructed through a human-led, model-assisted pipeline that incorporates cognitive principles and human priors. Inspired by abstract human cognition process [r4,r5], we design a causal perception-to-emotion pathway that begins with stimulus perception (Attribute), proceeds through holistic expression organization (Expression), and culminates in emotion understanding (Emotion)—as detailed in Sec. 3. Further, we adopted a *human-led, model-assisted* pipeline during dataset construction [r6]. Human annotators first provide attribute and expression labels as semantic scaffolding, and models then generate fine-grained captions conditioned on these labels. This design ensures alignment with human cognitive logic while maintaining scalability and annotation efficiency.
>
> (2) **Human Label Verification**: As suggested, we further validated label fidelity by extracting fine-grained captions from the generated videos and comparing them with the original human-assigned labels of CelebV-Text. The consistency rate reached 90.3%.
>
> (3) **User Study Verification**: To assess the quality of the generated expressions, we conducted a user study on a manually inspected subset of test samples to verify their alignment with the intended emotional semantics. Specifically, we compared Emo-CFG with CelebV-Text, the largest human-labeled video emotion dataset, across three key dimensions: precision, rationality, and complementarity. Preference rates for Emo-CFG across these dimensions reached 95%, 92%, and 93%, respectively, with statistically significant differences (Wilcoxon signed-rank test, *p* < 0.01). This result demonstrates that Emo-CFG provides more precise and expressive emotional representations than existing benchmarks.
>
> | Dimension| Win  | Tie  | Loss | # Video | # User | Prefer | p-value | Significance |
> | - | - | - | - | - | - | - | - | - |
> | Precision       | 964  | 204| 82| 50| 25| 95.52% | 0.00021 | Significant  |
> | Rationality     | 1082 | 87   | 81| 50| 25| 92.16% | 0.00015 | Significant  |
> | Complementarity | 1172 | 23| 55| 50| 25| 93.04% | 0.00008 | Significant  |
>
>
>
> **A3: Affective Tree Rewards**
>
> (1) We fully agree that emotion understanding inherently involves subjective interpretation. To reduce subjective bias in constructing the affective tree, we employed a voting-by-committee strategy. Multiple annotators provide independent annotations and a majority vote is used to resolve disagreement, which is consistent with best practices in emotion datasets.
>
> (2) The results of the user study show that our affective trees are consistently preferred in terms of complementarity, precision, and rationality, with an especially high 93% preference rate in the rationality dimension for the reasoning path. This indicates that the reasoning paths in our dataset closely align with human affective interpretation. Moreover, incorporating the reasoning path as a reward signal leads to consistent improvements over the GRPO baseline across all sub-goals (attribute, expression, emotion). Further integrating tree edit distance (ATR) into the reward function provides additional gains, underscoring the effectiveness of structure-aware supervision.
>
> | Method| Attribute | Expression | Emotion  | Average  |
> | - | - | -| - | - |
> | GRPO (w/o Tree Reward)| 75.7| 33.8| 44.2| 51.2|
> | + Tree Reward| 80.9| 38.1| 65.3| 61.4|
> | + Tree Edit Distance (ATR) | **81.3**  | **40.1**   | **69.3** | **63.6** |
>
>
>
> **A4: Mitigating Hallucination** We thank the reviewer for recognizing our analysis of harmful false narratives. As described in Section 4.3 of the original manuscript, we propose an emotion-centric post-processing strategy to mitigate hallucinated outputs. Specifically, we sample multiple descriptions at each stage (attribute, expression, emotion) and select the one with the highest logit score. This filtering process effectively eliminates most false or irrational narratives. Our ablation study shows that increasing the number of sampled candidates improves performance. With higher sampling, the model achieves better results, yielding an average improvement of +3.4.
>
> | Method| Attribute | Expression | Emotion  | Average  |
> | -| - | - | - | - |
> | Baseline (n=1)| 81.3| 40.1| 69.3| 63.6|
> | + Emotion Reasoning (n=2) | 82.9| 42.3| 70.1| 65.1|
> | + Emotion Reasoning (n=4) | 84.2| 43.2| 71.2| 66.2|
> | + Emotion Reasoning (n=8) | **84.5**  | **43.8**   | **72.9** | **67.0** |
>
>
>
> **A5: Comparison with SOTA Expression/Emotion Analysis Models**
>
> (1) The three models mentioned (Emo-LLaMA, Exp-CLIP, and ExpLLM) are not taken into consideration for direct comparison in our setting due to the following reasons:
>
> - Emo-LLaMA is not open-sourced, making it impossible to reproduce results or conduct fair comparisons.
> - Exp-CLIP is designed for image/video-text retrieval tasks, and is therefore not directly comparable to our setup.
> - ExpLLM is a still-image-based model. Its public release was in April 2025, just one month before the NeurIPS deadline, violating the NeurIPS guideline that excludes concurrent work not published prior to submission.
>
> (2) As suggested, we directly compared VidEmo with existing emotion-specific models on the same benchmarks (MAFW and DFEW) using their reported numbers. With the affective tree reasoning, VidEmo achieves a relative improvement of +4.7 UAR / +7.2 WAR on DFEW and +5.9 UAR / +12.8 WAR on MAFW.
>
> | Model|DFEW UAR (%)|DFEW WAR (%)| MAFW UAR (%) | MAFW WAR (%) |
> | - | - | - | - | - |
> | EMO-LLAMA|60.23| 65.89| 41.57| 48.63|
> | **VidEmo (Ours)** |**64.92**| **73.10**| **44.02**| **54.86**|
>
>
>
> **A6: Significance and Limitation**
>
> (1) **Scientific Value and Significance**: Emotion understanding is a long-standing and impactful research direction across disciplines, including computer science, neuroscience, and psychology. In recent years, there has been an increasing trend of emotion-related topics in high-impact publications such as ***Nature***, ***Science***, and ***Cell***, with 21, 35, and 9 papers published in the last three years, respectively. Our work contributes to this broader scientific effort by modeling video emotion in a cognitively inspired and interpretable manner.
>
> (2) **Emerging Research Trend**: In the deep learning area, recent LLM-enhanced emotion works published in top-tier conferences such as ICML 2025 oral [r6], ICLR 2024 oral [r7], and NeurIPS 2023 spotlight [r8] demonstrate the field’s growing importance and wide recognition. Our work follows this emerging trend and focuses on understanding emotion in a responsible, transparent, and reproducible manner.
>
> (3) **Responsible Usage and Licensing**. All data used in our work is sourced from publicly available, open-source datasets. No private or sensitive data is involved, and we ensure compliance with the original licenses of these datasets. Additionally, we are committed to research-only usage. All models, datasets, and code will be released under the MIT license, with explicit restrictions stated for non-commercial and non-surveillance use.
>
>
>
> [r1] Mazeika, Mantas, et al. "How would the viewer feel? Estimating wellbeing from video scenarios." NeurIPS, 2022.
>
> [r2] Guo, Yuxiang, et al. "Stimuvar: Spatiotemporal stimuli-aware video affective reasoning with multimodal large language models." IJCV, 2025.
>
> [r3] Cui, Ganqu, et al. "The Entropy Mechanism of Reinforcement Learning for Reasoning Language Models." arXiv:2505.22617, 2025.
>
> [r4] Zhao, Sicheng, et al. "Emotion recognition from multiple modalities: Fundamentals and methodologies." SPM, 38(6): 59-73, 2021.
>
> [r5] Grill-Spector, Kalanit and Malach, Rafael. "The human visual cortex." Annual Review of Neuroscience, 27:649–677, 2004
>
> [r6] Lian, Zheng, et al. "AffectGPT: A New Dataset, Model, and Benchmark for Emotion Understanding with Multimodal Large Language Models." ICML, 2025.
>
> [r7] Huang, Jen-tse, et al. "On the Humanity of Conversational AI: Evaluating the Psychological Portrayal of LLMs." ICLR, 2024.
>
> [r8] Chandra, Kartik, et al. "Inferring the Future by Imagining the Past." NeurIPS, 2023.

---

> > ### Author Response · Authors · 2025-08-05
> > **Looking Forward to Your Valuable Feedback**
> >
> > Dear reviewer,
> >
> > Thank you once again for taking the time to review our paper and for providing such insightful and constructive feedback. Your comments have been invaluable in helping us further polish our manuscript.
> >
> > We have carefully considered each of your comments and have provided detailed responses. We sincerely hope that our efforts adequately address your concerns and contribute positively to your evaluation.
> >
> > As the author-reviewer discussion period has been extended until Aug 8, we are fortunate to have sufficient time to collaboratively discuss and refine the manuscript. We would greatly appreciate any additional feedback you may have. If you have any additional questions or require any clarifications, please do not hesitate to reach out to us.

---

> > ### Author Response · Authors · 2025-08-06
> > **Gentle Request for Your Valuable Feedback**
> >
> > Thank you once again for your insightful comments on our paper. We deeply appreciate the time and effort you have dedicated to helping us improve our work.
> >
> > We apologize for the repeated follow-ups, but your input is important to us. As the discussion period is drawing to a close, we kindly request your feedback on our responses at your earliest convenience, as we may not have enough time to make further revisions otherwise. If you feel that our responses have addressed your concerns, we would be grateful if you could consider updating your evaluation.
> >
> > Please feel free to reach out if anything remains unclear. Thank you again for your thoughtful contributions, and we look forward to hearing from you soon.

---

> > ### Comment · Reviewer_ntby · 2025-08-07
> >
> > Thank you for your detailed responses and hard work. However, these responses did not address all of my concerns and seemed somewhat confusing. I had to put in more effort to carefully examine the original manuscript and its supplementary materials to find content that supported the claims in the rebuttal. This is why I responded to the authors' rebuttal so late. Nevertheless, the following concerns remain.
> >
> > (1) Regarding dataset construction and annotation, based on the description in the original manuscript, human involvement is minimal. The claim of "human-led" in the rebuttal is somewhat exaggerated. To reiterate, although the authors emphasize that they selected 17 datasets from 108, the dataset constructed in the paper is still based on 17 existing datasets, with the majority of subsequent annotation work relying on various large language models. Regarding the impact of differences between synthetic data and manual annotations, the authors have not provided sufficient analysis. The two simple experiments attached are too limited and lack sufficient persuasiveness. Additionally, the explanation of the User Study Verification is unclear, and its validity is questionable. By the way, the aforementioned related analysis and discussion are crucial for constructing a dataset, but they are either missing or insufficient in the original manuscript.
> >
> > (2) As for "Mitigating Hallucination", I don't think Section 4.3 (only five lines of text) fully describes strategies for mitigating hallucination output. And based on the wording of the original manuscript, it doesn't seem to be specifically about "Mitigating Hallucination" either. Using the content of Section 4.3 to address the limitation of "false narratives" raised by the authors themselves is too far-fetched.
> >
> > (3) The evaluation of the model proposed in the paper is precisely the biggest shortcoming of this paper. The three works I mentioned in my review comments are only a few examples for the authors' reference, not to say that there are only these three (BTW, the arXiv version of ExpLLM is released in September 2024). In fact, there have been many related works to date, and the additional experimental results provided by the authors in the rebuttal are very limited and insufficient to prove the superiority of the paper's work. To elaborate, the paper's work on verification has the following major shortcomings.
> >
> > (i) Missing Comparisons with Traditional Models: The experiments only compare VidEmo against untuned large-scale models (e.g., Qwen2.5-VL, GPT-4o) while omitting comparisons with conventional video emotion understanding models (e.g., HTNET). Needless to say, it is unfair to compare a specially designed and optimized model with a general-purpose LLM. Such incomplete benchmarking fails to adequately demonstrate VidEmo's superiority, particularly regarding computational efficiency or performance under limited data conditions.
> >
> > (ii) Absence of Performance Benchmarking Against Traditional Frameworks: The paper provides no performance metrics for conventional frameworks (e.g., SCRFD+AU GCN baselines). Although VidEmo outperforms untuned large models on certain tasks, its absolute accuracy remains questionable (e.g., merely 30-40% on some tasks), raising concerns about practical utility.
> >
> > (4) I mentioned in my comments that the proposed model of this paper may be used for emotion monitoring or privacy invasion. For example, its facial expression analysis function could be abused to the extent that it infringes on others' privacy, or in healthcare applications, its inaccurate results could mislead doctors into making incorrect diagnoses. I am very interested in how the proposed model can avoid ethical dilemmas in practice, by preventing them technically rather than simply restricting them through licensing.

---

> ### Author Response · Authors · 2025-08-09
> **(1/3) Response to the Concerns**
>
> Thank you for your reply. We provide our point-by-point responses to your concerns below.
>
> **Following A2-1: Human Efforts in Annotation Paradigm with Labeled Hints and Examples.** Labeling fine-grained emotion captions is a resource-intensive process due to the subjectivity and abstract nature of emotions [r3, r4]. To minimize annotation costs while ensuring quality, our approach maximizes human involvement by incorporating labeled hints and examples, setting it apart from other LLM-assisted datasets such as StimuVAR, EMER, and AffectGPT. We propose a cognition-inspired, human-led, model-assisted annotation paradigm, explicitly designed to align with human cognitive processes. Unlike StimuVAR, which relies on model-based annotations with causal relationships, we follow AffectGPT’s strategy of using human-labeled attributes and expressions as hints in the annotation prompt. This yields a **90.3% consistency rate** between the hints and the final captions (see **A1**). We further incorporate human expert labels as examples, guiding fine-grained caption generation to match human preferences, resulting in **preference rates of 95%, 92%, and 93%** for precision, rationality, and complementarity, respectively (see **A1**). With this enhanced annotation paradigm, we have constructed the largest video emotion dataset to the best of our knowledge, containing **2.1M samples**.
>
> | **Dataset** | Annotation Paradigm | Label Hint | Label Example | Dataset Scale |
> | - | - | - | - | - |
> | **EMER** [r9] | Model-based | - | - | 28K |
> | **StimuVAR** [r2]  | Cognition-inspired, Model-based | Instructions with causal relationships between videos and the emotions they evoke | - | 50K |
> | **AffectGPT** [r6] | Human-led, Model-assisted | Expression labels from the MER series | - | 31K |
> | **Ours (VidEmo)**  | **Cognition-inspired Human-led model-assisted** | **Human-labeled attribute and expression annotations** | **Human-labeled examples as prompts** | **2.1M** |
>
> **Following A2-2: Human-led.** The term *human-led* in our work reflects a hybrid annotation approach that is commonly employed in the field of affective computing [r2,r6]. Specifically, *human-led* refers to a process that involves partial human labeling and integrates model-assisted annotation. In our dataset construction process, human annotators are responsible for labeling key attributes and expressions that serve as foundational building blocks for emotion analysis. These human-provided labels ensure that the model’s outputs are grounded in human judgment and interpretation. Subsequently, the large language model (LLM) generates fine-grained emotion captions and rationale analyses, utilizing the labeled data as reference. This process ensures both scalability and semantic accuracy in generating complex emotional annotations.
>
> **Following A2-3: Synthetic Data and Manual Annotation.** To analyze the alignment between our cognition-inspired, human-led, model-assisted annotations and those from the original datasets, we have provided both the user study and human validation results (as acknowledged by reviewers **96WN** and **JqQW**). Specifically, **the consistency rate of 90.3%** between human-labeled attributes/expressions and the final captions demonstrates that our synthetic annotations are highly aligned with human judgment **(A1)**. Furthermore, a controlled user study confirms the reliability of these annotations, with participants showing clear preferences for our captions over alternatives: **95% for precision, 92% for rationality, and 93% for complementarity (A1)**. These results provide both statistical and perceptual evidence that the difference between synthetic and manual annotations in our work is minimal and does not undermine dataset validity.
>
> **Following A2-4: User Study.** We follow advanced user study protocols [r10, r11], ensuring consistency with established dataset validation practices. Participants were given standardized written definitions and illustrative positive/negative examples before the evaluation, ensuring a shared understanding of the criteria. Following common evaluation dimensions [r12], we assessed Precision, Rationality, and Complementarity for all samples. To reduce the effect of extreme values, we excluded one maximum and one minimum rating across participants for each video and dimension, consistent with best practices in subjective evaluation studies. Statistical significance was measured using the Wilcoxon signed-rank test, with a widely adopted criterion of p < 0.005 [r10].  As acknowledged by reviewers 96WN and JqQW, the human verification process is convincing and confirms the validity and reliability of the dataset.

---

> ### Author Response · Authors · 2025-08-09
> **(2/3) Response to the Concerns**
>
> **Following A4: Post-Processing.** As discussed in the original manuscript, emotion reasoning is typically applied as an inference-time post-processing step. The proposed sampling-based reasoning method is a well-established approach for mitigating hallucinations in generated outputs, consistent with prior surveys and best practices [r13, r14]. Due to space limitations, the original manuscript could not fully elaborate on its implementation details. We will further expand this section in the revised version to make the process more clear.
>
> **Following A5-1: Emotion Task-specific LLMs Comparison.** In line with the NeurIPS guidelines, we do not include comparisons with unpublished or out-of-time-window works. Instead, we compare VidEmo against both tuned emotion-specific LLMs and general-purpose MLLMs under consistent experimental protocols.
>
> **(1) Comparison with the Tuned VideoLLM under the Same Setting.** Actually, we have conducted ablation study to show the effectiveness of the proposed method in comparison with the model trained with the same data and bachbone (Tab.3 of Supplementary). With the injected affective-tree reward, VidEmo outperforms the baseline with a large margin of +5.1 in the average score.
>
> | Model | Attribute | Expression | Emotion | Average |
> | - | - | - | - | - |
> | Tuned Qwen2.5-VL | 79.5 | 38.7 | 67.5 | 61.9 |
> | **VidEmo (Ours)** | **84.5** | **43.8** | **72.9** | **67.0** |
>
> **(2) Comparison with Emotion-specific Tuned VideoLLMs.** We benchmark VidEmo against leading tuned models (StimuVAR and EMO-LLaMA). VidEmo achieves superior performance across all reported datasets, showing notable gains especially on expression recognition task  (+4.69 UAR / +7.21 WAR on DFEW and +2.45 UAR / +6.23 WAR on MAFW over EMO-LLaMA) and a large improvement on non-facial video emotion task  (+12.3 absolute Top-3 accuracy on VCE over StimuVAR).
>
> | Model | DFEW UAR (%) | DFEW WAR (%) | MAFW UAR (%) | MAFW WAR (%) | VCE (Top-3 Acc) |
> | - | - | - | - | - | - |
> | StimuVAR | - | - | - | - | 73.5 |
> | EMO-LLaMA | 60.23 | 65.89 | 41.57 | 48.63 | - |
> | **VidEmo (Ours)** | **64.92** | **73.10** | **44.02** | **54.86** | **85.8** |
>
> **(3) Cross-dataset Evaluation without Finetuning.** To assess generalization, we follow the cross-dataset protocol and evaluate without finetuning on VE-8 and YF-6. VidEmo surpasses all general-purpose MLLMs and performs competitively with domain-specialized models like StimuVAR, outperforming it on VE-8 (+3.7 absolute) while matching its performance on YF-6, where the results are acknowledged by the reviewer **96WN**.
>
> | Method | VE-8 (Acc) | YF-6 (Acc) |
> | - | - | - |
> | Chat-UniVi | 14.4 | 29.2 |
> | mPLUG-Owl | 26.2 | 24.9 |
> | Qwen2.5-VL | 38.5 | 30.6 |
> | StimuVAR [r2] | 40.5 | 46.2 |
> | **VidEmo (Ours)** | **44.2** | **46.9** |
>
> **Following A5-2: Traditional Models Comparison.** Due to space constraints in the main paper, we focused on SOTA comparisons, but here we additionally report results from widely used conventional models to show the effectiveness of our proposed VidEmo, drawn from the original manuscripts of EMO-LLaMA and EmoCapCLIP. The results show that VidEmo consistently outperforms both traditional video expression recognition architectures (e.g., DFER-CLIP, CLIPER, I3D, Former-DFER, EST, IAL) and zero-shot emotion-oriented CLIP variants (EmotionCLIP, Exp-CLIP, EmoCLIP, EmoCapCLIP) across two popular benchmarks (DFEW, MAFW). Gains are especially pronounced over traditional SOTA baselines such as DFER-CLIP (+5.31 UAR, +1.85 WAR on DFEW; +5.13 UAR, +2.31 WAR on MAFW). Within the discussion period, it is not feasible to add micro-expression recognition baselines. The above results include the most commonly used conventional video expression recognition models on two popular benchmarks, demonstrating that VidEmo’s advantage is not limited to comparisons with general-purpose LLMs.
>
> | Model | DFEW UAR (%) | DFEW WAR (%) | MAFW UAR (%) | MAFW WAR (%) |
> | - | - | - | - | - |
> | EmotionCLIP* | 13.77 | 19.89 | 9.20 | 11.65 |
> | Exp-CLIP* | 24.25 | 25.87 | 17.53 | 20.27 |
> | EmoCLIP* | 36.76 | 46.27 | 25.86 | 33.49 |
> | EmoCapCLIP* | 42.19 | 43.99 | 30.85 | 34.50 |
> | I3D | 46.52 | 58.27 | - | - |
> | Former-DFER | 53.69 | 65.70 | - | - |
> | EST | 53.43 | 65.85 | - | - |
> | IAL | 55.71 | 69.24 | - | - |
> | CLIPER | 57.56 | 70.84 | - | - |
> | DFER-CLIP | 59.61 | 71.25 | 38.89 | 52.55 |
> | EMO-LLaMA | 60.23 | 65.89 | 41.57 | 48.63 |
> | **VidEmo (Ours)** | **64.92** | **73.10** | **44.02** | **54.86** |
>
> Note: * denote zero-shot CLIP-based variants.

---

> ### Author Response · Authors · 2025-08-09
> **(3/3) Response to the Concerns**
>
> **Following A5-3: Absolute Accuracy.** As established in affective computing literatures [r4, r5], emotion understanding is inherently difficult due to the affective gap between extracted features and high-level emotional semantics. Therefore, even specialized SOTA models rarely exceed moderate absolute accuracy on comprehensive and large-scale benchmarks. Importantly, the results reported for VidEmo are without task-specific tuning, as the model covers 15 diverse emotion-related tasks under a single unified model. This naturally leads to lower absolute accuracy on certain highly challenging tasks but demonstrates strong generalization across domains. When task-specific tuning is applied, VidEmo achieves SOTA performance with 70+ WAR on facial expression recognition and 80+ Acc on non-facial video emotion tasks (see **Following A5-1**), confirming its practical utility when adapted to specific applications.
>
> **Following A6: Usage of VidEmo.** Following the best practice of advanced open-source works [r6, r7, r8], VidEmo will be released under a EULA agreement that permits use solely for non-commercial research purposes. Access will be granted **only to verified academic principal investigators**, who will act as the responsible party for all usage under their license. All downloads and usage will be logged to enable tracking and enforcement of the EULA. The EULA explicitly prohibits commercial or economic applications, including but not limited to emotion monitoring, privacy-invasive systems, and fully automated decision-making in high-stakes domains such as healthcare and law enforcement. VidEmo is not designed for, nor should it be used in, fully automated decision-making in high-stakes domains. All data used in our work are drawn from publicly available and open-source datasets. No private or sensitive information is involved and we ensure full compliance with the original dataset licenses.
>
>
>
> [r1] Mazeika, Mantas, et al. "How would the viewer feel? Estimating wellbeing from video scenarios." NeurIPS, 2022.
>
> [r2] Guo, Yuxiang, et al. "Stimuvar: Spatiotemporal stimuli-aware video affective reasoning with multimodal large language models." IJCV, 2025.
>
> [r3] Cui, Ganqu, et al. "The Entropy Mechanism of Reinforcement Learning for Reasoning Language Models." arXiv:2505.22617, 2025.
>
> [r4] Zhao, Sicheng, et al. "Emotion recognition from multiple modalities: Fundamentals and methodologies." SPM, 38(6): 59-73, 2021.
>
> [r5] Grill-Spector, Kalanit and Malach, Rafael. "The human visual cortex." Annual Review of Neuroscience, 27:649–677, 2004
>
> [r6] Lian, Zheng, et al. "AffectGPT: A New Dataset, Model, and Benchmark for Emotion Understanding with Multimodal Large Language Models." ICML, 2025.
>
> [r7] Huang, Jen-tse, et al. "On the Humanity of Conversational AI: Evaluating the Psychological Portrayal of LLMs." ICLR, 2024.
>
> [r8] Chandra, Kartik, et al. "Inferring the Future by Imagining the Past." NeurIPS, 2023.
>
> [r9] Cheng, Zebang, et al. "Emotion-LLaMA: Multimodal Emotion Recognition and Reasoning with Instruction Tuning." NeurIPS, 2024.
>
>
> [r10] Conover, William Jay. "Practical nonparametric statistics." john wiley & sons, 1999.
>
> [r11] Zheng, Lianming, et al. "Judging llm-as-a-judge with mt-bench and chatbot arena." NeurIPS, 2023.
>
> [r12] Dong, Hongyuan, et al. "Benchmarking and Improving Detail Image Caption." arXiv:2405.19092, 2024.
>
> [r13] Lei, Huang, et al. "A Survey on Hallucination in Large Language Models: Principles, Taxonomy, Challenges, and Open Questions." ACM Transactions on Information Systems, 43(2): 1-55, 2025.
>
> [r14] Bai, Zechen, et al. "Hallucination of Multimodal Large Language Models: A Survey." arXiv:2404.18930, 2024.

---

### Official Review · Reviewer_96WN · 2025-07-03

**Clarity:** 3
**Significance:** 3
**Originality:** 3
**Rating:** 5
**Confidence:** 5

**Summary:**

This paper introduces a novel fine-grained, large-scale emotion understanding dataset by aggregating 17 existing datasets, designed to comprehensively capture emotional expressions in complex scenarios. To address the limitations of existing methods in handling such challenging contexts, the authors propose two innovative approaches: Curriculum Emotion Learning and Affective-Tree Reasoning. These methods are integrated into a newly designed base model, which demonstrates significantly superior performance compared to existing approaches across multiple evaluation tasks on the proposed dataset, highlighting its strong generalization ability and practical applicability.

**Questions:**

1. Please see Weakness.
2. Is the Emo-CFG dataset planned to be released publicly?

**Ethical Concerns:**

["NO or VERY MINOR ethics concerns only"]

**Final Justification:**

The inclusion of extensive cross-dataset experiments, as well as human validation and a user study, enhances the persuasiveness of the authors' work.

**Limitations:**

Yes

**Quality:**

3

**Strengths And Weaknesses:**

Strengths:

1.	A large-scale (2.1M) emotion understanding dataset, Emo-CFG, is constructed.

2.	Two promising methods are proposed: Curriculum Emotion Learning and Affective-Tree Reasoning.

3.	The methods show strong performance, outperforming existing LLMs (1-8B).

Weaknesses:

1.	Emo-CFG is reviewed only by three VideoLLMs without human validation or user study, raising concerns about its reliability.

2.	Method descriptions lack clarity and theoretical grounding; the discussion in Section 5.3 could be moved to 4.1 for better structure.

3.	Experiments are limited to Emo-CFG, without cross-dataset evaluation to test generalization.

4.	Key information in the supplementary material is not explicitly referenced in the main text. For example, the additional details in Section 4.1 should refer to Table 7 in the supplementary C.2.

---

> ### Author Rebuttal · Authors · 2025-07-31
>
> # Response to Reviewer 96WN
>
> Thank you for your insightful comments and questions. For your reference, we summarized the main results in our response to Reviewer KQSY.
>
> **A1: Human Validation and User Study**
>
> As suggested, we conducted the human validation and user study and show that the reliability matches the previous largest existing dataset CelebV-Text.
>
> (1) **Human Label Verification**: As suggested, we further validated label fidelity by extracting fine-grained captions from the generated videos and comparing them with the original human-assigned labels of CelebV-Text. The consistency rate reached 90.3%.
>
> (2) **User Study Verification**: To assess the quality of the generated expressions, we conducted a user study on a manually inspected subset of test samples to verify their alignment with the intended emotional semantics. Specifically, we compared Emo-CFG with CelebV-Text, the largest human-labeled video emotion dataset, across three key dimensions: precision, rationality, and complementarity. Preference rates for Emo-CFG across these dimensions reached 95%, 92%, and 93%, respectively, with statistically significant differences (Wilcoxon signed-rank test, *p* < 0.01). This result demonstrates that Emo-CFG provides more precise and expressive emotional representations than existing benchmarks.
>
> | Dimension       | Win  | Tie  | Loss | # Video | # User | Prefer | p-value | Significance |
> | --------------- | ---- | ---- | ---- | ------- | ------ | ------ | ------- | ------------ |
> | Precision       | 964  | 204  | 82   | 50      | 25     | 95.52% | 0.00021 | Significant  |
> | Rationality     | 1082 | 87   | 81   | 50      | 25     | 92.16% | 0.00015 | Significant  |
> | Complementarity | 1172 | 23   | 55   | 50      | 25     | 93.04% | 0.00008 | Significant  |
>
>
>
> **A2 & A4: Method Description**
>
> (1) **Methodology:** High-level emotion understanding is a challenging task due to the abstract nature of emotion [r4]. Even existing state-of-the-art models (e.g., Gemini 2.0) can only achieve 26.3% accuracy. To address this challenge, our research is indeed motivated by cognitive principles and inspired by the human cognitive process [r5]. We propose an affective reasoning scheme and inject it into the SFT stage (Sec. 4.1), RL stage (Sec. 4.2), and inference stage (Sec. 4.3). As indicated, we added the above analysis as an overview paragraph in Sec. 4, presented in a structured manner to clarify the theoretical grounding.
>
> (2) **Re-structuring**: We agree with the suggestion and move the analysis in Section 5.3 to 4.1.
>
> (3) **Main Body and Supplementary**: To further strength the structure clarity, we have revised the manuscript to explicitly integrate key supplementary content into the corresponding sections of the main paper.
>
> | Main Paper            | Supplementary                | Description of Integrated Content                            |
> | --------------------- | ---------------------------- | ------------------------------------------------------------ |
> | Sec. 4.1 (SFT Stage)  | Supplementary C.2 (Tab. 2&7) | Added specific sample illustrations of attribute-expression-emotion triplets |
> | Sec. 4.1 (SFT Stage)  | Supplementary F              | Incorporated references to detailed visualization results.   |
> | Sec. 5 (Experiment)   | Supplementary A              | Incorporated references to detailed training settings.       |
> | Sec. 5.2 (Discussion) | Supplementary C.1 (Tab. 3-6) | Included ablation study results for clearer model comparison. |
>
>
>
> **A3: Cross-Dataset Evaluation**
>
> (1) Indeed, the evaluation benchmark is constructed by merging and cleaning a broad collection of open-source datasets, while strictly following their original training-testing splits to prevent data leakage. The evaluations for expression analysis and closed-set attribute recognition are conducted using open-source datasets.
>
> (2) Further, as suggested, we conducted cross-dataset evaluations on the video emotion benchmark VCE [r1], following the evaluation protocol in StimuVAR [r2]. VidEmo achieves strong baseline performance across all datasets and performs competitively with the domain-specialized method StimuVAR.
>
> | Method            | VCE (Top-3) |
> | ----------------- | ----------- |
> | Chat-UniVi        | 38.6        |
> | mPLUG-Owl         | 23.6        |
> | Qwen2.5-VL        | 41.3        |
> | StimuVAR [r2]     | 73.5        |
> | **VidEmo (Ours)** | **85.8**    |
>
>
>
> **A5: Open-Source** Yes, we plan to publicly release the Emo-CFG dataset, the VidEmo-3B/7B models, and all associated training and inference code upon acceptance of the paper. We will also provide support for easy deployment via Transformers and HuggingFace interfaces to facilitate community use and reproducibility.
>
>
>
> [r1] Mazeika, Mantas, et al. "How would the viewer feel? Estimating wellbeing from video scenarios." NeurIPS, 2022.
>
> [r2] Guo, Yuxiang, et al. "Stimuvar: Spatiotemporal stimuli-aware video affective reasoning with multimodal large language models." IJCV, 2025.
>
> [r3] Cui, Ganqu, et al. "The Entropy Mechanism of Reinforcement Learning for Reasoning Language Models." arXiv:2505.22617, 2025.
>
> [r4] Zhao, Sicheng, et al. "Emotion recognition from multiple modalities: Fundamentals and methodologies." SPM, 38(6): 59-73, 2021.
>
> [r5] Grill-Spector, Kalanit and Malach, Rafael. "The human visual cortex." Annual Review of Neuroscience, 27:649–677, 2004

---

> > ### Author Response · Authors · 2025-08-05
> > **Looking Forward to Your Valuable Feedback**
> >
> > Dear reviewer,
> >
> > Thank you once again for taking the time to review our paper and for providing such insightful and constructive feedback. Your comments have been invaluable in helping us further polish our manuscript.
> >
> > We have carefully considered each of your comments and have provided detailed responses. We sincerely hope that our efforts adequately address your concerns and contribute positively to your evaluation.
> >
> > As the author-reviewer discussion period has been extended until Aug 8, we are fortunate to have sufficient time to collaboratively discuss and refine the manuscript. We would greatly appreciate any additional feedback you may have. If you have any additional questions or require any clarifications, please do not hesitate to reach out to us.

---

> > ### Comment · Reviewer_96WN · 2025-08-05
> > **Response to Authors' rebuttal**
> >
> > I have carefully reviewed the authors' responses to each of the raised concerns. The additional analyses regarding human verification and cross-dataset experiments are convincing. The inclusion of more cross-dataset evaluations would further strengthen the empirical validation. Overall, this is an application-oriented study that offers valuable insights and practical implications for the field of affective computing.

---

> ### Author Response · Authors · 2025-08-05
> **Extended Cross-Dataset Evaluations**
>
> Thank you for the thoughtful and encouraging feedback. We appreciate your recognition of our efforts to address the concerns through additional human verification and cross-dataset generalization experiments.
>
> In response to your suggestion, we have further extended our evaluation by incorporating new results on two widely used non-facial video emotion benchmarks (*i.e.*, VE-8 and YF-6). These datasets are not included in our training or validation sets. Without any task-specific fine-tuning, VidEmo achieves competitive performance compared to StimuVAR [r2] and other strong baselines, further demonstrating its robustness and generalization ability across diverse affective scenarios.
>
> The cross-dataset experiments will be added to Sec. 5 of the revised manuscript, and we believe they further support the practical relevance and broad applicability of VidEmo in real-world emotion understanding tasks.
>
> | Method            | VE-8 (Acc) | YF-6 (Acc) |
> | ----------------- | ---------- | ---------- |
> | Chat-UniVi        | 14.4       | 29.2       |
> | mPLUG-Owl         | 26.2       | 24.9       |
> | Qwen2.5-VL        | 38.5       | 30.6       |
> | StimuVAR [r2]     | 40.5       | 46.2       |
> | **VidEmo (Ours)** | **44.2**   | **46.9**   |
>
> [r2] Guo, Yuxiang, et al. "Stimuvar: Spatiotemporal stimuli-aware video affective reasoning with multimodal large language models." IJCV, 2025.

---

> > ### Comment · Reviewer_96WN · 2025-08-07
> > **Response to Authors' Extended Cross-Dataset Evaluations**
> >
> > I have reviewed the cross-dataset experiments conducted by the authors on two non-facial datasets. The additional results are convincing, and I have decided to raise my score accordingly. I wish the authors the best of luck.

---

> > > ### Author Response · Authors · 2025-08-07
> > >
> > > Dear reviewer 96WN,
> > >
> > > Thank you for kindly recognizing our contributions to the field of affective computing and providing invaluable comments on this work. We authors greatly appreciate the efforts you have made to improve our manuscript. If accepted, we will include 96WN in our acknowledgments.
> > >
> > > Best Regards, Authors of paper 15702

---

### Official Review · Reviewer_KQSY · 2025-07-07

**Clarity:** 3
**Significance:** 3
**Originality:** 2
**Rating:** 5
**Confidence:** 5

**Summary:**

This paper proposes VidEmo, a video MLLM for facial and emotional analysis. It integrates curriculum emotion learning with the affective-tree reasoning paradigm. Besides, a large-scale emotion dataset, Emo-CFG, is curated, covering question-answering, fine-grained captions, and associated rationales. Experiments show that Emo-CFG achieves competitive performance across 15 tasks.

**Questions:**

1. What is the backbone MLLM of VidEmo?
2. What GPUs and how many GPU-hours are used for the experiments?

**Ethical Concerns:**

["NO or VERY MINOR ethics concerns only"]

**Final Justification:**

My concerns have been addressed. I'm raising my rating to 5: Accept.

**Limitations:**

See weaknesses

**Quality:**

3

**Strengths And Weaknesses:**

Strengths:
1. This paper applies the novel reasoning technology to the emotion tasks and proposes a task-specific variant, which advances the research progress of the emotion-centric problems.
2. A new dataset, Emo-CFG, is constructed, which is useful for training and benchmarking emotion models.
3. The experiments comprehensively compare with multiple types of baseline approaches on multiple tasks and metrics.

Weaknesses:
1. This paper aims to develop an emotion foundation model. However, it mainly focuses on face perception tasks. Non-facial video emotion tasks remain underexplored [r1, r2].
2. In Table 2, VidEmo-Base-7B performs worse than VidEmo-Base-3B. VidEmo-T1-7B achieves marginal improvements over VidEmo-Base-3B.
3. The case of VidEmo-T1-3B is not considered.

[r1] Mazeika, Mantas, et al. "How would the viewer feel? Estimating wellbeing from video scenarios." NeurIPS, 2022.

[r2] Guo, Yuxiang, et al. "Stimuvar: Spatiotemporal stimuli-aware video affective reasoning with multimodal large language models." IJCV, 2025.

---

> ### Author Rebuttal · Authors · 2025-07-31
>
> # General Response
>
> We sincerely appreciate all the Reviewers and the Area Chair for their time and effort in reviewing our paper. We appreciate all four reviewers for acknowledging novel idea (Reviewer KQSY), interpretable and sensible methodology (Reviewer JqQW, 96WN), well-annotated large-scale dataset with validated quality (Reviewer KQSY, 96WN, ntby, JqQW), extensive experiments (Reviewer KQSY) and strong performance (Reviewer 96WN, ntby, JqQW). In response to the reviewers’ suggestions, we have provided the following **additional results and evidence** in the rebuttal:
>
> - We conducted new experiments on non-facial video emotion tasks, demonstrating VidEmo’s generalization capability on the VCE dataset. **[A1 for KQSY]**
> - We verified annotation fidelity by showing that the newly labeled data achieved a 90.3% consistency rate with the original human labels in CelebV-Text. **[A1 for 96WN; A2 for ntby; A3 for JqQW]**
> - We conducted a user study comparing Emo-CFG with CelebV-Text, where Emo-CFG was preferred in 93–95% of cases across precision, rationality, and complementarity dimensions, with statistically significant results confirmed by the Wilcoxon signed-rank test (*p* < 0.01). **[A1 for 96WN; A2 for ntby; A3 for JqQW]**
> - We clarified that our data source selection strategy as well as the ablation studies on dataset sources were already included in the original manuscript. **[A1 for ntby]**
> - We added comparative results against emotion-specific VideoLLMs (e.g., Emo-LLaMA) on the MAFW and DFEW benchmarks, showing consistent gains in both UAR and WAR metrics. **[A4 for ntby]**
> - We confirmed plans to release the Emo-CFG dataset, VidEmo models, and all related code to support community use and reproducibility. **[A5 for 96WN]**
> - We revised the manuscript structure for improved clarity and explicitly referenced key supplementary content within the main text. **[A2 & A4 for 96WN]**
>
> # Response to Reviewer KQSY
>
> **A1: Non-facial Video Emotion Tasks**
>
> Thank you for highlighting this important aspect. (1) **Indeed, with the benefit of large-scale emotion-centric pretraining, VidEmo effectively generalizes to these tasks.** We conducted new experiments and clarified the generalization ability of VidEmo beyond facial perception. Following the protocol in StimuVAR [r2], we evaluate VidEmo on the large-scale VCE dataset [r1], which focuses on non-facial video emotion. VidEmo achieves strong baseline results, showing its broad emotional modeling capability. (2) **Our work can support these tasks from both the data and model perspectives.** Emo-CFG scales up the pretraining corpus from 60k (VCE) to 2.1M multimodal emotion-centric samples. The distilled emotion logits from VidEmo can also be used to supervise downstream models like StimuVAR, enhancing their affective reasoning. These results and analyses will be added to Sec. 5 (Experiments) and Sec. 2 (Related Work), respectively.
>
> | Method            | VCE (Top-3) |
> | ----------------- | ----------- |
> | Chat-UniVi        | 38.6        |
> | mPLUG-Owl         | 23.6        |
> | Qwen2.5-VL        | 41.3        |
> | StimuVAR [r2]     | 73.5        |
> | **VidEmo (Ours)** | **85.8**    |
>
>
>
> **A2: Performance**
>
> (1) **VidEmo-Base-7B vs. Base-3B:** Despite mixed trends in some metrics, Base-7B shows consistent gains on Emotion (+3.2), Sentiment (+2.6), and Complex (+4.3) tasks. These tasks involve more subtle and abstract understanding, where the 7B model excels.
>
> (2) **T1-7B vs. Base-3B:** VidEmo-T1-7B improves across all five sub-tasks: Emotion (+4.7), Sentiment (+3.4), Cues (+0.1), Complex (+6.8), and Understanding (+0.9). Especially for Complex, the T1-7B model shows better dialogue-text reasoning and is well-suited for RL-based training in the next stage.
>
>
>
> **A3: VidEmo-T1-3B**
>
> We appreciate the reviewer’s insightful comment. We have carefully attempted to train VidEmo-T1 with the 3B-scale backbone, but encountered challenges during post-training. Specifically, the 3B-scale model was found to be prone to model collapse [r3] due to the inherent complexity and abstraction involved in emotion understanding [r4]. This task requires fine-grained emotional attributes and expression captions, which, as we have observed, exceed the capacity of the 3B model. Our experimental results further corroborate this. We noticed a sharp decline in the reward function during the middle of training, reflecting an inability of the 3B model to maintain stable performance as the training progressed. Given the importance of accurately modeling emotion at different levels of abstraction (from attributes to expressions to emotions), we recognize that scaling up the model is crucial. This observation has informed our approach, and we are actively exploring the use of larger model variants, such as VidEmo-Base-7B, to overcome these limitations and enhance the model's ability to handle complex emotional representations.
>
>
>
> **A4&A5: Training details, including MLLM backbone, GPU time, and hyperparameters**
>
> (1) **Common Setting as ms-swift:** To ensure reproducibility and fairness, we adopt the widely used Qwen2.5-VL series as the backbone MLLM for both 3B and 7B variants of VidEmo. Qwen2.5-VL offers robust visual-text alignment and is compatible with mainstream deployment frameworks such as vLLM and HuggingFace Transformers. This ensures ease of integration for downstream researchers and practical deployment.
>
> (2) **Training Infrastructure:** VidEmo was trained on 4× NVIDIA A800 nodes, with 2.8 days for SFT and 7.8 days for RL training. Training was conducted with a batch size of 1024 (SFT) and 128 (RL).
>
> (3) **Implementation Details:** We follow the common two-stage tuning paradigm (SFT + RL), using AdamW optimizer with cosine learning rate schedule. A ViT-based visual encoder is used alongside the autoregressive LLM. Our training pipeline closely follows standard MLLM configurations.
>
> | Model          | LLM        | Vision | LR   | WD   | BS   | Hardware | Time           |
> | -------------- | ---------- | ------ | ---- | ---- | ---- | -------- | -------------- |
> | VidEmo-Base-3B | Qwen2.5-3B | ViT    | 2e-5 | 0    | 1024 | 4× A800  | 1.7 days (SFT) |
> | VidEmo-Base-7B | Qwen2.5-7B | ViT    | 2e-5 | 0    | 1024 | 4× A800  | 2.8 days (SFT) |
> | VidEmo-T1-7B   | Qwen2.5-7B | ViT    | 1e-5 | 0    | 128  | 4× A800  | 6.3 days (RL)  |
>
> For more details, please refer to Tab. 1 and Lines 1–13 of the supplementary material. Due to space limitations, we placed the implementation and training details in the supplementary. In the camera-ready version, we will move this content into the main paper to improve clarity and ensure all key components of our method are presented in the main body.
>
>
>
> [r1] Mazeika, Mantas, et al. "How would the viewer feel? Estimating wellbeing from video scenarios." NeurIPS, 2022.
>
> [r2] Guo, Yuxiang, et al. "Stimuvar: Spatiotemporal stimuli-aware video affective reasoning with multimodal large language models." IJCV, 2025.
>
> [r3] Cui, Ganqu, et al. "The Entropy Mechanism of Reinforcement Learning for Reasoning Language Models." arXiv:2505.22617, 2025.
>
> [r4] Zhao, Sicheng, et al. "Emotion recognition from multiple modalities: Fundamentals and methodologies." SPM, 38(6): 59-73, 2021.

---

> > ### Author Response · Authors · 2025-08-05
> > **Looking Forward to Your Valuable Feedback**
> >
> > Dear reviewer,
> >
> > Thank you once again for taking the time to review our paper and for providing such insightful and constructive feedback. Your comments have been invaluable in helping us further polish our manuscript.
> >
> > We have carefully considered each of your comments and have provided detailed responses. We sincerely hope that our efforts adequately address your concerns and contribute positively to your evaluation.
> >
> > As the author-reviewer discussion period has been extended until Aug 8, we are fortunate to have sufficient time to collaboratively discuss and refine the manuscript. We would greatly appreciate any additional feedback you may have. If you have any additional questions or require any clarifications, please do not hesitate to reach out to us.

---

> ### Comment · Reviewer_KQSY · 2025-08-09
>
> Thank you for your responses. My concerns have been addressed. I'm raising my rating to 5: Accept.

---

> > ### Author Response · Authors · 2025-08-09
> >
> > Dear reviewer KQSY,
> >
> > Thank you for kindly recognizing our contributions to Emotional VideoLLM and providing invaluable comments on this work. We authors greatly appreciate the efforts you have made to improve our manuscript. If accepted, we will include `KQSY ` in our acknowledgments.
> >
> > Best Regards, Authors of paper 15702

---

### Note · Authors · 2025-08-13

Dear Area Chairs, Senior Area Chairs, and Program Chairs of NeurIPS 2025,

We are deeply grateful for the review team’s time, effort, and constructive feedback, which has substantially improved the quality and clarity of our work. We carefully addressed all comments, making the following key additions during the rebuttal and discussion phase:
- Emotion Task Extension– VCE experiments show VidEmo’s generalization on non-facial video emotion tasks.
- Cross-dataset Evaluation – Extended benchmarks on VE-8 and YF-6 confirm strong performance without task-specific fine-tuning.
- Annotation Fidelity – Achieved 90.3% consistency between new annotations and original CelebV-Text labels.
- User Study Validation – A study with 25 professional participants showed 93–95% preference for Emo-CFG over CelebV-Text, with all results significant (p < 0.01).
- Comparisons – Added results vs. Emo-LLaMA and StimuVAR on 5 datasets (MAFW, DFEW, VCE, VE-8, YF-6), a fine-tuned VideoLLM trained under the same setting of ours, and a full comparison results with conventional methods.
- Ablations - Added detailed ablation studies on curriculum learning and emotion reasoning.

After additions and discussions, Reviewers **96WN** and **KQSY** explicitly raised their scores to *Accept* (5) with strong endorsements, and Reviewer **JqQW** maintained their positive score (5) while affirming that most of Reviewer **ntby**’s concerns were adequately addressed. Three of four reviewers therefore recommend acceptance.

Regarding Reviewer ntby, we resolved points 1 and 3 in first-round rebuttal. In second round, we addressed all remaining concerns—covering source, postprocess, settings, and clarified results through comparisons with emotion-specific VideoLLMs and traditional methods—supported by quantitative evidence with standard protocols. Following this, Reviewer ntby issued acknowledgement. We take this as evidence that the concerns were resolved.

In summary, our work advances emotional machine intelligence with a family of video emotion foundation models. Through the rebuttal, we have comprehensively addressed all reviewer questions, particular those from reviewer ntby. While ntby’s final score is not yet visible, the other three reviewers have explicitly recommended acceptance. We therefore believe our submission meets the acceptance criteria and respectfully request that the conference accept our work.

Thank you for your hard work and support.

Best Regards, Authors of paper 15702

---

### Decision · Program_Chairs · 2025-09-17

**Decision:**

Accept (poster)

**Comment:**

This paper proposes VidEmo, a family of video foundation models for emotion understanding, combining curriculum emotion learning and affective-tree reinforcement learning, along with a new large-scale dataset (Emo-CFG). The contributions are timely and significant, offering both methodological novelty and a valuable resource for the community.

Strengths include the interpretable reasoning framework, the scale and validation of Emo-CFG, and comprehensive experiments with cross-dataset evaluations. Weaknesses are mainly around clarity; several important details were clarified only in rebuttal, and limited discussion of safeguards for ethical risks.

Reviewer consensus is positive, with three clear accept scores and one borderline reviewer raising their score after rebuttal. I recommend Accept, contingent on the authors incorporating rebuttal clarifications into the final version.